# A Kinetic Energy Perspective of Flow Matching

**Ziyun Li** [* 1]   **Huancheng Hu** [2]   **Soon Hoe Lim** [1 3]   **Xuyu Li** [4]   **Fei Gao** [5]   **Enmao Diao** [6]   **Zezhen Ding** [7]
**Michalis Vazirgiannis** [8 9]   **Henrik Boström** [1]

## Abstract

Flow-based generative models can be viewed through a physics lens: sampling transports a particle from noise to data by integrating a learned velocity field, and each sample corresponds to a trajectory with its own dynamical effort. Motivated by classical mechanics, we introduce *Kinetic Path Energy (KPE)*, an action-like, per-sample diagnostic that measures the accumulated kinetic effort along an ordinary differential equation (ODE) trajectory. Empirically, KPE exhibits two robust correspondences: (*i*) higher KPE predicts stronger semantic fidelity; (*ii*) high-KPE trajectories land in sparse representation regions. We further provide theoretical guarantees linking trajectory energy to data sparsity. Paradoxically, this correlation is non-monotonic. At sufficiently high energy, generation can degenerate into memorization. Leveraging the closed-form formula of empirical flow matching, we show that extreme energies drive trajectories toward near-copies of training examples. This yields a *Goldilocks principle* and motivates *Kinetic Trajectory Shaping (KTS)*, a training-free two-phase inference strategy that boosts early motion and enforces a late-time soft landing, reducing memorization and improving generation quality across benchmark tasks.

## 1. Introduction

Flow-based generative models synthesize data by integrating a learned velocity field $v_\theta$, which defines a bridge that transports samples from a base (noise) distribution to the data distribution (Lipman et al., 2022; Liu et al., 2022; Song et al., 2021). Yet we still lack tools to understand *why individual samples differ in quality*. Standard metrics like FID (Heusel et al., 2017) are fundamentally trajectory-blind; they aggregate global statistics but overlook the dynamics of individual paths (Jayasumana et al., 2024). Sampling, however, can be viewed as navigation in a time-varying flow, where each sample is a particle continuously steered toward the data manifold (Chen et al., 2018; Song et al., 2021). In physics, the accumulation of kinetic energy along a path (the action) is a definitive measure of dynamical effort (Goldstein et al., 1950; Benamou & Brenier, 2000). An analogous per-sample quantity is readily available during flow-based sampling (Finlay et al., 2020; Tong et al., 2024), yet its connection to generation quality remains unexplored. This raises the question: *Does the kinetic effort expended reveal the intrinsic properties of the generated sample?*

To address this, we formalize *Kinetic Path Energy (KPE)* as the time integral of the squared velocity along a sample's trajectory $(x(t))_{t\in[0,1]}$, and is defined as $E := \frac{1}{2}\int_0^1 \|v_\theta(x(t),t)\|^2 \, dt$. KPE provides a zero-overhead diagnostic of individual transport efficiency. It is computed directly during ODE-based sampling, transforming complex flow dynamics into a scalar "sampling cost" that enables granular analysis of individual generation paths.

At a high level, KPE exhibits two key correspondences with the generated data (§4). *(i) Energy as a Proxy for Semantic Fidelity.* Higher-energy trajectories produce samples with sharper, class-specific features (Figure 1; §4.1): synthesizing precise semantic structure demands sustained high velocity, and hence greater accumulated energy. *(ii) Energy as a Proxy for Local Sparsity.* High-energy trajectories terminate in locally sparse regions of representation space, i.e., neighborhoods with few training samples (§4.2). Under a *posterior dominance regime*[1], instantaneous squared speed is affinely bounded by the negative log-density of the bridge mixture $\hat{p}_t(z)$ (Theorem 4.2). Together, these results establish KPE as a *dual indicator* of semantic fidelity

---

[1]Posterior dominance: for each $(z,t)$, there exists a dominant component $i^*$ with posterior weight $\lambda_{i^*}(z,t) \geq 1 - \varepsilon$ for some $\varepsilon \in (0, 1/2)$, where $\lambda_i(z,t) \propto p_t(z \mid x^{(i)})$.

---

*Corresponding author. [1]KTH Royal Institute of Technology [2]Hasso Plattner Institute, University of Potsdam [3]Nordita, Nordic Institute for Theoretical Physics [4]Trinity College Dublin [5]Hangzhou Institute of Technology, Xidian University [6]DreamSoul [7]The Hong Kong University of Science and Technology [8]Mohamed bin Zayed University of Artificial Intelligence [9]École Polytechnique. Correspondence to: Ziyun Li <ziyli@kth.se, liziyun2014@gmail.com>.

*Proceedings of the 43rd International Conference on Machine Learning*, Seoul, South Korea. PMLR 306, 2026. Copyright 2026 by the author(s).

and local sparsity, a path-level diagnostic inaccessible to endpoint-only metrics.

A natural follow-up question arises: *Does pushing energy higher always improve generation?* Paradoxically, the closed-form empirical flow matching (EFM) solution achieves $1.3\times - 3.9\times$ higher peak power than neural velocity fields, yet produces near-exact training replicas (98% memorization on CelebA; §5). We show that this failure is structural: the EFM velocity field contains a singular component that drives energy blow-up and forces trajectories to collide with discrete training atoms (Proposition 5.2). In short, energy is not a monotone knob: at the extreme, energy spikes drive *memorization*, not better generation.

These results suggest a *Goldilocks principle*: generation quality benefits from *moderate, well-timed* kinetic effort, whereas insufficient energy leads to trapping in dense regions, and excessive late-time energy induces terminal blow-up and memorization. Guided by this principle, we propose *Kinetic Trajectory Shaping (KTS)*, a training-free inference strategy with phase-specific velocity modulation (§6). In the early phase ($t < 0.6$), *Kinetic Launch* boosts velocity to raise KPE and pushes samples toward sparse, semantically rich regions. In the late phase ($t \geq 0.6$), *Kinetic Soft Landing* dampens velocity to suppress terminal singularities and prevents memorization. Experiments on CelebA demonstrate that KTS reduces memorization by 16% (from 37.3% to 31.2%) while improving generation quality (FID 14.35 vs. 16.68 baseline).

We summarize our main contributions as follows:

- We propose *Kinetic Path Energy (KPE)*, a per-sample, path-level diagnostic that quantifies the kinetic effort accumulated along a generation trajectory.

- We show empirically that KPE tracks both semantic fidelity and local sparsity in representation space, and we formalize the latter via an energy–density relation on the bridge mixture: $\left\|\hat{u}^*(z,t)\right\|^2 \asymp -\log \hat{p}_t(z)$ under posterior dominance (Theorem 4.2).

- We uncover an *energy paradox* in the regression-optimal EFM due to a structural $1/(1-t)$ terminal singularity that drives memorization (Proposition 5.2). We then address this with a phase-aware remedy called *Kinetic Trajectory Shaping (KTS)*, and evaluate its effectiveness in reducing memorization and improving sample quality across benchmark tasks.

## 2. Related Work

Flow matching learns a time-dependent velocity field whose ODE transport maps a base distribution to the data distribution (Lipman et al., 2022; Liu et al., 2022; Albergo & Vanden-Eijnden, 2023; Lipman et al., 2024). It can be viewed as a deterministic counterpart to diffusion's SDE formulations (Song et al., 2021). Our focus here is on flow matching and its empirical counterpart. Energy and action functionals are central in optimal transport and kinetic formulations of probability evolution, where probability paths are characterized via kinetic energy or action minimization (Benamou & Brenier, 2000; Shaul et al., 2023). In contrast to optimal or distribution-level analyses, we introduce KPE as a *per-sample, path-level* diagnostic computed along individual flow matching trajectories.

Recent work has studied memorization and generalization in flow matching and closely related approaches (Gao & Li, 2024; Bertrand et al., 2025; Baptista et al., 2025; Bonnaire et al., 2025; Scarvelis et al., 2023; Yoon et al., 2023; Pidstrigach, 2022). We complement these analyses by identifying a trajectory-level mechanism: the regression-optimal empirical flow matching solution exhibits a terminal velocity singularity that induces excessive late-time kinetic energy and drives memorization. Several training-free methods modify inference dynamics using classifier or energy-based signals (Ho & Salimans, 2022; Yu et al., 2023; Xu et al., 2024). Unlike approaches that modulate scores or endpoint objectives, our *Kinetic Trajectory Shaping* directly controls the velocity field over time, enabling phase-specific regulation of kinetic effort within flow matching models. See Appendix A for a more detailed discussion of related work.

## 3. Kinetic Analogy and Trajectory Energy

### 3.1. Conditional Flow Matching (CFM)

In CFM, we first construct a target conditional probability path from noise $\epsilon \sim p_0 := \mathcal{N}(0, I)$ to data $z \sim p_{\text{data}}$ over the time interval $[0, 1]$. We adopt the standard linear interpolation path:

$$x(t) = (1 - t)\,\epsilon + t\,z, \quad \epsilon \sim \mathcal{N}(0, I), \tag{1}$$

which defines a conditional flow with the vector field $u_t(x|z) = z - \epsilon$ (conditional on the data $z$). We then learn this velocity field using a neural network $v_\theta(x, t)$ by minimizing the regression loss:

$$\mathcal{L}(\theta) = \mathbb{E}_{t,z,\epsilon}[\left\|v_\theta(x(t), t) - (z - \epsilon)\right\|^2], \tag{2}$$

where $t \sim \mathcal{U}[0, 1]$. The population optimum recovers the conditional expectation $v^\star(x, t) = \mathbb{E}[z - \epsilon \mid x(t) = x]$.

### 3.2. Physical Motivation

In classical mechanics, the evolution of a physical system is characterized by the *action functional* (Goldstein et al., 1950; Feynman & Hibbs, 1965):

$$S[x(\cdot)] = \int_{t_0}^{t_1} L(x(t), \dot{x}(t), t)\, dt, \tag{3}$$

defined over the time interval $[t_0, t_1]$, where $L(x(t), \dot{x}(t), t) = T(\dot{x}(t)) - V(x(t))$ is the Lagrangian,

given by the difference between the kinetic energy $T(\dot{x})$ and the potential energy $V(x)$. This formulation embodies Hamilton's principle of least action (Goldstein et al., 1950) and Feynman's path integral framework (Feynman & Hibbs, 1965). For a free particle (i.e., when $V(x) = 0$), the action functional reduces to the kinetic term, $S_{\text{free}} = \int_{t_0}^{t_1} \frac{1}{2}\|\dot{x}(t)\|^2 \, dt$. This kinetic form is a fundamental example of an action functional in physics. Inspired by similar analogies (Zhang & Chen, 2023), we adopt a kinetic framework to define trajectory-level diagnostics to gain insight into the generation process in flow matching.

### 3.3. Kinetic Path Energy (KPE)

We interpret the flow matching sampling process as a particle moving through a velocity field. The sampling trajectory is governed by the learned velocity field $v_\theta(x, t)$ via the ODE (Liu et al., 2022; Lipman et al., 2022; Song et al., 2020; Chen et al., 2018):

$$\frac{dx}{dt} = v_\theta(x(t), t), \quad t \in [0, 1], \quad x(0) \sim \mathcal{N}(0, I), \quad (4)$$

which describes the probability flow (Song et al., 2021) from noise to data distribution. The trajectory represents the path of a particle driven by the velocity field.

Inspired by classical mechanics (Goldstein et al., 1950), we define *kinetic path energy* $E$ as:

$$E := \frac{1}{2} \int_0^1 \|v_\theta(x(t), t)\|^2 \, dt, \quad (5)$$

where we adopt the convention of unit mass ($m = 1$), standard in the free particle action formulation. KPE encapsulates the cumulative kinetic cost incurred during sampling. It quantifies the "energy" required to transport a sample from the noise distribution to the data manifold. A higher $E$ signifies that the model employs greater velocity magnitudes on average, reflecting a more energetically demanding generation process. We stress that $E$ is a *kinetic-inspired diagnostic*, not literally representing physical energy.

Practically, KPE is cheap to compute: during ODE sampling we simply accumulate $\|v_\theta(x(t), t)\|^2$ at each discrete time, adding negligible overhead. In expectation, when the learned flow realizes optimal transport (Tong et al., 2024; Pooladian et al., 2023), KPE coincides with the Benamou-Brenier dynamic formulation (Benamou & Brenier, 2000) of the 2-Wasserstein distance, grounding our metric in optimal transport theory (Villani et al., 2008).

## 4. Two Findings on KPE

### 4.1. KPE vs. Semantic Strength

Finding 1: Higher $E$ consistently correlates with stronger semantic alignment and discriminability.

**Setup and Metrics.** We examine this relationship on ImageNet-256 using pretrained SiT-XL/2 (Ma et al., 2024), generating 5,000 samples per CFG scale $\omega \in \{1.0, 1.5, 4.0\}$ and partitioning into low/mid/high KPE groups (0–33%, 33–67%, 67–100%). We evaluate using CLIP score (semantic alignment) and CLIP margin (semantic discriminability). See Appendix E for details.

**Results.** Figure 1 provides a qualitative comparison using paired samples from the same class (top row: higher KPE; bottom row: lower KPE). Higher-energy samples exhibit clearer, more class-specific semantic cues; see Appendix K for additional visualizations. Figure 2a and Figure 2b show that both CLIP score and CLIP margin increase with $E$ across different CFG settings. For instance, at CFG=1.5, the median CLIP score increases from 23.52 to 25.12 and the median CLIP margin rises from 7.05 to 10.02 as the KPE increases. Additionally, Table 1 shows that the difference between low-energy and high-energy groups is statistically significant for all 6 comparisons ($p < 0.008$).

Conceptually, KPE $E$ measures the cumulative kinetic effort along a sampling trajectory. Empirically, within each fixed CFG scale, higher $E$ is associated with higher CLIP score and CLIP margin, indicating that $E$ captures sample-level semantic variation beyond guidance strength.

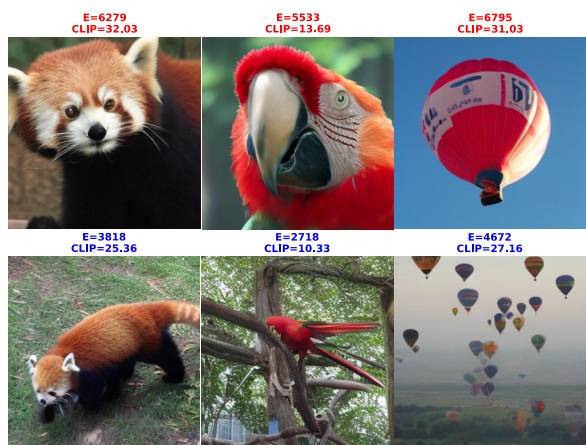

*Figure 1.* **Higher-$E$ samples show clearer semantic cues.** Paired samples from the same class on ImageNet-256 (CFG=4.0): top row corresponds to higher KPE, bottom row to lower KPE. Higher-energy samples exhibit more salient class-specific attributes.

### 4.2. KPE vs. Local Support

Finding 2: Higher $E$ consistently correlates with lower estimated local support.

**Setup and Metrics.** We define *local support* at a generated point as the concentration of training samples in a neighborhood of that point, where the neighborhood is computed either in the original data space or in a chosen feature space. We evaluate this inverse association on (i) three

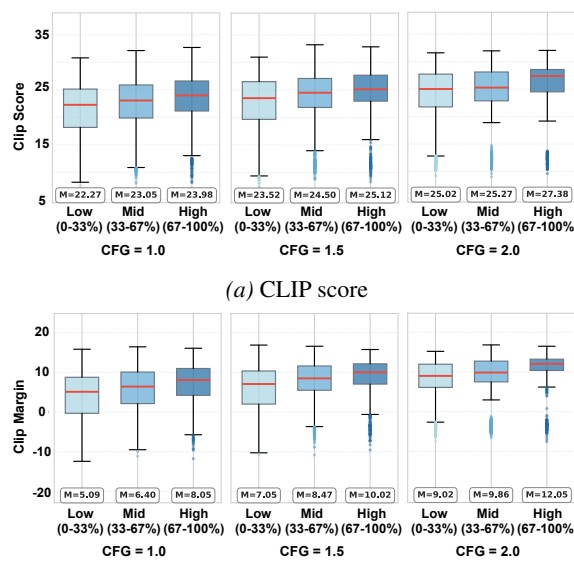

*(a)* CLIP score

*(b)* CLIP margin

*Figure 2.* **KPE correlates with semantic strength and discriminability across CFG scales.** Box plots of (a) CLIP score and (b) CLIP margin for low/mid/high KPE (0–33%, 33–67%, 67–100%) at CFG 1.0/1.5/4.0. Both metrics increase with KPE (medians labeled).

*Table 1.* **Higher KPE improves semantic strength and discriminability across CFG scales.** We compare low-energy (0–33% KPE) vs. high-energy (67–100% KPE) groups; all differences remain significant after Bonferroni correction (6 tests; $p < 0.008$).

| CFG Scale | Metric | Low Energy $\mu_{\pm\sigma}$ | High Energy $\mu_{\pm\sigma}$ | $\Delta\mu$ | Cohen's $d$ |
|---|---|---|---|---|---|
| 1.0 | CLIP Score | $21.22_{\pm5.35}$ | $23.43_{\pm4.44}$ | $+2.21$ | $0.450$ |
| | CLIP Margin | $4.15_{\pm5.99}$ | $7.09_{\pm5.00}$ | $+2.94$ | $0.534$ |
| 1.5 | CLIP Score | $21.87_{\pm5.99}$ | $24.62_{\pm4.29}$ | $+2.75$ | $0.527$ |
| | CLIP Margin | $5.66_{\pm6.17}$ | $8.93_{\pm4.54}$ | $+3.27$ | $0.603$ |
| 4.0 | CLIP Score | $23.23_{\pm5.89}$ | $25.87_{\pm4.39}$ | $+2.64$ | $0.509$ |
| | CLIP Margin | $7.44_{\pm5.95}$ | $10.82_{\pm4.40}$ | $+3.38$ | $0.646$ |

*Notes:* Values are reported as mean $\pm$ std. Cohen's $d$ is the effect size. Two-sample $t$-tests are Bonferroni-corrected ($\alpha = 0.05/6 \approx 0.008$); $n \approx 1,333$ per group.

synthetic 2D datasets with explicit density stratification (dense core + sparse ring, multiscale clusters, sandwich) and (ii) benchmark datasets: CIFAR-10 (OT-CFM (Tong et al., 2024)) and ImageNet-256 (SiT-XL/2 (Ma et al., 2024)). For (ii), we generate 2,000 samples via Euler integration (NFE $\in \{10, 50, 150\}$) and estimate local support using 22D descriptors, including RGB statistics, Gabor responses, and edge density. These descriptors are projected into a 2D space via PCA, where local support is quantified relative to the training set using $k$-NN distances with $k = 50$ and Gaussian KDE. We report Spearman's $\rho$ for monotonic correlation and Cliff's $\delta$ (top-20% vs. bottom-20% KPE samples) for effect size; see Appendix E for details.

Estimating density of natural images in pixel space is ill-posed. Accordingly, our k-NN/KDE estimates should be

*Table 2.* **KPE is negatively correlated with the estimated local support of the training set.** Spearman $\rho$ and Cliff's $\delta$ on CIFAR-10 and ImageNet-256 using $k$-NN and KDE support estimates for NFE $\in \{10, 50, 150\}$. The negative correlation strengthens with larger NFE on CIFAR-10 and remains weaker but consistently valid on ImageNet-256.

| Metric | | CIFAR-10 (NFE) 10 | 50 | 150 | ImageNet-256 (NFE) 10 | 50 | 150 |
|---|---|---|---|---|---|---|---|
| $\rho \downarrow$ | k-NN | $-0.54$ | $-0.61$ | $-0.65$ | $-0.38$ | $-0.42$ | $-0.38$ |
| $\delta \downarrow$ | | $-0.83$ | $-0.89$ | $-0.93$ | $-0.55$ | $-0.58$ | $-0.55$ |
| $\rho \downarrow$ | KDE | $-0.54$ | $-0.61$ | $-0.64$ | $-0.31$ | $-0.33$ | $-0.31$ |
| $\delta \downarrow$ | | $-0.82$ | $-0.88$ | $-0.92$ | $-0.43$ | $-0.47$ | $-0.43$ |

interpreted as *representation-dependent* proxies of local support in the descriptor/PCA space. They are useful for relative ranking and trend analysis, but should not be interpreted as calibrated estimates of data-manifold density.

**Results.** Figure 3 shows a consistent inverse relation on the 2D synthetic data: trajectories whose endpoints fall in low-density regions exhibit higher KPE (KPE vs. density strata). This is accompanied by larger/more persistent instantaneous power $\|v(t)\|^2$, leading to faster growth and higher final cumulative KPE. On CIFAR-10, Figure 4 shows (a) support and KPE surfaces that mirror each other (higher local support aligns with lower KPE), and (b) the top 10% highest-KPE samples overlaid on the support surface, concentrating in locally sparse regions.

Table 2 shows consistent evidence under both $k$-NN and KDE support estimates: on CIFAR-10, correlations strengthen with larger NFE ($\rho$: $-0.54 \rightarrow -0.61 \rightarrow -0.65$, $\delta$: $-0.83 \rightarrow -0.89 \rightarrow -0.93$ for $k$-NN; similar for KDE), while on ImageNet-256 they remain consistently negative but weaker ($\rho \approx -0.31$ to $-0.42$, $\delta \approx -0.43$ to $-0.58$). Figure 5 plots KPE against training log-support on CIFAR-10 (NFE $= 150$, $n = 2,000$): Spearman correlations are strongly negative under both $k$-NN and KDE ($\rho = -0.65/-0.64$).

Table 3 shows that the negative KPE–support relation is robust to the choice of representation. On ImageNet-256, the correlation already appears with low-dimensional handcrafted descriptors after PCA ($\rho = -0.38$), becomes substantially stronger when using the full 22D descriptor space ($\rho = -0.67$), and is strongest in VAE latent spaces ($\rho = -0.72$ for 10D PCA and $-0.74$ for 22D PCA).

### 4.3. Theoretical Analysis

To better understand the above empirical findings, we analyze neural flow matching through the lens of *empirical flow matching* (EFM) (Bertrand et al., 2025; Lim, 2025).

Let $z$ denote the latent variable (i.e., the $x(t)$ in §3) and let $\{x^{(i)}\}_{i=1}^N \subset \mathbb{R}^d$ be the training data. For $t \in [0, 1]$, the

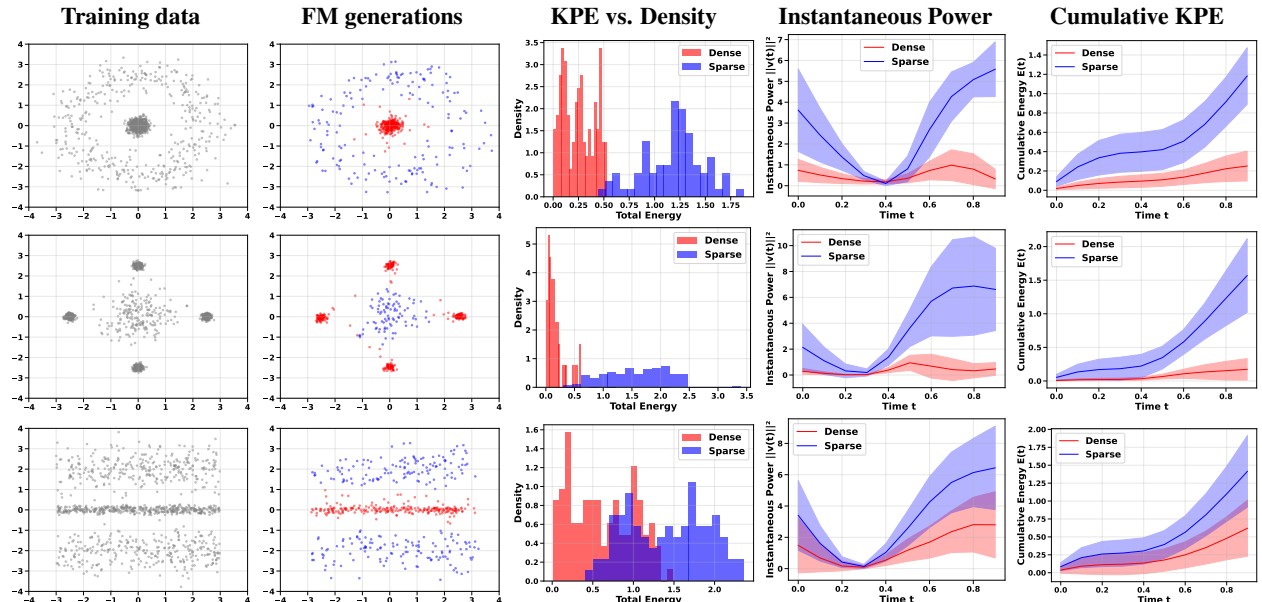

**Figure 3.** **Inverse KPE–density relation on 2D synthetic datasets.** Each row corresponds to one distribution (`dense_sparse`, `multiscale_clusters`, `sandwich`). Columns (left→right): training data distribution, FM generations, KPE vs. density strata, instantaneous power $\|v(t)\|^2$ over time, cumulative KPE. Across datasets, trajectories ending in low-density regions accumulate higher KPE (Mann-Whitney U (MWU) test $p < 10^{-3}$); details in Appendix F.1.

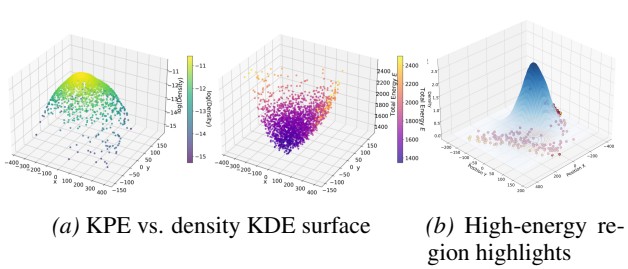

*(a)* KPE vs. density KDE surface      *(b)* High-energy region highlights

**Figure 4.** **High-KPE samples lie in locally sparse regions.** (a) On CIFAR-10 at $\mathrm{NFE} = 150$, the log(support) surface (left) is anti-aligned with KPE (right): higher local support corresponds to lower energy. (b) The top 10% KPE samples (overlaid) cluster in locally sparse areas, consistent with Theorem 4.2.

*Table 3.* **The negative KPE–support correlation is robust across feature spaces.** ImageNet-256, $\mathrm{NFE} = 10$, CFG= 1.5, $n = 4{,}000$ samples, $k$-NN with $k = 50$. All four feature spaces yield negative Pearson $r$ and Spearman $\rho$, confirming that the relation is not specific to one descriptor.

| Feature space | Dim | Pearson $r$ | Spearman $\rho$ |
|---|---|---|---|
| Descriptors+PCA | $22 \rightarrow 2$ | $-0.34$ | $-0.38$ |
| Descriptors | $22$ | $-0.66$ | $-0.67$ |
| VAE + PCA | $4096 \rightarrow 10$ | $-0.69$ | $-0.72$ |
| VAE + PCA | $4096 \rightarrow 22$ | $-0.72$ | $-0.74$ |

conditional probability path with a general schedule $\gamma(t)$ is:

$$p_t(z \mid x^{(i)}) = \mathcal{N}\big(z; \gamma(t)x^{(i)}, (1 - \gamma(t))^2 I_d\big), \quad (6)$$

where $\gamma : [0, 1] \rightarrow [0, 1]$ is differentiable, $\gamma(0) = 0$, and

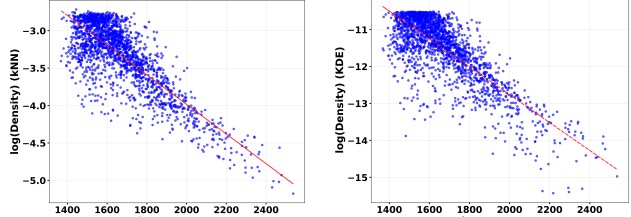

**Figure 5.** **Strong negative correlation between KPE and local training support.** Scatter plot of KPE versus training log-support on CIFAR-10 (NFE $= 150$, $n = 2{,}000$ samples). Left: $k$-NN; right: KDE. Each point represents one generated sample; red line shows linear regression fit. Spearman correlations are strongly negative (k-NN: $\rho = -0.65$; KDE: $\rho = -0.64$), indicating a strong monotonic inverse relationship.

$\gamma(1) = 1$. Define the mixture density and responsibilities

$$\hat{p}_t(z) = \frac{1}{N} \sum_{i=1}^{N} p_t(z \mid x^{(i)}), \quad \lambda_i(z, t) := \frac{p_t(z|x^{(i)})}{\sum_{j=1}^{N} p_t(z|x^{(j)})}$$

with $\lambda_i(z, t)$ indicating the contribution of component $i$ at $(z, t)$. Let $\hat{u}^*(z, t)$ be the optimal velocity under the EFM regression objective; when $\gamma(t) = t$, it reduces to the closed-form expression in §5. Denote $\sigma_t^2 := (1 - \gamma(t))^2$.

**Lemma 4.1** (Score-Based Decomposition). *The optimal velocity field minimizing the EFM regression objective admits the closed-from representation* (proof in Appendix B.3)*:*

$$\hat{u}^*(z, t) = \alpha(t) \, \nabla_z \log \hat{p}_t(z) + \beta(t) \, z, \quad (7)$$

*where* $\alpha(t) = \frac{\dot{\gamma}(t)\sigma_t^2}{\gamma(t)(1-\gamma(t))}$ *and* $\beta(t) = \frac{\dot{\gamma}(t)}{\gamma(t)}$.

**Theorem 4.2** (Energy-Density Relation)**.** *Under the* poste-*rior dominance regime, at each* $(z, t)$ *there exists a dominant component* $i^*$ *such that* $\lambda_{i*}(z, t) \geq 1 - \varepsilon$ *for some* $\varepsilon \in (0, 1/2)$*, the instantaneous kinetic energy is affinely bounded by the negative log-density* (proof in Appendix B.4.)*:*

$$c_1(t)\big(-\log \hat{p}_t(z)\big) - C_t' \leq \|\hat{u}^*(z, t)\|^2$$
$$\leq c_2(t)\big(-\log \hat{p}_t(z)\big) + C_t'. \tag{8}$$

*where the constants satisfy* $c_1(t), c_2(t) = \Theta(m(t)^2 \sigma_t^2)$ *with* $m(t) = -\dot{\gamma}(t)/(1 - \gamma(t))$*, and* $C_t' \in \mathbb{R}$ *depends on* $\log N$, $-\log(1 - \varepsilon)$*, and geometric properties of the dominant component.*

*Remark* 4.3 (Explicit Constants and Integrated Form)*.* The constants in Theorem 4.2 can be chosen explicitly as $c_1(t) = \frac{1}{2} m(t)^2 \sigma_t^2$ and $c_2(t) = 12\, m(t)^2 \sigma_t^2$ (see Theorem B.1).

Integrating (8) along a trajectory $z_{0\to 1} = \{z(t)\}_{t=0}^1$ yields:

$$E(z_{0\to 1}) = \Theta\left(\int_0^1 \big(-\log \hat{p}_t(z(t))\big)\, dt\right) + O(1). \tag{9}$$

This formalizes the inverse relationship: trajectories traversing low-density regions accumulate higher kinetic energy.

# 5. When Energy Backfires: Extreme Kinetic Energy Drives Memorization

Having shown that higher KPE correlates with higher sample quality, we ask: what happens if KPE is pushed to the extreme? Using EFM's closed-form expression for the optimal velocity field, we uncover a paradox: although EFM reaches much higher peak power than neural fields, it produces near-exact copies of training data. *Extreme energy drives memorization, not better generation.*

## 5.1. Closed-Form Formula

Let $\hat{p}_{\text{data}} = \frac{1}{N} \sum_{i=1}^N \delta_{x^{(i)}}$ denote the empirical data distribution. Under conditional flow matching with Gaussian bridges $p_t(x \mid x^{(i)}) = \mathcal{N}(x; tx^{(i)}, (1-t)^2 I_d)$, the optimal velocity field that minimizes the empirical regression objective admits the following closed-form expression (Bertrand et al., 2025) (see derivation in Appendix C):

$$\hat{u}^*(x, t) = \sum_{i=1}^N \lambda_i(x, t) \frac{x^{(i)} - x}{1 - t},$$
$$\lambda_i(x, t) = \frac{\exp\left(-\frac{\|x - tx^{(i)}\|^2}{2(1-t)^2}\right)}{\sum_{j=1}^N \exp\left(-\frac{\|x - tx^{(j)}\|^2}{2(1-t)^2}\right)}. \tag{10}$$

## 5.2. Why EFM Leads to Extreme Energy

The $1/(1-t)$ factor in Eq. (10) can create large terminal-time velocities. In particular, if a solution trajectory remains

a fixed distance away from the training set on a terminal interval and the softmax weights $\lambda_i(x(t), t)$ concentrate on a unique atom, then $\|\hat{u}^*(x(t), t)\| \gtrsim (1-t)^{-1}$ and the terminal contribution to $\text{KPE}[x] = \frac{1}{2} \int_0^1 \|\dot{x}(t)\|^2\, dt$ diverges.

**Lemma 5.1** (Informal: Terminal energy blow-up)**.** *Consider any trajectory segment* $t \in [1 - \varepsilon, 1)$ *on which there is a constant* $c > 0$ *such that* $\|x(t) - x^{(i)}\| \geq c$ *for all training samples* $\{x^{(i)}\}_{i=1}^N$ *and the terminal posterior concentrates on a unique training atom along the segment. Then*

$$\int_{1-\varepsilon}^1 \|\hat{u}^*(x(t), t)\|^2\, dt = +\infty. \tag{11}$$

**Proposition 5.2** (Informal: Extreme kinetic energy)**.** *The empirical closed-form velocity field can exhibit terminal-time kinetic energy blow-up.*

(a) **Unbounded terminal energy***: For trajectories that remain separated from all training points as* $t \to 1$ *and along which the terminal posterior* $\lambda_i(x(t), t)$ *concentrates on a unique training atom, the terminal kinetic energy diverges:* $\int_{1-\epsilon}^1 \|\hat{u}^*(x(t), t)\|^2 dt = +\infty$*.*

(b) **Minimum terminal cost to close a late-time gap***: Any absolutely continuous trajectory with* $x(1) = x^{(i)}$ *satisfies* $\int_t^1 \|\dot{x}(s)\|^2 ds \geq \|x^{(i)} - x(t)\|_2^2 / (1-t)$*.*

See Appendix D for detailed statements and proofs. Together, these results show that the $1/(1-t)$ factor creates a potential terminal singularity. Unless the gap to the selected training atom closes sufficiently fast, the terminal kinetic cost can diverge. Since exact matching of the empirical terminal distribution requires trajectories to terminate at discrete training atoms, EFM imposes a sharp terminal-closure requirement: samples must approach their selected atoms rapidly enough as $t \to 1$. If they remain too far away near the terminal time, the $1/(1-t)$ scaling produces a terminal energy spike, which is associated with memorization.

## 5.3. Experimental Validation

We compare *EFM* with *Vanilla FM* (the baseline where the velocity field is learned by a neural network) using the same ODE solver (the midpoint scheme with NFE = 100). EFM uses Eq. (10) with 100 nearest neighbors.

### 5.3.1. SYNTHETIC 2D DATASETS

We test the samplers on three density-stratified datasets (dense_sparse, multiscale_clusters, sandwich). Figure 6 plots cumulative energy (left) and instantaneous power $\|v(t)\|^2$ (right). The power curves reveal spikes emerging at $t > 0.50$–$0.70$, which is consistent with the prediction of Lemma 5.1. EFM achieves $1.3\times$–$3.9\times$ higher peak power than Vanilla FM (which exhibits no spikes), with corresponding KPE increases of 30%–50% across datasets.

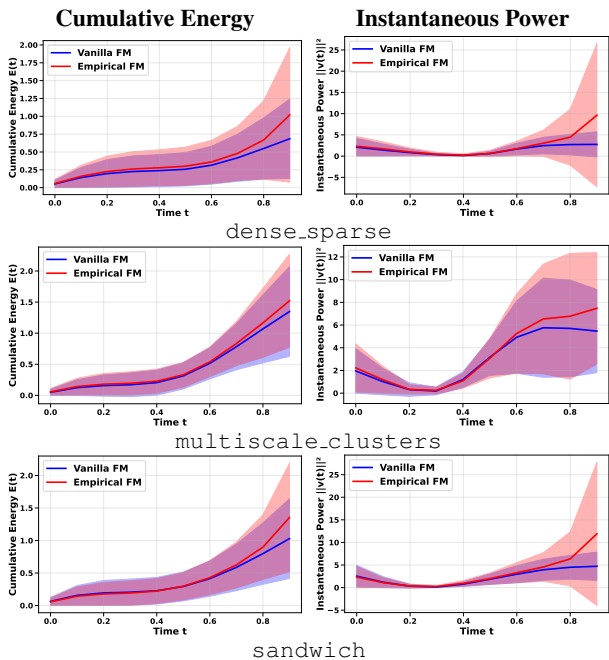

**Figure 6.** **Empirical FM shows terminal-time power spikes despite optimal regression loss.** Across three synthetic datasets, Empirical FM develops late-time spikes at $t > 0.50$–$0.70$ and reaches **1.3×–3.9×** higher peak power than Vanilla FM (Lemma 5.1). Left: cumulative kinetic energy $\frac{1}{2}\int_0^t \|v(\tau)\|^2 d\tau$; Right: instantaneous power $\|v(t)\|^2$. More visualizations are provided in Appendix J.

### 5.3.2. CELEBA-HQ: KPE ACROSS TRAINING

We track CelebA 64×64 training-time behavior as a U-Net FM model approaches the closed-form optimum, measuring FID, $F_{\text{mem}}$, and average KPE across checkpoints. We train on a subset of 1,024 images, following (Bonnaire et al., 2025). As shown in Figure 7a, FID improves early (decreasing from $\sim 280$ to $\sim 15$) and then plateaus after $10^4$ iterations, while both KPE and $F_{\text{mem}}$ continue to rise. Notably, $F_{\text{mem}}$ increases sharply after $10^4$ iterations and reaches 98% by 2M iterations, and KPE keeps increasing to 540. This suggests that, in the late-training regime, higher kinetic energy correlates with memorization rather than further quality gains. Figure 7b provides qualitative evidence: each panel pairs generated samples (left) with their nearest training neighbors (right). Early checkpoints show diverse generations with noticeable gaps to neighbors, whereas late checkpoints produce near-copies whose fine details closely match specific training images, consistent with rising $F_{\text{mem}}$. More visualizations are provided in Appendix J.

## 6. Kinetic Trajectory Shaping

*How can we exploit the positive KPE-semantic correlation (§4) while suppressing terminal energy spikes?* We propose *Kinetic Trajectory Shaping (KTS)*, a training-free two-phase energy modulation method.

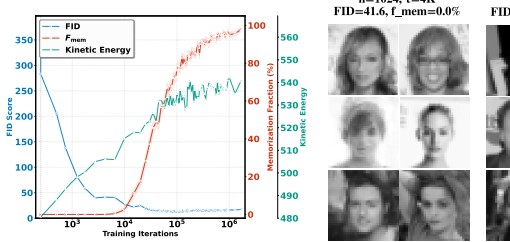

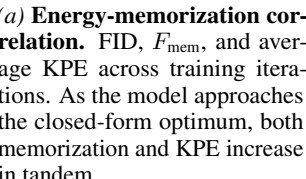 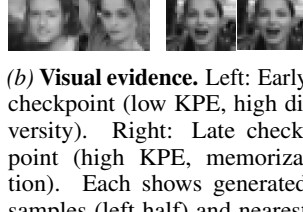

*(a)* **Energy-memorization correlation.** FID, $F_{\text{mem}}$, and average KPE across training iterations. As the model approaches the closed-form optimum, both memorization and KPE increase in tandem.

*(b)* **Visual evidence.** Left: Early checkpoint (low KPE, high diversity). Right: Late checkpoint (high KPE, memorization). Each shows generated samples (left half) and nearest training neighbors (right half).

*Figure 7.* **Energy increases lead to memorization on CelebA.** (a) KPE and $F_{\text{mem}}$ rise throughout training, whereas FID plateaus late. (b) Nearest-neighbor pairs show diverse samples early but near-copies at late checkpoints.

### 6.1. A Two-Phase Strategy for Reducing Memorization

**The core insight:** *KPE allocation matters.* Within the normal sampling regime, higher KPE correlates with stronger semantic fidelity (§4). However, in the extreme empirical-matching regime, KPE can blow up near the terminal time due to the $1/(1 - t)$ scaling, and this terminal concentration is associated with memorization rather than additional quality gains (§5). Thus, we *boost early* and *damp late*.

KTS rescales the velocity via a time-dependent gain $\eta(t)$:

$$\tilde{v}(x_t, t) = \eta(t) \cdot v_\theta(x_t, t), \tag{12}$$

with phase-specific modulation:

$$\eta(t) = \begin{cases} 1 + \alpha(t), & t < \tau_{\text{split}} & \text{(Kinetic Launch)} \\ 1 - \beta(t), & t \geq \tau_{\text{split}} & \text{(Kinetic Soft Landing)} \end{cases} \tag{13}$$

**Kinetic Launch** ($\alpha > 0$) boosts early velocity to increase KPE during the semantic formation phase. **Kinetic Soft Landing** ($\beta > 0$) damps late velocity to mitigate the $1/(1 - t)$ divergence, reducing terminal singular behavior and memorization.

**Launch:** we use a linear decay for a strong initial impulse and smooth transition:

$$\alpha(t) = \alpha_0 \cdot \max\left(0, 1 - \frac{t}{\tau_{\text{split}}}\right), \quad t \in [0, \tau_{\text{split}}]. \tag{14}$$

**Soft Landing:** we use exponential damping to directly suppress terminal singularities:

$$\beta(t) = \beta_0 \cdot [\exp\left(k \cdot (t - \tau_{\text{split}})\right) - 1], \quad t \in [\tau_{\text{split}}, 1]. \tag{15}$$

We set $\tau_{\text{split}} = 0.6$, aligned with the onset of the spike in Figure 6, and use $k = 3$ as the default. Algorithm 1

---

**Algorithm 1** Kinetic Trajectory Shaping (KTS)

1: **Input:** $v_\theta, x_0 \sim \mathcal{N}(0, I), \alpha_0, \beta_0, k, \tau_{\text{split}}, \Delta t$
2: **Output:** $x_1$
3: **for** $t = 0$ to $1$ with step $\Delta t$ **do**
4:    $v_t \leftarrow v_\theta(x_t, t)$                 // base velocity
5:    **if** $t < \tau_{\text{split}}$ **then**
6:       $\eta(t) \leftarrow 1 + \alpha_0 \cdot (1 - t/\tau_{\text{split}})$   // launch
7:    **else**
8:       $\eta(t) \leftarrow 1 - \beta_0 \cdot (\exp(k(t - \tau_{\text{split}})) - 1)$ // soft landing
9:    **end if**
10:   $x_{t+\Delta t} \leftarrow x_t + \eta(t)\, v_t\, \Delta t$         // Euler step
11: **end for**
12: **return** $x_1$

---

outlines the implementation. Under Euler integration, KTS acts as a time-dependent effective step-size schedule. Since $v_\theta(x, t)$ is explicitly time-conditioned, it is not merely a time reparameterization: it rescales the velocity magnitude while preserving the model time input $t$.

## 6.2. Experiments

**Setup.** We validate KTS on CelebA at $32 \times 32$ resolution (1024 grayscale training images similar to (Bonnaire et al., 2025)), and evaluate generation quality and memorization under a controlled setup. We train conditional flow matching models (Lipman et al., 2022) with a U-Net (32 base channels; 3 resolution levels with attention at higher resolutions) using Adam ($10^{-4}$), batch size 512, for $2 \times 10^6$ iterations, and sample with Euler ODE solver (NFE = 100). We compare baseline FM and KTS (with $\tau = 0.6$) across hyperparameters, and report FID (held-out reference statistics) as well as the memorization fraction $F_{\text{mem}}$ defined below.

**Memorization metric.** Following (Yoon et al., 2023; Bonnaire et al., 2025), let $\mathbf{a}^{\mu_1}(\mathbf{x}), \mathbf{a}^{\mu_2}(\mathbf{x})$ denote the nearest and second-nearest training neighbors of $\mathbf{x} \in [0, 1]^d$ in $\ell_2$ pixel space, and $r_{\text{gap}}(\mathbf{x}) = \|\mathbf{x} - \mathbf{a}^{\mu_1}\|_2 / \|\mathbf{x} - \mathbf{a}^{\mu_2}\|_2$. We declare $\mathbf{x}$ memorized when $r_{\text{gap}}(\mathbf{x}) < \tau_{\text{gap}}$ and report

$$F_{\text{mem}} = \frac{1}{n} \sum_{i=1}^{n} \mathbf{1}\left[ r_{\text{gap}}(\mathbf{x}^{(i)}) < \tau_{\text{gap}} \right], \qquad (16)$$

with $\tau_{\text{gap}} = 1/3$ and $n = 10{,}000$.

**Main results.** Table 4 reports CelebA results at 30K-step checkpoint. Compared with FM (FID@10k 16.68, $F_{\text{mem}}$ 37.34%), KTS offers a tunable trade-off via $\alpha_0$ and $\beta_0$. Increasing $\beta_0$ reduces memorization (lowest $F_{\text{mem}}$ **19.36%** at $\alpha_0 = 0, \beta_0 = 0.02$) but at the cost of worse sample quality (FID@10k 86.56), whereas increasing $\alpha_0$ improves quality (best FID@10k **11.27** at $\alpha_0 = 0.02, \beta_0 = 0$) with little change in $F_{\text{mem}}$. Importantly, enabling both terms ($\alpha_0 = \beta_0 = 0.01$) yields a sweet spot, improving over FM in both quality and memorization (FID@10k 14.35, $F_{\text{mem}}$ 31.22%).

Table 5 summarizes results on ImageNet-256. KTS matches

or improves FM, while offering a tunable precision–recall trade-off. Increasing $\alpha_0$ improves quality and alignment: $\alpha_0 = 0.05$ gives the best FID (**11.59**), CLIP (**24.34**), and precision (**0.731**), but lowers recall (0.630). Increasing $\beta_0$ improves coverage: $\beta_0 = 0.05$ achieves the highest recall (**0.657**) with slightly worse FID (12.45) and precision (0.721). Enabling both ($\alpha_0 = \beta_0 = 0.01$) yields a balanced point (FID 11.63, CLIP 24.20), consistent with KPE trends: $\alpha_0$ increases $\text{KPE}_{\text{early}}$ and $\beta_0$ reduces $\text{KPE}_{\text{late}}$.

*Table 4.* **Performance comparison on CelebA dataset** at the 30K-step checkpoint. We compare KTS with varying hyperparameters against the FM baseline. **Bold** indicates the best result.

| Method | Hyperparams | | Metrics | | |
|---|---|---|---|---|---|
| | $\alpha_0$ | $\beta_0$ | FID@10k ↓ | $F_{\text{mem}}$ ↓ (%) | $\text{KPE}_{\text{early}}$ / $\text{KPE}_{\text{late}}$ |
| FM (Baseline) | 0.00 | 0.00 | 16.68 | 37.34 | 310.6 / 212.8 |
| KTS (Ours) | 0.00 | 0.01 | 35.04 | 30.17 | 311.7 / 202.1 |
| | 0.00 | 0.02 | 86.56 | **19.36** | 311.7 / 188.9 |
| | 0.01 | 0.00 | **11.30** | 37.44 | 312.8 / 211.1 |
| | 0.02 | 0.00 | **11.27** | 36.78 | 315.8 / 211.4 |
| | 0.01 | 0.01 | 14.35 | 31.22 | 313.7 / 201.3 |

*Table 5.* **Performance comparison on ImageNet-256 dataset.** We compare KTS with varying hyperparameters against the FM baseline. **Bold** indicates the best result.

| Method | Hyperparams | | Metrics | | | | |
|---|---|---|---|---|---|---|---|
| | $\alpha_0$ | $\beta_0$ | FID@10K↓ | CLIP↑ | Prec.↑ | Rec.↑ | $\text{KPE}_{\text{early}}$ / $\text{KPE}_{\text{late}}$ |
| FM (Baseline) | 0.00 | 0.00 | 11.70 | 24.11 | 0.728 | 0.655 | 1081.0 / 470.0 |
| KTS (Ours) | 0.00 | 0.01 | 11.84 | 24.10 | 0.728 | 0.653 | 1081.0 / 464.4 |
| | 0.00 | 0.05 | 12.45 | 24.05 | 0.721 | **0.657** | 1081.0 / 442.3 |
| | 0.01 | 0.00 | 11.61 | 24.16 | 0.730 | 0.648 | 1094.5 / 470.0 |
| | 0.05 | 0.00 | **11.59** | **24.34** | **0.731** | 0.630 | 1149.8 / 470.0 |
| | 0.01 | 0.01 | 11.63 | 24.20 | 0.729 | 0.653 | 1094.5 / 464.4 |

**Choice of $\tau_{\text{split}}$.** Table 6 sweeps the phase boundary under fixed $\alpha_0 = \beta_0 = 0.01$. Early splits ($\tau_{\text{split}} = 0.2, 0.4$) suppress memorization more strongly but increase FID, indicating premature damping of semantic formation. A late split ($\tau_{\text{split}} = 0.8$) leaves more late-time dynamics undamped and raises $F_{\text{mem}}$. We therefore use $\tau_{\text{split}} = 0.6$, which lies in the observed spike-onset interval $[0.50, 0.70]$ (Figure 6) and results in the best FID among the tested values.

**Schedule functional form.** Table 7 varies the early launch and late soft-landing functions while fixing $\tau_{\text{split}} = 0.6$ and $\alpha_0 = \beta_0 = 0.01$. Across all tested linear, constant, and exponential variants, the two-phase boost-then-damp design improves FID over FM (16.68), achieving values between 14.35 and 11.72. This robustness suggests that the benefit

*Table 6.* $\tau_{\text{split}}$ **sensitivity on CelebA** $32 \times 32$. Fixed $\alpha_0 = \beta_0 = 0.01$, Euler, NFE = 100, uniform schedule. Among the tested phase boundaries, $\tau_{\text{split}} = 0.6$ achieves the best FID.

| $\tau_{\text{split}}$ | FID@10k ↓ | $F_{\text{mem}}$ (%) ↓ |
|---|---|---|
| 0.2 | 60.31 | 23.66 |
| 0.4 | 48.58 | 27.15 |
| **0.6** | **14.35** | **31.22** |
| 0.8 | 21.07 | 34.30 |

comes from the phase structure: increasing early motion supports sample formation, while reducing late motion controls terminal behavior. The exact functional form affects the magnitude of the gains, but improvements persist across all tested choices. In this sweep, linear–exponential gives the largest memorization reduction (37.34% to 31.22%).

*Table 7.* **Functional form of launch/soft-landing schedules (CelebA** $32\times32$**).** $\tau_{\text{split}}$=0.6, $\alpha_0$=$\beta_0$=0.01, Euler, NFE=100. All the two-phase combinations improve over the FM baseline; [†] denotes the schedule used throughout the paper.

| Early $\alpha(t)$ | Late $\beta(t)$ | FID@10k↓ | $F_{\text{mem}}$ (%)↓ |
|---|---|---|---|
| *Baseline (no KTS)* | | 16.68 | 37.34 |
| linear[†] | exponential[†] | 14.35 | **31.22** |
| linear | constant | 12.47 | 35.50 |
| linear | linear | 11.91 | 36.73 |
| constant | linear | **11.72** | 37.15 |
| exponential | linear | 11.93 | 36.76 |

**Robustness across solvers, NFE, and schedules.** Table 8 applies the same balanced KTS configuration ($\alpha_0$=$\beta_0$=0.01, $\tau_{\text{split}}$=0.6, no per-configuration retuning) to Euler/Midpoint solvers, NFE $\in \{100, 250\}$, and uniform/cosine time schedules. KTS reduces $F_{\text{mem}}$ by 6–10 percentage points in *every* configuration while keeping FID comparable or better, indicating that the gains transfer without per-setup retuning.

*Table 8.* **Robustness across solvers, NFE, and schedules (CelebA** $32\times32$**).** Balanced KTS uses $\alpha_0$=$\beta_0$=0.01, $\tau_{\text{split}}$=0.6 (no per-configuration retuning). KTS lowers $F_{\text{mem}}$ by 6–10 pp in *every* configuration; FID is comparable or better. Best per row in **bold**.

| Solver | NFE | Sched. | FID@10k↓ | | $F_{\text{mem}}$ (%)↓ | |
|---|---|---|---|---|---|---|
| | | | FM | KTS | FM | KTS |
| Euler | 100 | unif. | 16.68 | **14.35** | 37.34 | **31.22** |
| Euler | 250 | unif. | 13.11 | **12.87** | 37.83 | **30.23** |
| Midpoint | 100 | unif. | 19.20 | **18.88** | 37.86 | **30.19** |
| Midpoint | 250 | unif. | 15.80 | **15.00** | 37.68 | **29.72** |
| Euler | 100 | cos. | 15.21 | **15.20** | 38.19 | **28.54** |
| Midpoint | 100 | cos. | 15.42 | **15.36** | 37.80 | **30.84** |

**Ablation study on $\alpha_0$ over training iterations.** Figure 8 shows the training-time behavior of $\Delta$FID and $\Delta F_{\text{mem}}$ for different $\alpha_0$ values (with $\beta_0$=0). We report *per-iteration* differences relative to the FM baseline at the same iteration (baseline is 0). Negative $\Delta$FID indicates better sample quality (a 27.6–28.3% reduction vs. FM), and negative $\Delta F_{\text{mem}}$ indicates reduced memorization. Increasing $\alpha_0$ improves FID early and consistently, with gains that gradually saturate, while $\Delta F_{\text{mem}}$ stays near zero after the early phase (small early fluctuations likely reflect estimator noise). Overall, $\alpha_0$ mainly accelerates quality acquisition and $\beta_0$ is needed for stronger memorization suppression.

**Ablation study on $\beta_0$ over training iterations.** Figure 9 shows the training-time behavior of $\Delta$FID and $\Delta F_{\text{mem}}$ for

different $\beta_0$ ($\alpha_0$=0). We report per-iteration differences relative to the FM baseline at the same iteration (baseline is 0). Increasing $\beta_0$ yields a clear and sustained reduction in memorization, with the gap widening in the mid-to-late regime (more negative $\Delta F_{\text{mem}}$). However, too large $\beta_0$ can over-damp the dynamics and hurt quality: $\beta_0$=0.02 reduces memorization the most but makes $\Delta$FID positive, while $\beta_0$=0.01 keeps both $\Delta$FID and $\Delta F_{\text{mem}}$ negative, improving quality and memorization simultaneously.

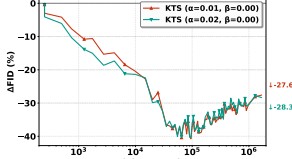 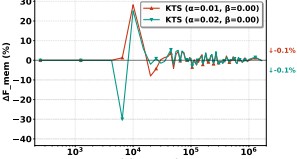

*(a)* FID improvement with $\alpha$    *(b)* Memorization risk with $\alpha$

*Figure 8.* **Ablation on $\alpha := \alpha_0$ over training iterations.** We plot the relative changes $\Delta$FID and $\Delta F_{\text{mem}}$ with respect to the FM baseline evaluated at the same training iteration (baseline is 0). Negative $\Delta$FID indicates improved sample quality, while positive $\Delta F_{\text{mem}}$ indicates increased memorization.

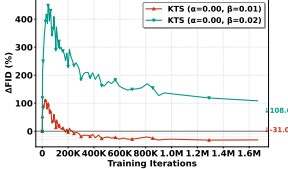 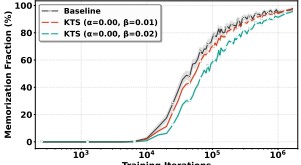

*(a)* FID change with $\beta$    *(b)* $F_{\text{mem}}$ reduction with $\beta$

*Figure 9.* **Ablation on $\beta := \beta_0$ over training iterations.** We plot the relative changes $\Delta$FID and $\Delta F_{\text{mem}}$ with respect to the FM baseline evaluated at the same training iteration (baseline is 0). While excessively high $\beta_0$ can hurt sample quality (positive $\Delta$FID), increasing $\beta_0$ generally reduces memorization (negative $\Delta F_{\text{mem}}$), revealing a tunable quality–memorization trade-off.

# 7. Conclusions and Limitations

We introduce *Kinetic Path Energy (KPE)*, an action-like per-sample metric, to quantify the accumulated *kinetic effort* along the sampling trajectory. Empirically, higher KPE is associated with samples of higher semantic fidelity and with trajectories that end in locally sparser regions of representation space, but the relationship is *not monotone*: pushing energy to extremes can instead induce memorization. Our analysis reveals a structural terminal-time singular component that leads to late-time energy spikes and collapses trajectories toward near-copies of training examples. This *Goldilocks principle* motivates *Kinetic Trajectory Shaping (KTS)*, a lightweight, training-free inference-time procedure that redistributes energy over time to improve quality while mitigating memorization. Our study focuses on ODE-based flows and specific theoretical regimes; extending energy-based diagnostics and controls to other models and stochastic samplers is an important direction for future work.

## Impact Statement

This paper presents work whose goal is to advance the field of Machine Learning, and more specifically, the theoretical understanding of implicit regularization as a tool for structured recovery problems. There are many potential societal consequences of our work, none which we feel must be specifically highlighted here.

## Acknowledgements

This work was supported by the Wallenberg AI, Autonomous Systems and Software Program (WASP) funded by the Knut and Alice Wallenberg Foundation. The computations were enabled by resources provided by the National Academic Infrastructure for Supercomputing in Sweden (NAISS), partially funded by the Swedish Research Council through grant agreement no. 2022-06725. SHL would like to acknowledge support from the Wallenberg Initiative on Networks and Quantum Information (WINQ) and the Swedish Research Council (VR/2021-03648). XL is supported by a Trinity College Dublin PhD Award and the Research Ireland Centre for Research Training in Artificial Intelligence (Grant No. 18/CRT/6223). FG is supported in part by the National Natural Science Foundation of China under Grant 62571395. We thank the anonymous reviewers for their constructive feedback.

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

# Appendix

## A. Related Work

In this section, we discuss related work and position our paper relative to them.

**Flow Matching and Kinetic Views of Generative Transport.**    Flow matching formulates generative modeling as learning a time-dependent velocity field whose induced ODE deterministically transports samples from a base distribution to the data distribution (Lipman et al., 2022; Liu et al., 2022; Lipman et al., 2024; Wald & Steidl, 2025). This trajectory-based formulation places flow matching within the broader perspective of *dynamical measure transport*, where probability measures evolve under continuity equations driven by velocity fields. In optimal transport, the classical Benamou–Brenier formulation characterizes feasible transport paths as solutions of a dynamic optimization problem that minimizes an integrated kinetic energy subject to mass conservation (Benamou & Brenier, 2000; Villani et al., 2008). While flow matching does not explicitly enforce optimality with respect to such action functionals (Hertrich et al., 2025), it induces a learned transport dynamics that maps a base distribution to the data distribution through continuous-time particle motion. Error bounds (Zhou & Liu) and statistical guarantees (Kunkel, 2025; Mena et al., 2025) for flow matching have also been studied recently.

This connection highlights both the relevance and the limitation of existing energy-based analyses. Recent work revisits action and kinetic principles in the context of generative modeling by studying kinetically optimal or constrained probability paths (Shaul et al., 2023), and related perspectives from nonequilibrium thermodynamics and path-integral formulations interpret diffusion and transport dynamics through energetic and variational lenses (Seifert, 2012; Hirono et al., 2024; Ikeda et al., 2025). However, these approaches primarily characterize *distribution-level* or *optimal* energetic properties. In contrast, trained flow matching models realize specific, sample-dependent transport trajectories whose kinetic behavior is neither optimized nor directly constrained. This gap motivates our focus on analyzing the *realized kinetic effort* accumulated along individual generation trajectories.

**Memorization and Generalization in Flow-Based Generative Models.**    A growing body of work investigates memorization and generalization phenomena in modern generative models, including both diffusion-based and flow-based approaches. Several studies analyze memorization through the lens of implicit regularization, dynamical stability, and generalization theory, characterizing when generative models interpolate or collapse to training data (Baptista et al., 2025; Bonnaire et al., 2025; Ye et al., 2025; Yoon et al., 2023). Complementary work emphasizes the role of data geometry, score structure, or closed-form dynamics in shaping model behavior, revealing how learned generative dynamics interact with low-dimensional manifolds and discrete datasets (Pidstrigach, 2022; Gao & Li, 2024; Bertrand et al., 2025; Wan et al.).

While these analyses provide important statistical and structural insights, they largely abstract away the kinematic behavior of the sampling process itself. In particular, they do not directly characterize how individual sampling trajectories evolve in time, nor how energetic properties of these trajectories contribute to memorization. Our work complements this literature by identifying a trajectory-level mechanism specific to empirical flow matching (Lim, 2025). The terminal-time singularity of the optimal velocity field leads to large kinetic energy near the end of sampling, forcing trajectories to collapse onto training samples and inducing memorization, even in the absence of model approximation error. This perspective reframes memorization as a dynamical consequence of how kinetic effort is allocated along the generation path.

**Trajectory-Level Diagnostics and Inference-Time Control.**    Beyond training objectives, several approaches aim to improve generation quality or controllability by modifying inference dynamics without retraining. Examples include classifier-free guidance and energy-guided conditional generation, which bias sampling by scaling scores, adjusting likelihood contributions, or introducing auxiliary energy terms (Ho & Salimans, 2022; Yu et al., 2023; Xu et al., 2024; Du et al., 2024). Such methods provide powerful controls, but they typically operate through local modifications of scores or endpoint objectives, offering limited visibility into the global dynamics of the sampling trajectory.

In contrast, our approach is explicitly trajectory-centric. We introduce *Kinetic Path Energy* as a path-level diagnostic that quantifies the kinetic effort accumulated along each individual generation trajectory. This diagnostic reveals systematic relationships between energy, semantic fidelity, and manifold rarity that are not captured by endpoint-only metrics. Building on these insights, we propose *Kinetic Trajectory Shaping*, a simple yet effective training-free inference strategy that directly modulates the velocity field over time, redistributing kinetic effort across different phases of sampling. By encouraging exploration early and suppressing terminal energy blow-up late, this framework enables phase-specific control of generation dynamics and provides a unified mechanistic view of sample quality, rarity, and memorization in flow-based models.

# B. KPE and Data Density

In this section, we establish the theoretical basis for the KPE-density relationship observed empirically in §4. The main result, Theorem B.1, quantifies how instantaneous energy $\|\hat{u}^*(z, t)\|^2$ relates to negative log-density $-\log \hat{p}_t(z)$ under posterior dominance.

We develop the framework for a differentiable schedule $\gamma : [0, 1] \to [0, 1]$ and focus on the linear case $\gamma(t) = t$. The results are applied at nondegenerate times $t \in (0, 1)$ with $\gamma(t) \in (0, 1)$ and $\dot{\gamma}(t) \neq 0$.

## B.1. Preliminaries and Main Result

### B.1.1. NOTATION

We collect here all notation used throughout this appendix. For clarity, we present the key symbols in tabular form:

| Symbol | Definition |
|---|---|
| $I_d$ | $d$-dimensional identity matrix |
| $\{x^{(i)}\}_{i=1}^N$ | Training data points in $\mathbb{R}^d$ |
| $\gamma(t)$ | Interpolation schedule, $\gamma : [0, 1] \to [0, 1]$ with $\gamma(0) = 0, \gamma(1) = 1$ |
| $\dot{\gamma}(t)$ | Time derivative of $\gamma(t)$ |
| $x_0$ | Source random variable, $x_0 \sim \mathcal{N}(0, I_d)$ |
| $z_t$ | Conditional bridge: $z_t = (1 - \gamma(t))x_0 + \gamma(t)x^{(i)}$ |
| $p_t(z \mid x^{(i)})$ | Conditional probability distribution at time $t$: $p_t(z \mid x^{(i)}) = \mathcal{N}(z; \mu_i(t), \sigma_t^2 I_d)$ |
| $\mu_i(t)$ | Mean of $i$-th component: $\mu_i(t) = \gamma(t)x^{(i)}$ |
| $\sigma_t^2$ | Variance: $\sigma_t^2 = (1 - \gamma(t))^2$ |
| $\hat{p}_t(z)$ | Intermediate mixture density: $\hat{p}_t(z) = \frac{1}{N} \sum_{i=1}^N p_t(z \mid x^{(i)})$ |
| $\lambda_i(z, t)$ | Posterior responsibility: $\lambda_i(z, t) = \frac{p_t(z\mid x^{(i)})}{\sum_j p_t(z\mid x^{(j)})}$ |
| $\hat{u}^*(z, t)$ | Empirical optimal velocity field (see Lemma B.2) |
| $\alpha(t)$ | Velocity field coefficient: $\alpha(t) = \frac{\dot{\gamma}(t)\sigma_t^2}{\gamma(t)(1-\gamma(t))}$ |
| $\beta(t)$ | Velocity field coefficient: $\beta(t) = \frac{\dot{\gamma}(t)}{\gamma(t)}$ |
| $m(t)$ | Combined coefficient: $m(t) = \beta(t) - \frac{\alpha(t)}{\sigma_t^2} = -\frac{\dot{\gamma}(t)}{1-\gamma(t)}$ |

### B.1.2. MAIN THEOREM

Before proceeding to the detailed derivation, we state our main theoretical result, which establishes the quantitative relationship between instantaneous energy and mixture density.

**Theorem B.1** (Instantaneous Energy vs. Mixture Density). *Consider a closed-form flow matching model with the following intermediate mixture density*

$$\hat{p}_t(z) = \frac{1}{N} \sum_{i=1}^N p_t(z \mid x^{(i)}). \tag{17}$$

*Fix $t \in (0, 1)$ with $\gamma(t) \in (0, 1)$ and $\dot{\gamma}(t) \neq 0$, and fix $z \in \mathbb{R}^d$. Suppose there exists an index $i^*$ and a parameter $\varepsilon \in (0, 1/2)$ such that*

$$\lambda_{i^*}(z, t) := \frac{p_t(z \mid x^{(i^*)})}{\sum_{j=1}^N p_t(z \mid x^{(j)})} \geq 1 - \varepsilon. \tag{18}$$

*Then there exist constants $c_1'(t), c_2'(t) > 0$ and $C_t' \in \mathbb{R}$, depending only on $t$, $\varepsilon$, the dimension $d$, the sample size $N$, and the data $\{x^{(i)}\}$, such that*

$$c_1'(t)\left(-\log \hat{p}_t(z)\right) - C_t' \leq \left\|\hat{u}^*(z, t)\right\|^2 \leq c_2'(t)\left(-\log \hat{p}_t(z)\right) + C_t'. \tag{19}$$

*In particular, under posterior dominance, $\|\hat{u}^*(z,t)\|^2$ is comparable to $-\log\hat{p}_t(z)$ up to multiplicative and additive constants.*

*Moreover, the constants can be chosen explicitly as:*

$$c_1'(t) = \frac{1}{2}m(t)^2\sigma_t^2, \qquad c_2'(t) = 12\,m(t)^2\sigma_t^2, \tag{20}$$

*where $m(t) = -\frac{\dot{\gamma}(t)}{1-\gamma(t)}$.*

The proof of this theorem is deferred to §B.4.1, after we establish the necessary technical lemmas.

**Remark.** Theorem B.1 shows that, under posterior dominance, the instantaneous kinetic energy $\|\hat{u}^*(z,t)\|^2$ is comparable to the negative log-density $-\log\hat{p}_t(z)$ up to multiplicative and additive constants. Thus, regions with sufficiently large $-\log\hat{p}_t(z)$ have correspondingly large instantaneous energy, up to additive constants. Applied along trajectory segments where posterior dominance holds and away from singular endpoints, this provides a theoretical basis for the KPE–density trend in Finding 2.

### B.1.3. PROOF ROADMAP

We now outline the overall proof strategy for Theorem B.1 and describe the role of each intermediate lemma.

**Proof Strategy.** The core challenge is to establish a quantitative relationship between the instantaneous energy $\|\hat{u}^*(z,t)\|^2$ and the negative log-density $-\log\hat{p}_t(z)$, when one mixture component dominates at $(z,t)$ (i.e., $\lambda_{i^*}(z,t) \geq 1-\varepsilon$ for small $\varepsilon$). Our strategy proceeds in four steps:

1. **Express velocity in terms of mixture score** (Lemma B.2)

   We first derive a closed-form expression for the optimal velocity field $\hat{u}^*(z,t)$ in terms of the mixture score $\nabla_z\log\hat{p}_t(z)$. Specifically, we show that

   $$\hat{u}^*(z,t) = \alpha(t)\,\nabla_z\log\hat{p}_t(z) + \beta(t)\,z, \tag{21}$$

   where $\alpha(t)$ and $\beta(t)$ are explicit functions of $\gamma(t)$ and $\dot{\gamma}(t)$. This decomposition is the foundation for all subsequent analysis.

2. **Local Gaussian approximation for the mixture density** (Lemma B.3)

   Under the posterior dominance condition, we show that the mixture density $\hat{p}_t(z)$ can be locally approximated by a single dominant Gaussian component. Specifically, we establish that

   $$-\log\hat{p}_t(z) = \frac{1}{2\sigma_t^2}\left\|z - \mu_{i^*}(t)\right\|^2 + C_t^{(0)} + R_t(z), \tag{22}$$

   where $C_t^{(0)}$ is a constant depending only on $t$, $d$, and $N$, and the remainder $R_t(z)$ is quantitatively bounded in terms of the dominance parameter $\varepsilon$.

   This lemma establishes the relationship between $-\log\hat{p}_t(z)$ and the quadratic form $A_t(z) := \frac{1}{2\sigma_t^2}\|z - \mu_{i^*}(t)\|^2$.

3. **Decompose mixture score into dominant component plus remainder** (Lemma B.4)

   Similarly, we decompose the mixture score as

   $$\nabla_z\log\hat{p}_t(z) = -\frac{1}{\sigma_t^2}\left(z - \mu_{i^*}(t)\right) + r_t(z), \tag{23}$$

   where $r_t(z)$ is a remainder term that is explicitly bounded in terms of $\varepsilon$, $\sigma_t^2$, and the spread of mixture means.

4. **Establish energy bounds in terms of the dominant quadratic form** (Lemma B.5)

   Combining Lemma B.2 and Lemma B.4, we derive explicit upper and lower bounds for the instantaneous energy $\|\hat{u}^*\|^2$ in terms of the quadratic form $A_t(z)$:

   $$c_1(t)\,A_t(z) - C_-(t) \;\leq\; \left\|\hat{u}^*(z,t)\right\|^2 \;\leq\; c_2(t)\,A_t(z) + C_+(t), \tag{24}$$

   with explicit constants $c_1(t), c_2(t), C_\pm(t)$.

Finally, we combine Lemma B.3 (relating $A_t(z)$ to $-\log \hat{p}_t(z)$) and Lemma B.5 (relating $\|\hat{u}^*\|^2$ to $A_t(z)$) to obtain Theorem B.1, which directly relates $\|\hat{u}^*\|^2$ to $-\log \hat{p}_t(z)$.

**Organization.** The remainder of this section is organized as follows:

- §B.2: We define the closed-form interpolating bridge and intermediate mixture density.

- §B.3: We derive the velocity-score representation (Lemma B.2).

- §B.4: We establish the energy-density relationship through three technical lemmas (Lemma B.3, Lemma B.4, Lemma B.5) and prove the main theorem (Theorem B.1).

- §B.5: We specialize the results to the linear bridge $\gamma(t) = t$ and provide explicit formulas for the constants appearing in the theorem.

With this roadmap in place, we now proceed to the detailed technical development.

## B.2. Closed-Form Interpolating Bridge and Intermediate Mixture

We now establish the basic setup by defining the conditional bridge and the intermediate mixture density that will be analyzed throughout this section.

Let $\{x^{(i)}\}_{i=1}^N \subset \mathbb{R}^d$ be the training data and let $x_0 \sim \mathcal{N}(0, I_d)$ be a random variable with the source distribution. For each data point $x^{(i)}$, we consider the conditional bridge

$$z_t = (1 - \gamma(t)) x_0 + \gamma(t) x^{(i)}, \qquad t \in [0,1], \tag{25}$$

where $\gamma : [0,1] \to [0,1]$ is differentiable with $\gamma(0) = 0$ and $\gamma(1) = 1$. Note that the analysis that follows could also be extended straightforwardly to the more general case of $z_t = \beta(t)x_0 + \gamma(t)x^{(i)}$ for some $t$-differentiable $\beta(t)$ and $\gamma(t)$ satisfying $\beta(0) = \gamma(1) = 1$, $\beta(1) = \gamma(0) = 0$.

For fixed $t$ and $x^{(i)}$, the random variable $z_t$ is an affine transformation of $x_0$ and thus follows a Gaussian distribution:

$$p_t(z \mid x^{(i)}) = \mathcal{N}(z; \mu_i(t), \Sigma_t), \quad \mu_i(t) = \gamma(t)x^{(i)}, \ \Sigma_t = \sigma_t^2 I_d, \ \sigma_t^2 = (1 - \gamma(t))^2. \tag{26}$$

Averaging over the empirical data distribution $\hat{p}_{\text{data}} = \frac{1}{N} \sum_i \delta_{x^{(i)}}$ yields the intermediate density mixture

$$\hat{p}_t(z) = \frac{1}{N} \sum_{i=1}^N p_t(z \mid x^{(i)}). \tag{27}$$

Thus $\hat{p}_t$ is exactly the marginal density of $z_t$ under the model where $i$ is sampled uniformly from $\{1, \ldots, N\}$ and $x_0 \sim \mathcal{N}(0, I_d)$.

## B.3. Empirical Optimal Velocity and the Mixture Score

Given a finite training set $\{x^{(i)}\}_{i=1}^N$, we first establish the optimal flow matching velocity field $\hat{u}^*(z, t)$ in terms of the mixture score $\nabla_z \log \hat{p}_t(z)$. This lemma provides the foundation for all subsequent analysis, as it decomposes the velocity field into a score term (weighted by $\alpha(t)$) and a drift term (weighted by $\beta(t)$). The explicit representation derived here will be essential for relating instantaneous energy to density in later sections.

Consider the bridge (25) and the intermediate mixture (27). For $t \in (0, 1)$ with $\gamma(t) \in (0, 1)$ and $\dot{\gamma}(t) \neq 0$, define

$$\lambda_i(z, t) = \mathbb{P}(i \mid z, t) = \frac{p_t(z \mid x^{(i)})}{\sum_{j=1}^N p_t(z \mid x^{(j)})}, \tag{28}$$

the posterior responsibilities of the mixture components.

**Lemma B.2** (Closed-form optimal velocity). *The empirical optimal velocity field $\hat{u}^*(z,t)$ can be written as*

$$\hat{u}^*(z,t) = \alpha(t)\,\nabla_z \log \hat{p}_t(z) + \beta(t)\,z, \quad \alpha(t) = \frac{\dot{\gamma}(t)(1-\gamma(t))}{\gamma(t)}, \quad \beta(t) = \frac{\dot{\gamma}(t)}{\gamma(t)}. \tag{29}$$

*Proof.* By definition of the bridge (25), for fixed $(x_0, x^{(i)})$ we have

$$z_t = (1 - \gamma(t))x_0 + \gamma(t)x^{(i)}. \tag{30}$$

Differentiating with respect to $t$ yields the conditional velocity

$$\dot{z}_t = u_{\text{cond}}(z_t, t, x^{(i)}, x_0) = -\dot{\gamma}(t)\,x_0 + \dot{\gamma}(t)\,x^{(i)} = \dot{\gamma}(t)\,(x^{(i)} - x_0). \tag{31}$$

We now express $x_0$ in terms of $(z, t, x^{(i)})$ where $z = z_t$. From $z = (1-\gamma)x_0 + \gamma x^{(i)}$ we obtain

$$x_0 = \frac{\gamma(t)x^{(i)} - z}{\gamma(t) - 1}. \tag{32}$$

Substituting into (31) gives

$$x^{(i)} - x_0 = x^{(i)} - \frac{\gamma x^{(i)} - z}{\gamma - 1} = \frac{z - x^{(i)}}{\gamma - 1} = \frac{x^{(i)} - z}{1 - \gamma(t)}, \tag{33}$$

and hence

$$u_{\text{cond}}(z, t, x^{(i)}) = \dot{\gamma}(t)\,(x^{(i)} - x_0) = \frac{\dot{\gamma}(t)}{1 - \gamma(t)}\,(x^{(i)} - z). \tag{34}$$

The optimal velocity $\hat{u}^*(z,t)$ is the conditional expectation of (34), where the randomness is over the choice of $x^{(i)}$ (equivalently, the index $i$). Using the posterior weights (28), we obtain

$$\hat{u}^*(z,t) = \mathbb{E}\big[u_{\text{cond}}(z, t, x^{(i)}) \,\big|\, z, t\big] = \sum_{i=1}^{N} \lambda_i(z,t)\, u_{\text{cond}}(z, t, x^{(i)}) = \frac{\dot{\gamma}(t)}{1 - \gamma(t)} \sum_{i=1}^{N} \lambda_i(z,t)\,(x^{(i)} - z). \tag{35}$$

We now relate the mixture score $\nabla_z \log \hat{p}_t(z)$ to the same posterior weights. For the Gaussian mixture $\hat{p}_t(z) = \frac{1}{N}\sum_i \mathcal{N}(z; \mu_i(t), \Sigma_t)$ with common covariance $\Sigma_t = \sigma_t^2 I_d$, the gradient of the log-density can be computed to be:

$$\nabla_z \log \hat{p}_t(z) = \sum_{i=1}^{N} \lambda_i(z,t)\, \Sigma_t^{-1}\big(\mu_i(t) - z\big), \tag{36}$$

which is the standard expression for the score of a Gaussian mixture. Since $\Sigma_t = \sigma_t^2 I_d$, we have $\Sigma_t^{-1} = \frac{1}{\sigma_t^2} I_d$, so

$$\nabla_z \log \hat{p}_t(z) = \frac{1}{\sigma_t^2} \sum_{i=1}^{N} \lambda_i(z,t)\,\big(\mu_i(t) - z\big), \qquad \mu_i(t) = \gamma(t)x^{(i)}. \tag{37}$$

Next, we express the sum $S_1 := \sum_i \lambda_i(z,t)(x^{(i)} - z)$ in (35) in terms of the sum $S_2 := \sum_i \lambda_i(z,t)(\mu_i(t) - z)$ appearing in (37). Note that

$$\mu_i(t) - z = \gamma(t)x^{(i)} - z, \tag{38}$$

and we can rewrite

$$x^{(i)} - z = \frac{1}{\gamma(t)}\big(\gamma(t)x^{(i)} - \gamma(t)z\big) = \frac{1}{\gamma(t)}\big(\mu_i(t) - z\big) + \frac{1 - \gamma(t)}{\gamma(t)}\,z. \tag{39}$$

Therefore,

$$S_1 = \sum_{i=1}^{N} \lambda_i(z,t)\,(x^{(i)} - z) \tag{40}$$

$$= \sum_{i=1}^{N} \lambda_i(z,t) \left[ \frac{1}{\gamma(t)} \big(\mu_i(t) - z\big) + \frac{1-\gamma(t)}{\gamma(t)}\, z \right] \tag{41}$$

$$= \frac{1}{\gamma(t)} \sum_{i=1}^{N} \lambda_i(z,t)\big(\mu_i(t) - z\big) + \frac{1-\gamma(t)}{\gamma(t)}\, z \sum_{i=1}^{N} \lambda_i(z,t) \tag{42}$$

$$= \frac{1}{\gamma(t)} S_2 + \frac{1-\gamma(t)}{\gamma(t)}\, z, \tag{43}$$

where we used $\sum_i \lambda_i(z,t) = 1$. Using $S_2 = \sigma_t^2 \nabla_z \log \hat{p}_t(z)$ from (37), we obtain

$$S_1 = \frac{\sigma_t^2}{\gamma(t)} \nabla_z \log \hat{p}_t(z) + \frac{1-\gamma(t)}{\gamma(t)}\, z. \tag{44}$$

Substituting this into (35) and using the formula $\sigma_t^2 = (1-\gamma(t))^2$ yield

$$\hat{u}^*(z,t) = \frac{\dot{\gamma}(t)}{1-\gamma(t)} S_1 \tag{45}$$

$$= \frac{\dot{\gamma}(t)}{1-\gamma(t)} \left[ \frac{\sigma_t^2}{\gamma(t)} \nabla_z \log \hat{p}_t(z) + \frac{1-\gamma(t)}{\gamma(t)}\, z \right] \tag{46}$$

$$= \underbrace{\frac{\dot{\gamma}(t)(1-\gamma(t))}{\gamma(t)}}_{\alpha(t)} \nabla_z \log \hat{p}_t(z) + \underbrace{\frac{\dot{\gamma}(t)}{\gamma(t)}}_{\beta(t)}\, z, \tag{47}$$

which is exactly (29). □

## B.4. Instantaneous Energy vs. Negative Log-Density

We now establish the quantitative relationship between the instantaneous energy $\|\hat{u}^*(z,t)\|^2$ and the negative log-density $-\log \hat{p}_t(z)$. The key technical condition is posterior dominance: at each point $(z,t)$, we assume there exists a dominant mixture component $i^*$ such that $\lambda_{i^*}(z,t) \geq 1 - \varepsilon$ for some small $\varepsilon \in (0, 1/2)$.

Under this condition, we will show that the mixture density $\hat{p}_t(z)$ and mixture score $\nabla_z \log \hat{p}_t(z)$ can both be well-approximated by the corresponding quantities for the dominant Gaussian component. This local approximation enables us to derive explicit bounds relating the instantaneous energy to the negative log-density.

The development proceeds through three lemmas:

- Lemma B.3 establishes a local Gaussian approximation for $-\log \hat{p}_t(z)$ in terms of the dominant quadratic form.

- Lemma B.4 decomposes the mixture score as the dominant component's score plus a controlled remainder.

- Lemma B.5 combines these results with Lemma B.2 to bound the instantaneous energy.

Finally, we combine all lemmas to prove Theorem B.1.

**Lemma 2: Local Gaussian Approximation.** The following lemma shows that under posterior dominance, the mixture log-density can be locally approximated by the log-density of a single dominant Gaussian component, with a quantitatively controlled remainder which is small when $\varepsilon$ is small.

**Lemma B.3** (Local Gaussian approximation with quantitative constants). *Fix $t \in (0,1)$ and $z \in \mathbb{R}^d$. Suppose there exists an index $i^*$ and a parameter $\varepsilon \in (0, 1/2)$ such that the posterior responsibility*

$$\lambda_{i^*}(z,t) = \frac{p_t(z \mid x^{(i^*)})}{\sum_{j=1}^{N} p_t(z \mid x^{(j)})} \geq 1 - \varepsilon.$$

*Let $\sigma_t^2 = (1 - \gamma(t))^2$ and $\mu_{i^*}(t) = \gamma(t) \, x^{(i^*)}$. Then*

$$-\log \hat{p}_t(z) = \frac{1}{2\sigma_t^2} \left\| z - \mu_{i^*}(t) \right\|^2 + C_t^{(0)} + R_t(z), \tag{48}$$

*where*

$$C_t^{(0)} := \frac{d}{2} \log(2\pi) + \frac{d}{2} \log \sigma_t^2 + \log N, \tag{49}$$

*and the remainder $R_t(z)$ satisfies*

$$\log(1 - \varepsilon) \leq R_t(z) \leq 0, \qquad \left| R_t(z) \right| \leq -\log(1 - \varepsilon). \tag{50}$$

*Proof.* We have

$$\hat{p}_t(z) = \frac{1}{N} \sum_{i=1}^{N} p_t(z \mid x^{(i)}) = \frac{1}{N} \, p_t(z \mid x^{(i^*)}) \left[ 1 + \sum_{j \neq i^*} \frac{p_t(z \mid x^{(j)})}{p_t(z \mid x^{(i^*)})} \right].$$

Define

$$\delta(z, t) := \sum_{j \neq i^*} \frac{p_t(z \mid x^{(j)})}{p_t(z \mid x^{(i^*)})} \geq 0.$$

Then

$$\hat{p}_t(z) = \frac{1}{N} \, p_t(z \mid x^{(i^*)}) \big( 1 + \delta(z, t) \big).$$

The dominance assumption $\lambda_{i^*}(z, t) \geq 1 - \varepsilon$ implies

$$\lambda_{i^*}(z, t) = \frac{p_t(z \mid x^{(i^*)})}{\sum_k p_t(z \mid x^{(k)})} \geq 1 - \varepsilon \implies \sum_k p_t(z \mid x^{(k)}) \leq \frac{1}{1 - \varepsilon} p_t(z \mid x^{(i^*)}).$$

Hence

$$\sum_{j \neq i^*} p_t(z \mid x^{(j)}) \leq \frac{1}{1 - \varepsilon} p_t(z \mid x^{(i^*)}) - p_t(z \mid x^{(i^*)}) = \frac{\varepsilon}{1 - \varepsilon} p_t(z \mid x^{(i^*)}),$$

which gives

$$0 \leq \delta(z, t) = \frac{1}{p_t(z \mid x^{(i^*)})} \sum_{j \neq i^*} p_t(z \mid x^{(j)}) \leq \frac{\varepsilon}{1 - \varepsilon}.$$

Thus

$$1 \leq 1 + \delta(z, t) \leq \frac{1}{1 - \varepsilon}.$$

Taking logarithms,

$$\log \hat{p}_t(z) = \log \frac{1}{N} + \log p_t(z \mid x^{(i^*)}) + \log(1 + \delta(z, t)).$$

For the Gaussian component $p_t(z \mid x^{(i^*)}) = \mathcal{N}(z; \mu_{i^*}(t), \sigma_t^2 I_d)$,

$$\log p_t(z \mid x^{(i^*)}) = -\frac{d}{2} \log(2\pi) - \frac{d}{2} \log \sigma_t^2 - \frac{1}{2\sigma_t^2} \left\| z - \mu_{i^*}(t) \right\|^2.$$

Therefore,

$$-\log \hat{p}_t(z) = \frac{1}{2\sigma_t^2} \left\| z - \mu_{i^*}(t) \right\|^2 + \frac{d}{2} \log(2\pi) + \frac{d}{2} \log \sigma_t^2 + \log N - \log \big( 1 + \delta(z, t) \big).$$

This is (48) with

$$C_t^{(0)} := \frac{d}{2} \log(2\pi) + \frac{d}{2} \log \sigma_t^2 + \log N, \quad R_t(z) := -\log \big( 1 + \delta(z, t) \big).$$

Since $1 \leq 1 + \delta(z,t) \leq 1/(1-\varepsilon)$, we have

$$0 \leq \log\big(1 + \delta(z,t)\big) \leq \log\frac{1}{1-\varepsilon},$$

so

$$\log(1-\varepsilon) \leq R_t(z) \leq 0,$$

and $\big|R_t(z)\big| \leq -\log(1-\varepsilon)$. $\qquad\square$

**Lemma 3: Mixture Score Decomposition.** Just as the mixture density can be locally approximated by the dominant component, the mixture score $\nabla_z \log \hat{p}_t(z)$ can be decomposed as the dominant component's score plus a small remainder. This decomposition is crucial for controlling the instantaneous energy.

**Lemma B.4** (Mixture score versus dominant component). *Under the assumptions of Lemma B.3, the score of the mixture can be written as*

$$\nabla_z \log \hat{p}_t(z) = -\frac{1}{\sigma_t^2}\big(z - \mu_{i^*}(t)\big) + r_t(z), \tag{51}$$

*where the remainder $r_t(z)$ admits the explicit bound*

$$\big\|r_t(z)\big\| \leq \frac{\varepsilon}{\sigma_t^2}\,\Delta_t, \qquad \Delta_t := \max_{1 \leq j \leq N}\big\|\mu_j(t) - \mu_{i^*}(t)\big\|. \tag{52}$$

*Proof.* By definition of the mixture score with common covariance $\Sigma_t = \sigma_t^2 I_d$,

$$\nabla_z \log \hat{p}_t(z) = \frac{1}{\sigma_t^2}\sum_{i=1}^{N}\lambda_i(z,t)\big(\mu_i(t) - z\big).$$

The score of the single Gaussian component $\mathcal{N}(\mu_{i^*}(t), \sigma_t^2 I_d)$ is

$$\frac{1}{\sigma_t^2}\big(\mu_{i^*}(t) - z\big) = -\frac{1}{\sigma_t^2}\big(z - \mu_{i^*}(t)\big).$$

Define

$$r_t(z) := \nabla_z \log \hat{p}_t(z) + \frac{1}{\sigma_t^2}\big(z - \mu_{i^*}(t)\big).$$

Substituting the expressions above,

$$r_t(z) = \frac{1}{\sigma_t^2}\left(\sum_{i=1}^{N}\lambda_i(z,t)\mu_i(t) - \mu_{i^*}(t)\right) = \frac{1}{\sigma_t^2}\sum_{j \neq i^*}\lambda_j(z,t)\big(\mu_j(t) - \mu_{i^*}(t)\big).$$

Taking norms and using the triangle inequality,

$$\big\|r_t(z)\big\| \leq \frac{1}{\sigma_t^2}\sum_{j \neq i^*}\lambda_j(z,t)\big\|\mu_j(t) - \mu_{i^*}(t)\big\| \leq \frac{1}{\sigma_t^2}\left(\max_j\big\|\mu_j(t) - \mu_{i^*}(t)\big\|\right)\sum_{j \neq i^*}\lambda_j(z,t).$$

By the dominance assumption $\lambda_{i^*}(z,t) \geq 1 - \varepsilon$,

$$\sum_{j \neq i^*}\lambda_j(z,t) = 1 - \lambda_{i^*}(z,t) \leq \varepsilon.$$

Hence

$$\big\|r_t(z)\big\| \leq \frac{\varepsilon}{\sigma_t^2}\max_j\big\|\mu_j(t) - \mu_{i^*}(t)\big\| = \frac{\varepsilon}{\sigma_t^2}\,\Delta_t.$$

$\qquad\square$

**Lemma 4: Instantaneous Energy Bounds.** We now combine Lemma B.2 (velocity-score representation) and Lemma B.4 (score decomposition) to establish explicit upper and lower bounds for the instantaneous energy $\|\hat{u}^*(z,t)\|^2$ in terms of the dominant quadratic form $A_t(z) := \frac{1}{2\sigma_t^2}\|z - \mu_{i^*}(t)\|^2$. This lemma provides the key technical link between energy and the distance from the dominant component.

The proof employs careful norm inequalities to derive bounds with explicit multiplicative constants $c_1(t), c_2(t)$ and additive constants $C_\pm(t)$. These explicit constants will propagate through to our main theorem.

**Lemma B.5** (Instantaneous energy vs. dominant quadratic form). *Fix $t \in (0,1)$ and suppose $\gamma(t) \in (0,1)$ and $\dot{\gamma}(t) \neq 0$. Let $\sigma_t^2 = (1 - \gamma(t))^2$ and $\mu_i(t) = \gamma(t)x^{(i)}$. Assume that at $(z,t)$ there exists an index $i^*$ and $\varepsilon \in (0, 1/2)$ such that $\lambda_{i^*}(z,t) \geq 1 - \varepsilon$.*

*Define the quadratic form*

$$A_t(z) := \frac{1}{2\sigma_t^2}\left\|z - \mu_{i^*}(t)\right\|^2.$$

*Let*

$$\alpha(t) = \frac{\dot{\gamma}(t)\,\sigma_t^2}{\gamma(t)(1 - \gamma(t))}, \qquad \beta(t) = \frac{\dot{\gamma}(t)}{\gamma(t)},$$

*and*

$$m(t) := \beta(t) - \frac{\alpha(t)}{\sigma_t^2} = -\frac{\dot{\gamma}(t)}{1 - \gamma(t)} \neq 0.$$

*Set*

$$b_t := \frac{\alpha(t)}{\sigma_t^2}\,\mu_{i^*}(t), \qquad \Delta_t := \max_j \left\|\mu_j(t) - \mu_{i^*}(t)\right\|,$$

$$R_t := \frac{\varepsilon}{\sigma_t^2}\,\Delta_t, \qquad E_t := |\alpha(t)|\,R_t, \qquad F_t := \|b_t\| + E_t.$$

*Then the instantaneous energy satisfies*

$$c_1(t)\,A_t(z) - C_-(t) \;\leq\; \left\|\hat{u}^*(z,t)\right\|^2 \;\leq\; c_2(t)\,A_t(z) + C_+(t), \tag{53}$$

*with explicit constants*

$$c_1(t) = \frac{1}{2}\,m(t)^2\,\sigma_t^2, \qquad c_2(t) = 12\,m(t)^2\,\sigma_t^2,$$

*and*

$$C_-(t) := \frac{m(t)^2}{2}\left\|\mu_{i^*}(t)\right\|^2 + 2F_t^2, \qquad C_+(t) := 6\,m(t)^2\left\|\mu_{i^*}(t)\right\|^2 + 3\left(\|b_t\|^2 + E_t^2\right).$$

*Proof.* By Lemma B.2 and Lemma B.4,

$$\hat{u}^*(z,t) = \alpha(t)\,\nabla_z \log \hat{p}_t(z) + \beta(t)\,z = \alpha(t)\left[-\frac{1}{\sigma_t^2}\left(z - \mu_{i^*}(t)\right) + r_t(z)\right] + \beta(t)\,z.$$

Rearrange as

$$\hat{u}^*(z,t) = \left(\beta(t) - \frac{\alpha(t)}{\sigma_t^2}\right)z + \frac{\alpha(t)}{\sigma_t^2}\mu_{i^*}(t) + \alpha(t)\,r_t(z) = M_t z + b_t + e_t(z),$$

where $M_t := m(t)I_d$, $b_t := \frac{\alpha(t)}{\sigma_t^2}\mu_{i^*}(t)$ and $e_t(z) := \alpha(t)r_t(z)$.

By Lemma B.4,

$$\|r_t(z)\| \leq R_t = \frac{\varepsilon}{\sigma_t^2}\Delta_t, \quad \Rightarrow \quad \|e_t(z)\| \leq |\alpha(t)|\,R_t =: E_t,$$

so $\|e_t(z)\|$ is bounded uniformly in $z$.

**Upper bound.** Using $\|a + b + c\|^2 \leq 3(\|a\|^2 + \|b\|^2 + \|c\|^2)$,

$$\left\|\hat{u}^*(z,t)\right\|^2 = \left\|M_t z + b_t + e_t(z)\right\|^2 \leq 3\left(\|M_t z\|^2 + \|b_t\|^2 + \|e_t(z)\|^2\right) \leq 3m(t)^2\|z\|^2 + 3\left(\|b_t\|^2 + E_t^2\right).$$

Using $\|z\|^2 \leq 2\|z - \mu_{i^*}(t)\|^2 + 2\|\mu_{i^*}(t)\|^2$, we get

$$\left\|\hat{u}^*(z,t)\right\|^2 \leq 3m(t)^2\big(2\|z - \mu_{i^*}(t)\|^2 + 2\|\mu_{i^*}(t)\|^2\big) + 3\big(\|b_t\|^2 + E_t^2\big),$$

hence

$$\left\|\hat{u}^*(z,t)\right\|^2 \leq 6m(t)^2\|z - \mu_{i^*}(t)\|^2 + C_+(t),$$

with $C_+(t)$ as stated. Using $\|z - \mu_{i^*}(t)\|^2 = 2\sigma_t^2 A_t(z)$, we obtain

$$\left\|\hat{u}^*(z,t)\right\|^2 \leq 12\,m(t)^2\,\sigma_t^2\,A_t(z) + C_+(t),$$

which gives the upper bound in (53) with $c_2(t) = 12\,m(t)^2\,\sigma_t^2$.

**Lower bound.** Now, expanding the squared norm and using Cauchy-Schwarz inequality,

$$\begin{aligned}
\left\|\hat{u}^*(z,t)\right\|^2 &= \|M_t z + b_t + e_t(z)\|^2 \\
&= \|M_t z\|^2 + 2\langle M_t z, b_t + e_t(z)\rangle + \|b_t + e_t(z)\|^2 \\
&\geq \|M_t z\|^2 + 2\langle M_t z, b_t + e_t(z)\rangle \\
&\geq \|M_t z\|^2 - 2\|M_t z\| \cdot \|b_t + e_t(z)\| =: A^2 - 2AB,
\end{aligned}$$

where $A = \|M_t z\|$ and $B = \|b_t + e_t(z)\|$.

Applying Young's inequality $2AB \leq \frac{1}{2}A^2 + 2B^2$, we obtain:

$$\begin{aligned}
\left\|\hat{u}^*(z,t)\right\|^2 &\geq A^2 - 2AB \\
&\geq A^2 - \left(\frac{1}{2}A^2 + 2B^2\right) = \frac{1}{2}A^2 - 2B^2.
\end{aligned}$$

We bound $B^2 \leq (\|b_t\| + \|e_t(z)\|)^2 \leq (\|b_t\| + E_t)^2 =: F_t^2$, so

$$\left\|\hat{u}^*(z,t)\right\|^2 \geq \frac{1}{2}\|M_t z\|^2 - 2F_t^2 = \frac{1}{2}m(t)^2\,\|z\|^2 - 2F_t^2.$$

Next, we relate $\|z\|^2$ and $\|z - \mu_{i^*}(t)\|^2$. From

$$\|z - \mu_{i^*}(t)\|^2 = \|z\|^2 + \|\mu_{i^*}(t)\|^2 - 2\langle z, \mu_{i^*}(t)\rangle \leq 2\|z\|^2 + 2\|\mu_{i^*}(t)\|^2,$$

we obtain

$$\|z\|^2 \geq \frac{1}{2}\|z - \mu_{i^*}(t)\|^2 - \|\mu_{i^*}(t)\|^2.$$

Substituting into the previous bound,

$$\begin{aligned}
\left\|\hat{u}^*(z,t)\right\|^2 &\geq \frac{1}{2}m(t)^2\left(\frac{1}{2}\|z - \mu_{i^*}(t)\|^2 - \|\mu_{i^*}(t)\|^2\right) - 2F_t^2 \\
&= \frac{m(t)^2}{4}\|z - \mu_{i^*}(t)\|^2 - \frac{m(t)^2}{2}\|\mu_{i^*}(t)\|^2 - 2F_t^2.
\end{aligned}$$

In terms of $A_t(z)$, we have $\|z - \mu_{i^*}(t)\|^2 = 2\sigma_t^2 A_t(z)$, thus

$$\frac{m(t)^2}{4}\|z - \mu_{i^*}(t)\|^2 = \frac{m(t)^2}{4}\cdot 2\sigma_t^2 A_t(z) = \frac{1}{2}m(t)^2\sigma_t^2 A_t(z).$$

Therefore

$$\left\|\hat{u}^*(z,t)\right\|^2 \geq \underbrace{\frac{1}{2}m(t)^2\sigma_t^2}_{c_1(t)}\,A_t(z) - \underbrace{\left(\frac{m(t)^2}{2}\|\mu_{i^*}(t)\|^2 + 2F_t^2\right)}_{C_-(t)},$$

which is the lower bound in (53). $\qquad\square$

### B.4.1. PROOF OF THE MAIN THEOREM

We are now ready to prove Theorem B.1, which establishes the direct relationship between instantaneous energy $\|\hat{u}^*\|^2$ and negative log-density $-\log \hat{p}_t(z)$. The proof combines Lemma B.3 (which relates $-\log \hat{p}_t$ to the quadratic form $A_t(z)$) with Lemma B.5 (which relates $\|\hat{u}^*\|^2$ to $A_t(z)$).

*Proof of Theorem B.1.* From Lemma B.3,

$$-\log \hat{p}_t(z) = A_t(z) + C_t^{(0)} + R_t(z),$$

with $A_t(z) = \frac{1}{2\sigma_t^2}\|z - \mu_{i^*}(t)\|^2$ and $\left|R_t(z)\right| \leq -\log(1-\varepsilon) =: C_t^{(\mathrm{mix})}$. Thus

$$A_t(z) = -\log \hat{p}_t(z) - C_t^{(0)} - R_t(z),$$

and hence

$$-\log \hat{p}_t(z) - K_t \ \leq \ A_t(z) \ \leq \ -\log \hat{p}_t(z) + K_t,$$

where

$$K_t := \left|C_t^{(0)}\right| + C_t^{(\mathrm{mix})} = \left|\frac{d}{2}\log(2\pi) + \frac{d}{2}\log \sigma_t^2 + \log N\right| - \log(1-\varepsilon).$$

By Lemma B.5,

$$c_1(t)\, A_t(z) - C_-(t) \leq \|\hat{u}^*(z,t)\|^2 \leq c_2(t)\, A_t(z) + C_+(t),$$

with explicit $c_1(t), c_2(t), C_\pm(t)$ as in Lemma B.5.

For the lower bound, we use $A_t(z) \geq -\log \hat{p}_t(z) - K_t$:

$$\|\hat{u}^*(z,t)\|^2 \geq c_1(t)\big(-\log \hat{p}_t(z) - K_t\big) - C_-(t) = c_1(t)\big(-\log \hat{p}_t(z)\big) - \big(C_-(t) + c_1(t)K_t\big).$$

For the upper bound, we use $A_t(z) \leq -\log \hat{p}_t(z) + K_t$:

$$\|\hat{u}^*(z,t)\|^2 \leq c_2(t)\big(-\log \hat{p}_t(z) + K_t\big) + C_+(t) = c_2(t)\big(-\log \hat{p}_t(z)\big) + \big(C_+(t) + c_2(t)K_t\big).$$

Therefore (19) holds with

$$c_1'(t) := c_1(t) = \frac{1}{2}m(t)^2\sigma_t^2, \qquad c_2'(t) := c_2(t) = 12\, m(t)^2\sigma_t^2,$$

$$C_t' := \max\big\{C_-(t) + c_1(t)K_t,\ C_+(t) + c_2(t)K_t\big\}.$$

$\square$

### B.4.2. CONNECTION TO KINETIC PATH ENERGY

Finally, we connect Theorem B.1 to the kinetic path energy defined in the main paper. For a sample trajectory $z_{0\to1} = (z(t))_{t\in[0,1]}$ generated by the flow matching sampler, the kinetic path energy is

$$E(z_{0\to1}) = \frac{1}{2}\int_0^1 \big\|\hat{u}^*(z(t),t)\big\|^2 \, dt. \tag{54}$$

On any compact interval $[\delta, 1-\delta] \subset (0,1)$ where the posterior dominance condition holds along the trajectory, the pointwise bounds (19) imply that:

$$\frac{1}{2}\int_\delta^{1-\delta} c_1'(t)\left(-\log \hat{p}_t(z(t))\right) dt - \frac{1}{2}\int_\delta^{1-\delta} C_t' \, dt \ \leq \ E_\delta(z) \ \leq \ \frac{1}{2}\int_\delta^{1-\delta} c_2'(t)\left(-\log \hat{p}_t(z(t))\right) dt + \frac{1}{2}\int_\delta^{1-\delta} C_t' \, dt, \tag{55}$$

where $E_\delta(z) := \frac{1}{2}\int_\delta^{1-\delta} \|\hat{u}^*(z(t),t)\|^2 \, dt$.

Thus, on trajectory segments away from the singular endpoints and under posterior dominance, kinetic energy is controlled by the time-integrated negative log-density up to time-dependent prefactors and additive constants. This supports the interpretation that trajectory segments passing through lower density regions of the intermediate mixture tend to incur larger kinetic energy, consistent with our empirical findings.

## B.5. Explicit Constants for the Linear Bridge

The most commonly used choice in practice is the linear interpolation $\gamma(t) = t$. For this choice, all the constants in Theorem B.1 can be computed explicitly, yielding particularly simple and interpretable expressions. This special case corresponds to the standard straight-line interpolating path used in flow matching (Lipman et al., 2022) and rectified flow models (Liu et al., 2022).

We now state the main result for the linear bridge as a corollary, and then provide the detailed derivation of the explicit constants.

**Corollary B.6** (Explicit constants for the linear bridge $\gamma(t) = t$). *Consider the closed-form flow matching model with the linear bridge $\gamma(t) = t$. Fix $t \in (0, 1)$ and suppose there exists $i^*$ and $\varepsilon \in (0, 1/2)$ such that $\lambda_{i^*}(z, t) \geq 1 - \varepsilon$. Then the instantaneous energy and the negative log-mixture-density satisfy*

$$\frac{1}{2}\left(-\log \hat{p}_t(z)\right) - C'_t \;\leq\; \|\hat{u}^*(z,t)\|^2 \;\leq\; 12\left(-\log \hat{p}_t(z)\right) + C'_t,$$

*for some $C'_t \in \mathbb{R}$ depending only on $t$, $\varepsilon$, the dimension $d$, the sample size $N$, and the data $\{x^{(i)}\}$ (through $\mu_{i^*}(t)$ and the spread of the means).*

*In particular, the dominant quadratic term of $-\log \hat{p}_t(z)$ is $\frac{1}{2(1-t)^2}\|z - \mu_{i^*}(t)\|^2$, whereas the dominant quadratic term of the instantaneous energy is $\frac{1}{(1-t)^2}\|z\|^2$.*

*For fixed $t$, the two leading quadratic forms differ asymptotically only by a factor of 2 as $\|z\| \to \infty$, with all remaining lower-order terms absorbed into $C'_t$. Consequently,*

$$\|\hat{u}^*(z,t)\|^2 \asymp -\log \hat{p}_t(z)$$

*up to explicit multiplicative constants and a $t$-dependent additive constant.*

**Derivation of explicit constants for $\gamma(t) = t$.** We now derive the explicit expressions claimed in Corollary B.6 by specializing the general results to the linear bridge $\gamma(t) = t$. Throughout we write

$$\sigma_t^2 = (1-t)^2, \qquad \mu_i(t) = t\,x^{(i)}, \qquad p_t(z \mid x^{(i)}) = \mathcal{N}(z; \mu_i(t), \sigma_t^2 I_d).$$

Lemma B.2 states that the optimal velocity is:

$$\hat{u}^*(z,t) = \alpha(t)\,\nabla_z \log \hat{p}_t(z) + \beta(t)\,z, \tag{56}$$

where

$$\alpha(t) = \frac{\dot{\gamma}(t)\sigma_t^2}{\gamma(t)(1-\gamma(t))}, \qquad \beta(t) = \frac{\dot{\gamma}(t)}{\gamma(t)}. \tag{57}$$

For the linear bridge $\gamma(t) = t$ and $\dot{\gamma}(t) = 1$, we obtain

$$\alpha(t) = \frac{1-t}{t}, \qquad \beta(t) = \frac{1}{t}, \tag{58}$$

and hence

$$\hat{u}^*(z,t) = \frac{1-t}{t}\,\nabla_z \log \hat{p}_t(z) + \frac{1}{t}\,z. \tag{59}$$

In this case, $\sigma_t^2 = (1-t)^2$ and

$$m(t) = \beta(t) - \frac{\alpha(t)}{\sigma_t^2} = \frac{1}{t} - \frac{1-t}{t}\cdot\frac{1}{(1-t)^2} = -\frac{1}{1-t}.$$

Assume there exists an index $i^*$ and $\varepsilon \in (0, 1/2)$ such that $\lambda_{i^*}(z,t) \geq 1 - \varepsilon$. Then Lemma B.3 gives

$$-\log \hat{p}_t(z) = \frac{1}{2(1-t)^2}\left\|z - \mu_{i^*}(t)\right\|^2 + C_t^{(0)} + R_t(z), \tag{60}$$

where $C_t^{(0)}$ depends only on $t$, $d$ and $N$, and $\log(1 - \varepsilon) \leq R_t(z) \leq 0$.

Similarly, Lemma B.4 yields

$$\nabla_z \log \hat{p}_t(z) = -\frac{1}{(1-t)^2}\big(z - \mu_{i^*}(t)\big) + r_t(z), \tag{61}$$

with

$$\|r_t(z)\| \leq \frac{\varepsilon}{(1-t)^2}\,\Delta_t, \qquad \Delta_t := \max_j \big\|tx^{(j)} - tx^{(i^*)}\big\|.$$

Substituting (61) into (59) and simplifying gives the affine representation

$$\hat{u}^*(z,t) = -\frac{1}{1-t}\,z + \frac{1}{t(1-t)}\,\mu_{i^*}(t) + \tilde{r}_t(z), \tag{62}$$

where $\tilde{r}_t(z)$ is uniformly bounded in $z$ for fixed $t$, $\varepsilon$ and data $\{x^{(i)}\}$ (as in Lemma B.5).

Ignoring the bounded remainder $\tilde{r}_t(z)$, the leading quadratic term of the instantaneous energy is

$$B_t(z) := \frac{1}{(1-t)^2}\,\|z\|^2. \tag{63}$$

On the other hand, from (60) the leading quadratic term of $-\log\hat{p}_t(z)$ is

$$A_t(z) := \frac{1}{2(1-t)^2}\,\big\|z - \mu_{i^*}(t)\big\|^2. \tag{64}$$

Thus, at the level of leading quadratic forms,

$$B_t(z) = 2A_t(z) + \text{(lower-order terms)}.$$

Specializing Lemma B.5 to $\gamma(t) = t$, we have $m(t) = -1/(1-t)$ and $\sigma_t^2 = (1-t)^2$, and hence

$$c_1(t) = \frac{1}{2}\,m(t)^2\,\sigma_t^2 = \frac{1}{2}, \qquad c_2(t) = 12\,m(t)^2\,\sigma_t^2 = 12.$$

Combining this with Theorem B.1, we obtain the bounds stated in Corollary B.6.

## C. Derivation of the Closed-Form Velocity Field in EFM

In this section, we provide a detailed derivation of the closed-form empirical velocity field $\hat{u}^\star(x,t)$ used in Section 5. The derivation is analogous to the one in (Bertrand et al., 2025) but we provide the details here for completeness.

### C.1. Setup and notation

As before, we follow the conditional flow matching setup with a Gaussian bridge.

Let $p_0 = \mathcal{N}(0, I_d)$ and the empirical data distribution $\hat{p}_{\text{data}}(x) = \frac{1}{N}\sum_{i=1}^N \delta_{x^{(i)}}(x)$. We consider the conditional bridge (for $t \in [0,1]$) defined by

$$p(x \mid z = x^{(i)}, t) = \mathcal{N}\big(x; tx^{(i)}, (1-t)^2 I_d\big). \tag{65}$$

For this bridge, the associated conditional velocity field is:

$$u_{\text{cond}}(x, t; z = x^{(i)}) = \frac{x^{(i)} - x}{1 - t}. \tag{66}$$

The (empirical) optimal velocity field is the conditional expectation

$$\hat{u}^\star(x, t) = \mathbb{E}[u_{\text{cond}}(x, t; z) \mid x, t] = \sum_{i=1}^N u_{\text{cond}}(x, t; z = x^{(i)})\,\hat{p}(z = x^{(i)} \mid x, t). \tag{67}$$

Thus, it remains to compute the posterior responsibilities $\hat{p}(z = x^{(i)} \mid x, t)$.

## C.2. Case I: $z \sim \hat{p}_{\text{data}}$ (discrete Bayes posterior)

Assume $z$ is distributed as $\hat{p}_{\text{data}}$, i.e., $z$ takes values in $\{x^{(1)}, \ldots, x^{(N)}\}$ with $\hat{p}(z = x^{(i)}) = 1/N$.

**Step 1: Bayes rule on the discrete support.** For each $i \in \{1, \ldots, N\}$,

$$\hat{p}(z = x^{(i)} \mid x, t) = \frac{\hat{p}(x, t, z = x^{(i)})}{\hat{p}(x, t)}. \tag{68}$$

Factor the joint as

$$\hat{p}(x, t, z = x^{(i)}) = \hat{p}(x \mid t, z = x^{(i)}) \, \hat{p}(t, z = x^{(i)}). \tag{69}$$

Summing over the discrete support gives

$$\hat{p}(x, t) = \sum_{j=1}^{N} \hat{p}(x, t, z = x^{(j)}) = \sum_{j=1}^{N} \hat{p}(x \mid t, z = x^{(j)}) \, \hat{p}(t, z = x^{(j)}). \tag{70}$$

Since $t$ is independent of $z$ and $\hat{p}(z = x^{(i)}) = 1/N$, we have

$$\hat{p}(t, z = x^{(i)}) = \hat{p}(t)\hat{p}(z = x^{(i)}) = \hat{p}(t) \cdot \frac{1}{N}. \tag{71}$$

Plugging (69)–(71) into (68), the factors $\hat{p}(t)$ and $1/N$ cancel, yielding

$$\hat{p}(z = x^{(i)} \mid x, t) = \frac{\hat{p}(x \mid t, z = x^{(i)})}{\sum_{j=1}^{N} \hat{p}(x \mid t, z = x^{(j)})}. \tag{72}$$

**Step 2: Plug in the Gaussian likelihood and simplify.** From (65),

$$\hat{p}(x \mid t, z = x^{(i)}) = \frac{1}{(2\pi)^{d/2}(1 - t)^d} \exp\left(-\frac{\|x - tx^{(i)}\|_2^2}{2(1 - t)^2}\right). \tag{73}$$

In (72), the prefactor $(2\pi)^{-d/2}(1 - t)^{-d}$ cancels across $i$, so

$$\hat{p}(z = x^{(i)} \mid x, t) = \frac{\exp\left(-\frac{\|x - tx^{(i)}\|_2^2}{2(1-t)^2}\right)}{\sum_{j=1}^{N} \exp\left(-\frac{\|x - tx^{(j)}\|_2^2}{2(1-t)^2}\right)}. \tag{74}$$

Define $\lambda_i(x, t) := \hat{p}(z = x^{(i)} \mid x, t)$; equivalently,

$$\lambda(x, t) = \text{softmax}\left(-\frac{\|x - tx^{(1)}\|_2^2}{2(1 - t)^2}, \ldots, -\frac{\|x - tx^{(N)}\|_2^2}{2(1 - t)^2}\right) \in \mathbb{R}^N. \tag{75}$$

**Step 3: Closed-form velocity.** Substituting (66) and (74) into (67), we obtain the closed-form empirical velocity field

$$\boxed{\hat{u}^{\star}(x, t) = \sum_{i=1}^{N} \lambda_i(x, t) \frac{x^{(i)} - x}{1 - t}.} \tag{76}$$

## C.3. Case II: $z \sim p_0 \times \hat{p}_{\text{data}}$ (Dirac constraint derivation)

We now show that the same softmax weights arise when the conditioning variable includes the source draw. Let $z = (x_0, x_1)$ with $x_0 \sim p_0$ and $x_1 \sim \hat{p}_{\text{data}}$ independent, and define the deterministic interpolation

$$x = (1 - t)x_0 + tx_1. \tag{77}$$

**Step 1: Deterministic conditional law.** Conditioned on $(x_0, x_1, t)$, $x$ is deterministic. Hence its conditional law is the Dirac measure concentrated at $(1 - t)x_0 + tx_1$:

$$p(x \mid t, x_0, x_1) = \delta(x - (1 - t)x_0 - tx_1). \tag{78}$$

**Step 2: Marginalize $x_0$ to obtain $p(x \mid t, x_1)$.** Fix $x_1 = x^{(i)}$. Then

$$p(x \mid t, x_1 = x^{(i)}) = \int_{\mathbb{R}^d} \delta(x - (1 - t)x_0 - tx^{(i)}) \, p_0(x_0) \, dx_0. \tag{79}$$

Using the scaling identity in $\mathbb{R}^d$, $\delta(Ay) = |\det A|^{-1}\delta(y)$, with $A = (1 - t)I_d$, we rewrite:

$$\delta(x - (1 - t)x_0 - tx^{(i)}) = \frac{1}{(1 - t)^d} \delta\left(x_0 - \frac{x - tx^{(i)}}{1 - t}\right). \tag{80}$$

Plugging (80) into (79) yields

$$p(x \mid t, x_1 = x^{(i)}) = \frac{1}{(1 - t)^d} p_0\left(\frac{x - tx^{(i)}}{1 - t}\right). \tag{81}$$

For $p_0 = \mathcal{N}(0, I_d)$,

$$p_0(y) = \frac{1}{(2\pi)^{d/2}} \exp\left(-\frac{\|y\|_2^2}{2}\right), \tag{82}$$

so (81) becomes

$$p(x \mid t, x_1 = x^{(i)}) = \frac{1}{(2\pi)^{d/2}(1 - t)^d} \exp\left(-\frac{\|x - tx^{(i)}\|_2^2}{2(1 - t)^2}\right), \tag{83}$$

which matches the Gaussian likelihood in (73).

**Step 3: Posterior over the discrete index $i$.** Since $x_1 \sim \hat{p}_{\text{data}}$ is uniform over the $N$ atoms, Bayes rule gives

$$\hat{p}(x_1 = x^{(i)} \mid x, t) = \frac{p(x \mid t, x_1 = x^{(i)})}{\sum_{j=1}^{N} p(x \mid t, x_1 = x^{(j)})}. \tag{84}$$

Plugging (83) into (84) cancels the same prefactors and recovers the softmax form (74).

**Step 4: Closed-form velocity.** For the deterministic interpolation (77),

$$u_{\text{cond}}(x, t; x_0, x_1) = x_1 - x_0, \qquad x_0 = \frac{x - tx_1}{1 - t} \implies u_{\text{cond}}(x, t; x_1) = \frac{x_1 - x}{1 - t}. \tag{85}$$

Taking the conditional expectation over $x_1 \mid x, t$ yields the same closed-form formula (76).

### C.4. Summary

In both cases ($z \sim \hat{p}_{\text{data}}$ or $z \sim p_0 \times \hat{p}_{\text{data}}$ with independent coupling), the empirical optimal velocity field admits the closed-form formula:

$$\hat{u}^\star(x, t) = \sum_{i=1}^{N} \lambda_i(x, t) \frac{x^{(i)} - x}{1 - t}, \qquad \lambda_i(x, t) \propto \exp\left(-\frac{\|x - tx^{(i)}\|_2^2}{2(1 - t)^2}\right),$$

where the proportionality constant is the normalization across $i \in \{1, \ldots, N\}$.

## D. Proofs for Terminal-Time Energy Blow-Up

In this section, we provide detailed versions of Proposition 5.2 and Lemma 5.1, together with their proofs.

**Setting.** Let $\{x^{(i)}\}_{i=1}^N \subset \mathbb{R}^d$ be a given training set, and define the closed-form empirical velocity field

$$\hat{u}^\star(x, t) = \sum_{i=1}^N \lambda_i(x, t) \frac{x^{(i)} - x}{1 - t}, \qquad \sum_{i=1}^N \lambda_i(x, t) = 1, \qquad \lambda_i(x, t) \geq 0, \tag{86}$$

with softmax weights

$$\lambda_i(x, t) = \frac{\exp\left(-\frac{\|x - tx^{(i)}\|^2}{2(1-t)^2}\right)}{\sum_{j=1}^N \exp\left(-\frac{\|x - tx^{(j)}\|^2}{2(1-t)^2}\right)}. \tag{87}$$

For a trajectory $x(\cdot) := (x(t))_{t \in [0,1)}$, we write $\lambda_i(t) := \lambda_i(x(t), t)$ for brevity.

### D.1. A softmax concentration lemma

**Lemma D.1** (Terminal-time posterior concentration under a margin condition). *Fix a trajectory $x(\cdot)$ and define the bridge scores $s_i(t) := \|x(t) - tx^{(i)}\|^2$. Assume there exist constants $t_0 \in [0, 1)$, $m > 0$, and an index $i^\star \in \{1, \ldots, N\}$ such that for all $t \in [t_0, 1)$ and all $j \neq i^\star$,*

$$s_j(t) - s_{i^\star}(t) \geq m. \tag{88}$$

*Then for all $t \in [t_0, 1)$,*

$$1 - \lambda_{i^\star}(t) \leq (N - 1) \exp\left(-\frac{m}{2(1-t)^2}\right). \tag{89}$$

*Proof.* By (87),

$$\lambda_{i^\star}(t) = \frac{1}{1 + \sum_{j \neq i^\star} \exp\left(-\frac{s_j(t) - s_{i^\star}(t)}{2(1-t)^2}\right)}.$$

Thus, using (88), we have:

$$1 - \lambda_{i^\star}(t) = \frac{\sum_{j \neq i^\star} \exp\left(-\frac{s_j(t) - s_{i^\star}(t)}{2(1-t)^2}\right)}{1 + \sum_{j \neq i^\star} \exp\left(-\frac{s_j(t) - s_{i^\star}(t)}{2(1-t)^2}\right)} \leq \sum_{j \neq i^\star} \exp\left(-\frac{m}{2(1-t)^2}\right),$$

which yields (89). $\qquad \square$

### D.2. A Detailed Lemma on Terminal Energy Blow-Up

The following is a detailed version of Lemma 5.1 from the main paper.

**Lemma D.2.** *Consider a trajectory segment $t \in [1 - \varepsilon, 1)$ such that:*

(i) *(non-collision) there is a constant $c > 0$ such that $\min_i \|x(t) - x^{(i)}\|_2 \geq c$ for all $t \in [1 - \varepsilon, 1)$;*

(ii) *(bounded geometry) $\max_i \|x^{(i)}\|_2 \leq R$ for some constant $R > 0$;*

(iii) *(terminal posterior concentration) there exist $t_0 \in [1 - \varepsilon, 1)$ and $i^\star$ such that the margin condition (88) holds on $[t_0, 1)$ for some $m > 0$.*

*Then there exists $\bar{t} \in [t_0, 1)$ such that for all $t \in [\bar{t}, 1)$,*

$$\|\hat{u}^\star(x(t), t)\|_2 \geq \frac{c}{2(1-t)}. \tag{90}$$

*Consequently, the terminal contribution to KPE diverges as an improper integral:*

$$\int_{\bar{t}}^1 \|\hat{u}^\star(x(t), t)\|_2^2 \, dt = +\infty. \tag{91}$$

*Proof of Lemma D.2.* Since $\sum_{i=1}^{N} \lambda_i(t) = 1$ for $t \in [0,1)$, we have:

$$\sum_{i=1}^{N} \lambda_i(t) \left(x^{(i)} - x(t)\right) = \lambda_{i^\star}(t)(x^{(i^\star)} - x(t)) + \sum_{j \neq i^\star} \lambda_j(t)(x^{(j)} - x(t))$$

$$= \lambda_{i^\star}(t)(x^{(i^\star)} - x(t)) + \sum_{j \neq i^\star} \lambda_j(t) \left[(x^{(j)} - x^{(i^\star)}) + (x^{(i^\star)} - x(t))\right]$$

$$= \left(\lambda_{i^\star}(t) + \sum_{j \neq i^\star} \lambda_j(t)\right)(x^{(i^\star)} - x(t)) + \sum_{j \neq i^\star} \lambda_j(t)(x^{(j)} - x^{(i^\star)})$$

$$= (x^{(i^\star)} - x(t)) + \sum_{j \neq i^\star} \lambda_j(t)(x^{(j)} - x^{(i^\star)}).$$

Taking norms and applying the reverse triangle inequality, followed by the triangle inequality, we obtain:

$$\left\|\sum_{i=1}^{N} \lambda_i(t) \left(x^{(i)} - x(t)\right)\right\| \geq \|x^{(i^\star)} - x(t)\| - \sum_{j \neq i^\star} \lambda_j(t) \|x^{(j)} - x^{(i^\star)}\|$$

$$\geq c - \sum_{j \neq i^\star} \lambda_j(t) \left(\|x^{(j)}\| + \|x^{(i^\star)}\|\right)$$

$$\geq c - 2R \sum_{j \neq i^\star} \lambda_j(t) = c - 2R \left(1 - \lambda_{i^\star}(t)\right).$$

By Lemma D.1, $1 - \lambda_{i^\star}(t) \to 0$ as $t \to 1$ and in fact satisfies (89). Hence we can choose $\bar{t} \in [t_0, 1)$ such that for all $t \in [\bar{t}, 1)$ we have $2R(1 - \lambda_{i^\star}(t)) \leq c/2$. Then

$$\left\|\sum_{i=1}^{N} \lambda_i(t) \left(x^{(i)} - x(t)\right)\right\| \geq c/2.$$

Plugging this into (86) gives (90).

Finally,

$$\int_{\bar{t}}^{1} \|\hat{u}^\star(x(t), t)\|_2^2 \, dt \geq \frac{c^2}{4} \int_{\bar{t}}^{1} (1-t)^{-2} \, dt.$$

Interpreting the integral on the right-hand side as an improper integral,

$$\int_{\bar{t}}^{1} (1-t)^{-2} \, dt := \lim_{\delta \to 0^+} \int_{\bar{t}}^{1-\delta} (1-t)^{-2} \, dt = \lim_{\delta \to 0^+} \left(\frac{1}{\delta} - \frac{1}{1-\bar{t}}\right) = \infty,$$

which proves (91). □

### D.3. A Detailed Version of Proposition 5.2

The following is a detailed version of Proposition 5.2 from the main paper.

**Proposition D.3** (Terminal-time kinetic energy blow-up and a universal lower bound). *Let $\{x^{(i)}\}_{i=1}^{N} \subset \mathbb{R}^d$ and define $\hat{u}^\star$ and $\lambda_i$ as in (86)–(87).*

(a) ***Blow-up for the EFM field under non-collision and posterior concentration.*** *Let $x : [0,1) \to \mathbb{R}^d$ be a trajectory segment on $[1 - \varepsilon, 1)$ satisfying assumptions (i)–(iii) of Lemma D.2. Then there exists $\bar{t} \in [1 - \varepsilon, 1)$ such that*

$$\int_{\bar{t}}^{1} \|\hat{u}^\star(x(t), t)\|_2^2 \, dt = +\infty \qquad \text{(improper integral)}.$$

(b) ***Universal lower bound for terminal closure.*** *Let $x : [0,1] \to \mathbb{R}^d$ be absolutely continuous and assume $x(1) = x^{(i)}$ for some $i$. Then for every $t < 1$,*

$$\int_{t}^{1} \|\dot{x}(s)\|_2^2 \, ds \geq \frac{\|x^{(i)} - x(t)\|_2^2}{1 - t}. \tag{92}$$

*Proof of Proposition D.3.* **Part (a).** This is an immediate corollary of Lemma D.2. More precisely, under the assumptions of Lemma D.2, we have $\|\hat{u}^\star(x(t), t)\| \geq \frac{c}{2(1-t)}$ for all $t$ sufficiently close to 1 and therefore the terminal kinetic energy diverges by (91).

**Part (b).** Let $x : [0, 1] \to \mathbb{R}^d$ be absolutely continuous with $x(1) = x^{(i)}$ for some $i$. For any $t < 1$, by Cauchy–Schwarz,

$$\|x^{(i)} - x(t)\|_2^2 = \left\| \int_t^1 \dot{x}(s)\, ds \right\|_2^2 \leq (1 - t) \int_t^1 \|\dot{x}(s)\|_2^2\, ds. \tag{93}$$

Thus,

$$\int_t^1 \|\dot{x}(s)\|_2^2\, ds \ \geq\ \frac{\|x^{(i)} - x(t)\|_2^2}{1 - t}. \tag{94}$$

This proves part (b).

$\square$

# E. Experimental Setup Details

In this section, we provide details on the experimental configurations for the findings presented in §4.

### E.1. Finding 1: Semantic Quality Experiments

**Model and Dataset.** We use the pretrained SiT-XL/2 flow matching model (Ma et al., 2024), a class-conditional transformer-based flow matching model trained on ImageNet-256 at $256 \times 256$ resolution.

**CLIP score** measures semantic alignment as $100\times$ the maximum cosine similarity between normalized image features and the true-class text features. **CLIP margin** measures semantic discriminability as the gap between the true-class similarity and the best competing-class similarity: $\text{Margin} = \text{Sim}_{\text{true}} - \max_{c \in \mathcal{C}_{\text{others}}} \text{Sim}(c)$, where $\mathcal{C}_{\text{others}}$ denotes competing classes. Higher margins imply stronger class-specific semantics.

**Sampling Configuration.**

- **ODE solver**: Forward Euler integration with $\text{NFE} = 10$

- **CFG scales**: $\omega \in \{1.0, 1.5, 4.0\}$ interpolating between unconditional and class-conditional generation

- **Sample size**: 4,000 samples per CFG (12,000 in total)

- **Random seeds**: Each sample is generated from independent Gaussian noise $z_0 \sim \mathcal{N}(0, I)$ with distinct random seed

- **Class selection**: Uniformly sampled from 1,000 ImageNet classes

**KPE Computation.** For each trajectory $(z(t))_{t \in [0,1]}$, we compute kinetic path energy via discrete approximation:

$$E = \frac{1}{2} \sum_{i=0}^{\text{NFE}-1} \|v_\theta(z(t_i), t_i)\|^2 \cdot \Delta t, \quad \Delta t = 1/\text{NFE}. \tag{95}$$

**Energy Stratification.** We partition samples into three groups based on KPE percentiles:

- **Low energy**: 0–33% percentile ($n = 1,333$ per energy group, 4,000 in total)

- **Mid energy**: 33–67% percentile ($n = 1,333$ per energy group, 4,000 in total)

- **High energy**: 67–100% percentile ($n = 1,333$ per energy group, in 4,000 total)

**CLIP Evaluation.**

- **Model**: CLIP ViT-L/14 with frozen weights

- **CLIP Score**: $\text{Score} = 100 \times \max_{c \in \mathcal{C}} \text{CosineSim}(\text{Embed}_{\text{img}}, \text{Embed}_{\text{text}}(c))$ where text prompt is "a photo of a [class]"

- **CLIP Margin**: $\text{Margin} = \text{Sim}_{\text{true}} - \max_{c \in \mathcal{C} \setminus \{\text{true}\}} \text{Sim}(c)$ measuring discriminability

**Statistical Testing.** Independent two-sample $t$-tests comparing low vs. high energy groups. Bonferroni correction applied for 6 comparisons (2 metrics × 3 CFG scales): corrected $\alpha = 0.05/6 \approx 0.008$. Cohen's $d$ for effect size.

# F. Detailed Synthetic Dataset Specifications

In this section, we provide detailed specifications for the synthetic 2D datasets used to validate the KPE-density relationship under controlled conditions (§4.2).

### F.1. Dataset Descriptions

We design three synthetic 2D datasets with explicitly controlled density stratification:

### 1. Dense Core + Sparse Ring (`dense_sparse`).

- **Dense core**: 60% of samples from $\mathcal{N}(\mathbf{0}, \sigma_{\text{core}}^2 I)$ with $\sigma_{\text{core}} = 0.15$

- **Sparse ring**: 40% of samples uniformly distributed on annulus with radius $r \in [2.3, 2.7]$, perturbed by $\mathcal{N}(0, \sigma_{\text{ring}}^2 I)$ with $\sigma_{\text{ring}} = 0.5$

- **Density ratio**: Core density $\approx 15\times$ higher than ring density

### 2. Multiscale Clusters (`multiscale_clusters`).

- **Sparse center**: 20% from $\mathcal{N}(\mathbf{0}, 0.6^2 I)$

- **Dense peripheral clusters**: 20% each from $\mathcal{N}(\mathbf{c}_i, 0.08^2 I)$ for $i \in \{1, 2, 3, 4\}$

- **Cluster centers**: $\mathbf{c}_1 = (2, 0)$, $\mathbf{c}_2 = (0, 2)$, $\mathbf{c}_3 = (-2, 0)$, $\mathbf{c}_4 = (0, -2)$

- **Density ratio**: Peripheral clusters $\approx 50\times$ denser than center

### 3. Sandwich (`sandwich`).

- **Dense middle band**: 60% from uniform $x \in [-3, 3]$, $y \in [-0.3, 0.3]$ plus $\mathcal{N}(0, 0.1^2 I)$

- **Sparse top band**: 20% from uniform $x \in [-3, 3]$, $y \in [1.5, 2.5]$ plus $\mathcal{N}(0, 0.3^2 I)$

- **Sparse bottom band**: 20% from uniform $x \in [-3, 3]$, $y \in [-2.5, -1.5]$ plus $\mathcal{N}(0, 0.3^2 I)$

- **Density ratio**: Middle band $\approx 10\times$ denser than outer bands

### F.2. Training Details

For each synthetic dataset, we train a standard flow matching model with the following configuration:

### Model Architecture.

- **Network**: 4-layer MLP with hidden dimensions [128, 256, 256, 128]

- **Activation**: SiLU (Swish) activation functions

- **Input**: Concatenation of $[z, t]$ where $z \in \mathbb{R}^2$ and $t \in [0, 1]$

- **Output**: Velocity field $v_\theta(z, t) \in \mathbb{R}^2$

- **Time encoding**: Sinusoidal positional encoding for $t$ (16 dimensions)

### Training Hyperparameters.

- **Optimizer**: AdamW with learning rate $3 \times 10^{-4}$, weight decay $10^{-4}$

- **Batch size**: 256

- **Training steps**: 50,000 iterations

- **Loss**: Standard flow matching loss $\mathcal{L} = \mathbb{E}_{t, x_0, x_1} [\|v_\theta(z_t, t) - (x_1 - z_t)/(1 - t)\|]^2$

- **Training data**: $N = 1{,}000$ samples per dataset

### Sampling and Evaluation.

- **ODE solver**: Forward Euler with $\text{NFE} = 100$

- **Test samples**: $M = 500$ trajectories per dataset

- **Density estimation**: Ground-truth KDE with Gaussian kernel, bandwidth $h = 0.1$

- **KPE computation**: $E = \frac{1}{2} \sum_{i=0}^{\text{NFE}-1} \|v_\theta(z(t_i), t_i)\|^2 \cdot \Delta t$

## G. KPE Stability Across Discretization Choices

In this section, we verify that the per-sample KPE diagnostic is stable across solver and NFE choices.

We fix 200 initial noise vectors (seed $= 42$) and a single CelebA checkpoint (training step 30,000, in the memorization regime), then re-integrate each trajectory under six ODE configurations: solvers $\in \{$Euler, Heun$\}$ and NFE $\in \{50, 100, 200\}$. For each configuration we compute the per-sample KPE and report the Spearman rank correlation between pairs of configurations.

Table 9 shows that all $\binom{6}{2} = 15$ pairwise Spearman rank correlations are $\geq 0.992$. The most adversarial pair (Euler 50 vs. Heun 200) still yields $\rho = 0.992$. Hence the KPE-based ranking of samples is essentially invariant to the discretization choice, and the qualitative groupings (low/mid/high-KPE bins) used throughout the paper transfer across solvers and NFE.

*Table 9.* **Pairwise Spearman $\rho$ of per-sample KPE rankings across six ODE configurations (CelebA, 200 fixed noise vectors).** All pairwise $\rho \geq 0.992$, confirming that the KPE diagnostic is essentially insensitive to the choice of solver and NFE.

|           | Euler 50 | Euler 100 | Euler 200 | Heun 50 | Heun 100 | Heun 200 |
|-----------|----------|-----------|-----------|---------|----------|----------|
| Euler 50  | 1.0000   | 0.9976    | 0.9949    | 0.9926  | 0.9924   | 0.9921   |
| Euler 100 | 0.9976   | 1.0000    | 0.9991    | 0.9975  | 0.9976   | 0.9974   |
| Euler 200 | 0.9949   | 0.9991    | 1.0000    | 0.9990  | 0.9992   | 0.9992   |
| Heun 50   | 0.9926   | 0.9975    | 0.9990    | 1.0000  | 0.9998   | 0.9997   |
| Heun 100  | 0.9924   | 0.9976    | 0.9992    | 0.9998  | 1.0000   | 0.9999   |
| Heun 200  | 0.9921   | 0.9974    | 0.9992    | 0.9997  | 0.9999   | 1.0000   |

## H. Sensitivity of the Local Support Estimator to $k$

Table 3 in the main text shows that the negative KPE–support correlation is robust across feature spaces. In this section, we complement that result with a sensitivity analysis of the $k$-NN neighborhood size $k$, the only free hyperparameter of the local support estimator used throughout §4.2.

We re-run the $k$-NN estimator on CIFAR-10 (descriptors feature space) for $k \in \{5, 10, 20, 50, 100\}$ at two extreme NFE settings (NFE $= 150$ and NFE $= 10$) and report Spearman $\rho$ and Cliff's $\delta$ between KPE and the resulting log-support. Table 10 shows that the negative correlation is stable across the entire range of $k$, with $\rho$ varying by at most $\pm 0.05$ around the $k = 50$ value used in the main text and $|\delta| \geq 0.70$ throughout. Hence the qualitative findings of §4.2 do not hinge on a particular choice of $k$.

*Table 10.* **$k$-NN $k$ sensitivity (CIFAR-10, Descriptors).** Spearman $\rho$ and Cliff's $\delta$ between KPE and $k$-NN log-support for $k \in \{5, 10, 20, 50, 100\}$ at NFE $\in \{10, 150\}$. The negative KPE–support relation is stable across $k$; the $k = 50$ row corresponds to the setting used in the main text.

|       | NFE $= 150$ | | NFE $= 10$ | |
|-------|--------------------|--------------------|--------------------|--------------------|
| $k$   | $\rho \downarrow$ | $\delta \downarrow$ | $\rho \downarrow$ | $\delta \downarrow$ |
| 5     | $-0.57$           | $-0.87$            | $-0.44$           | $-0.70$            |
| 10    | $-0.62$           | $-0.91$            | $-0.59$           | $-0.87$            |
| 20    | $-0.63$           | $-0.92$            | $-0.60$           | $-0.86$            |
| **50** | $\mathbf{-0.65}$ | $\mathbf{-0.93}$   | $\mathbf{-0.54}$  | $\mathbf{-0.83}$   |
| 100   | $-0.63$           | $-0.91$            | $-0.56$           | $-0.83$            |

## I. Stability Bound for Kinetic Trajectory Shaping

In this section, we provide a stability bound to make precise the sense in which Kinetic Trajectory Shaping (KTS) induces only a bounded shift in the generated samples, despite modifying the underlying ODE dynamics.

## I.1. Setup: KTS as time-dependent velocity rescaling

Recall the neural FM's ODE sampler and its KTS-modified counterpart:

$$\dot{x}(t) = v_\theta(x(t), t), \qquad \dot{\tilde{x}}(t) = \eta(t)\, v_\theta(\tilde{x}(t), t), \qquad x(0) = \tilde{x}(0) = x_0 \sim \mathcal{N}(0, I), \tag{96}$$

with the gain $\eta(t) > 0$ given in Eq. (13):

$$\eta(t) = \begin{cases} 1 + \alpha_0\,(1 - t/\tau_{\mathrm{split}}), & t < \tau_{\mathrm{split}}, \\ 1 - \beta_0\,\big(\exp(k(t - \tau_{\mathrm{split}})) - 1\big), & t \geq \tau_{\mathrm{split}}. \end{cases}$$

Because $\eta(t) > 0$ for the hyperparameters considered, the KTS's ODE preserves the direction of the velocity field; the only change is a time-dependent rescaling of its magnitude. We first collect two observations that will be useful later.

**(1) Small perturbation regime.** For the balanced setting $(\alpha_0, \beta_0) = (0.01, 0.01)$ used in our experiments, with $\tau_{\mathrm{split}} = 0.6$ and $k = 3$:

- In the launch phase, $|\eta(t) - 1| = \alpha(t) \leq \alpha_0 = 0.01$ for $t \in [0, \tau_{\mathrm{split}})$, and $\alpha(\tau_{\mathrm{split}}) = 0$.
- In the soft landing phase, $|\eta(t) - 1| = \beta(t)$ with maximum $\beta(1) = \beta_0\big(\exp(k(1 - \tau_{\mathrm{split}})) - 1\big) = 0.01\big(\exp(1.2) - 1\big) \approx 0.022$.

The deviation $|\eta(t) - 1|$ thus stays below $0.025$ over the entire trajectory. In particular, the gain schedule remains bounded and controlled.

**(2) Direction preservation.** Because the KTS gain only rescales the velocity by a positive scalar, KTS preserves direction-based structure of the learned field while modifying the speed at which trajectories move along it.

## I.2. A Grönwall-type stability bound

We now formalize how the small magnitude deviation $|\eta(t) - 1|$ propagates to a bound on the terminal sample shift $\|\tilde{x}(1) - x(1)\|$.

**Assumption I.1** (Uniform Lipschitz velocity). There exists a constant $L \geq 0$ such that, for all $t \in [0, 1]$ and all $x, y \in \mathbb{R}^d$,

$$\|v_\theta(x, t) - v_\theta(y, t)\| \leq L\, \|x - y\|. \tag{97}$$

This is a standard sufficient assumption for well-posedness of ODE samplers in diffusion and flow-based models. We do *not* require Lipschitz continuity in $t$, and the uniform spatial Lipschitz assumption above could be relaxed to a local Lipschitz condition at the cost of additional technicality.

**Proposition I.2** (Stability bound for KTS). *Let $x(t)$ and $\tilde{x}(t)$ be the base and KTS-modified trajectories in Eq. (96), with the same initial condition $x(0) = \tilde{x}(0)$. Under Assumption I.1, for every $t \in [0, 1]$,*

$$\|\tilde{x}(t) - x(t)\| \leq \exp\left(L \int_0^t |\eta(s)|\, ds\right) \int_0^t |\eta(s) - 1|\, \|v_\theta(x(s), s)\|\, ds. \tag{98}$$

*In particular, the terminal shift satisfies*

$$\|\tilde{x}(1) - x(1)\| \leq \exp\left(L \int_0^1 |\eta(s)|\, ds\right) \int_0^1 |\eta(s) - 1|\, \|v_\theta(x(s), s)\|\, ds. \tag{99}$$

*Proof.* Let $\Delta(t) := \tilde{x}(t) - x(t)$, so $\Delta(0) = 0$. From Eq. (96),

$$\dot{\Delta}(t) = \eta(t)\, v_\theta(\tilde{x}(t), t) - v_\theta(x(t), t)$$
$$= \eta(t)\, \big[v_\theta(\tilde{x}(t), t) - v_\theta(x(t), t)\big] + \big(\eta(t) - 1\big)\, v_\theta(x(t), t).$$

Taking norms and applying the triangle inequality together with Assumption I.1,

$$\frac{d}{dt}\, \|\Delta(t)\| \leq \|\dot{\Delta}(t)\| \leq L\, |\eta(t)|\, \|\Delta(t)\| + |\eta(t) - 1|\, \|v_\theta(x(t), t)\|, \tag{100}$$

where the left inequality follows from the standard fact that $\frac{d}{dt}\|\Delta(t)\| \le \|\dot{\Delta}(t)\|$ holds almost everywhere whenever $\Delta(\cdot)$ is absolutely continuous.

Eq. (100) is a scalar linear differential inequality of the form $\frac{d}{dt}y(t) \le a(t)\,y(t) + b(t)$ with $a(t) = L|\eta(t)|$ and $b(t) = |\eta(t) - 1|\,\|v_\theta(x(t), t)\|$. The standard Grönwall inequality then yields, for $y(0) = 0$,

$$
y(t) \;\le\; \int_0^t b(s)\,\exp\Big(\int_s^t a(r)\,dr\Big)\,ds \;\le\; \exp\Big(\int_0^t a(r)\,dr\Big)\int_0^t b(s)\,ds,
$$

which is exactly Eq. (98). $\qquad\square$

*Remark* I.3 (Interpreting the bound). Eq. (99) factorizes the terminal sample shift into:

- a *Lipschitz amplification factor* $\exp(L\int_0^1 |\eta(s)|\,ds)$, intrinsic to the base FM model;

- a *design-controlled deviation* $\int_0^1 |\eta(s) - 1|\,\|v_\theta(x(s), s)\|\,ds$, which we can make small by choosing $\alpha_0, \beta_0 \ll 1$.

Because $|\eta(s)| \le 1 + \max\{|\alpha(s)|, |\beta(s)|\} \le 1.025$ in our balanced setting, the amplification factor is close to that of the base FM ODE. The leading perturbation enters through the second integral, which scales linearly with $\alpha_0$ and $\beta_0$ for fixed base trajectory. Thus KTS induces a sample shift that vanishes as the hyperparameters tend to zero, and is bounded for the small values used in practice.

### I.3. Empirical confirmation

Proposition I.2 is a worst-case bound. In practice, the shift is far smaller and does not appear to substantially distort the generated distribution. Two observations support this:

**Magnitude rescaling stays below** $2.5\%$**.** Under the balanced setting, $|\eta(t) - 1|$ never exceeds $\approx 0.025$ along any trajectory. This bounded, dimensionless gain perturbation is qualitatively different from the unbounded terminal amplification in the closed-form EFM solution (cf. §5.2), so KTS operates in a small perturbation regime relative to the base neural FM sampler.

**Precision/Recall on ImageNet-256 are essentially unchanged.** If the KTS-induced shift caused substantive distributional change, it would show up in Precision/Recall, which separately track sample quality (precision) and coverage (recall). Table 5 (main text) reports:

- FM baseline: Precision 0.728, Recall 0.655;

- Balanced KTS ($\alpha_0 = \beta_0 = 0.01$): Precision 0.729, Recall 0.653;

- Quality-focused KTS ($\alpha_0 = 0.05, \beta_0 = 0$): Precision 0.731, Recall 0.630;

- Coverage-focused KTS ($\alpha_0 = 0, \beta_0 = 0.05$): Precision 0.721, Recall 0.657.

The balanced setting moves both metrics by at most 0.003, consistent with a small, distributionally benign perturbation. The directional asymmetry between quality-focused and coverage-focused KTS is also consistent with the intended interpretation of $\alpha_0$ (boost early kinetic effort $\to$ slightly higher precision) and $\beta_0$ (damp late kinetic spikes $\to$ slightly higher recall by mitigating terminal collapse toward training atoms).

### I.4. Comparison with inference-time noise injection

We compare KTS against the simplest training-free alternative, i.e., additive Gaussian noise injected into the velocity at each ODE step, under the same trained model and sampler. Table 11 shows that noise injection only marginally reduces $F_{\mathrm{mem}}$ and worsens FID, while balanced KTS attains both the best FID and the lowest $F_{\mathrm{mem}}$, confirming that the gain comes from energy-aware structure rather than generic stochastic perturbation.

*Table 11.* **KTS vs. inference-time noise injection (CelebA** $32 \times 32$, NFE $= 100$, **10K samples).** All methods use the same trained FM model; only the sampler is modified.

| Method | FID@10k $\downarrow$ | $F_{\text{mem}}$ (%) $\downarrow$ |
|---|---|---|
| FM baseline (Euler ODE) | 16.68 | 37.34 |
| + noise injection ($\sigma = 0.005$) | 17.48 | 35.31 |
| + noise injection ($\sigma = 0.01$) | 37.33 | 33.25 |
| **KTS** ($\alpha_0 = \beta_0 = 0.01$) | **14.35** | **31.22** |

# J. Additional Visualizations: Toy 2D Generation and Dynamics

In this section, we provide additional visualization results on the 2D synthetic datasets.

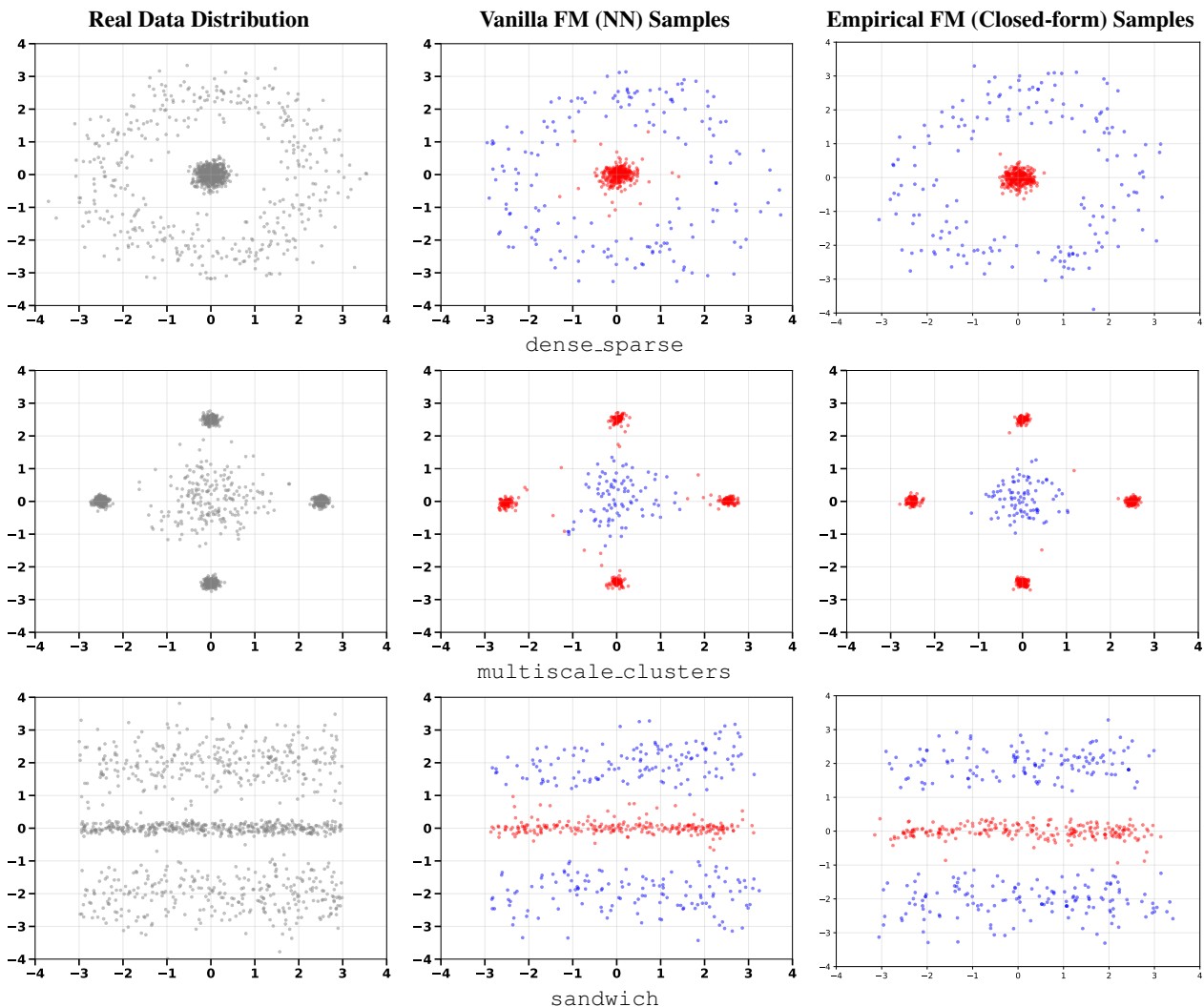

*Figure 10.* **Toy 2D generations: Real vs. Vanilla FM vs. Empirical FM.** For each dataset (row), we compare the target distribution (left) with samples generated by a neural vanilla FM (middle) and the empirical closed-form FM solution (right), using the same bridge family.

**Trajectories**                    **Velocity fields**

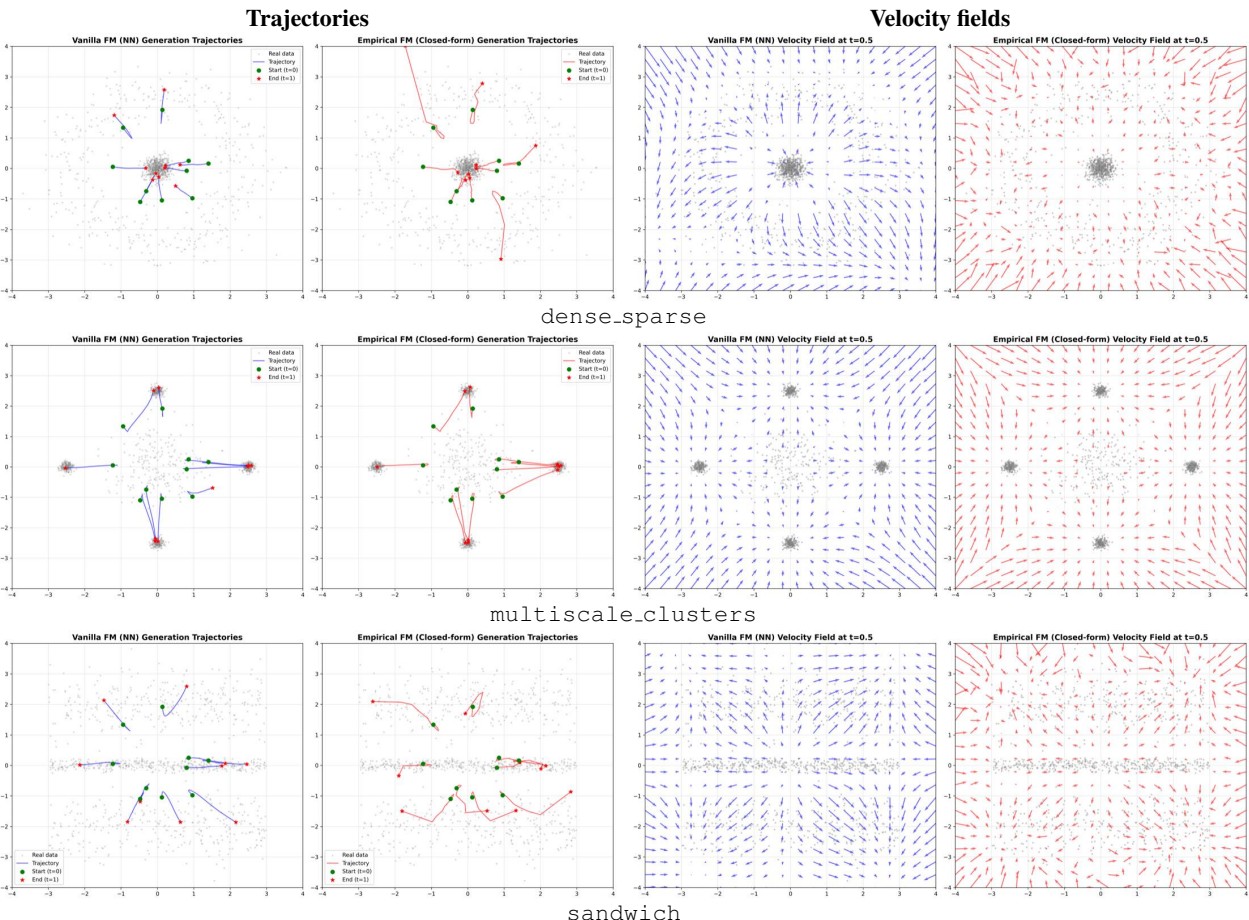

*Figure 11.* **Toy 2D dynamics: trajectories and velocity fields.** For each dataset (row), we visualize sampled trajectories under the learned/closed-form flows (left) and the corresponding velocity field structure (right). These dynamics complement the power/energy plots in the main text by showing *where* and *how* the flows move mass over time.

## K. Additional Visualizations of KPE vs. Semantic Strength

In this section, we provide qualitative visual comparisons across diverse ImageNet-256 classes to demonstrate the consistent semantic quality differences between high-energy and low-energy trajectories.

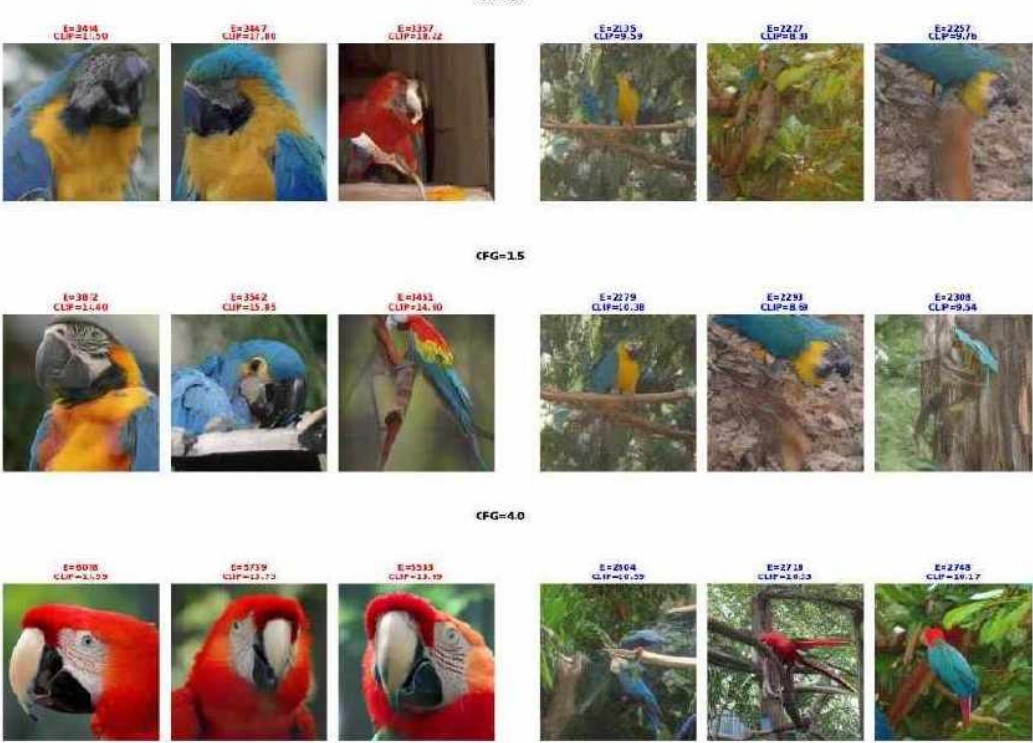

*Figure 12.* **Macaw** (ImageNet-256): High-KPE (left) vs. low-KPE (right) across CFG scales 1.0, 1.5, 4.0. Higher KPE yields richer semantic details, vibrant colors, and sharper textures.

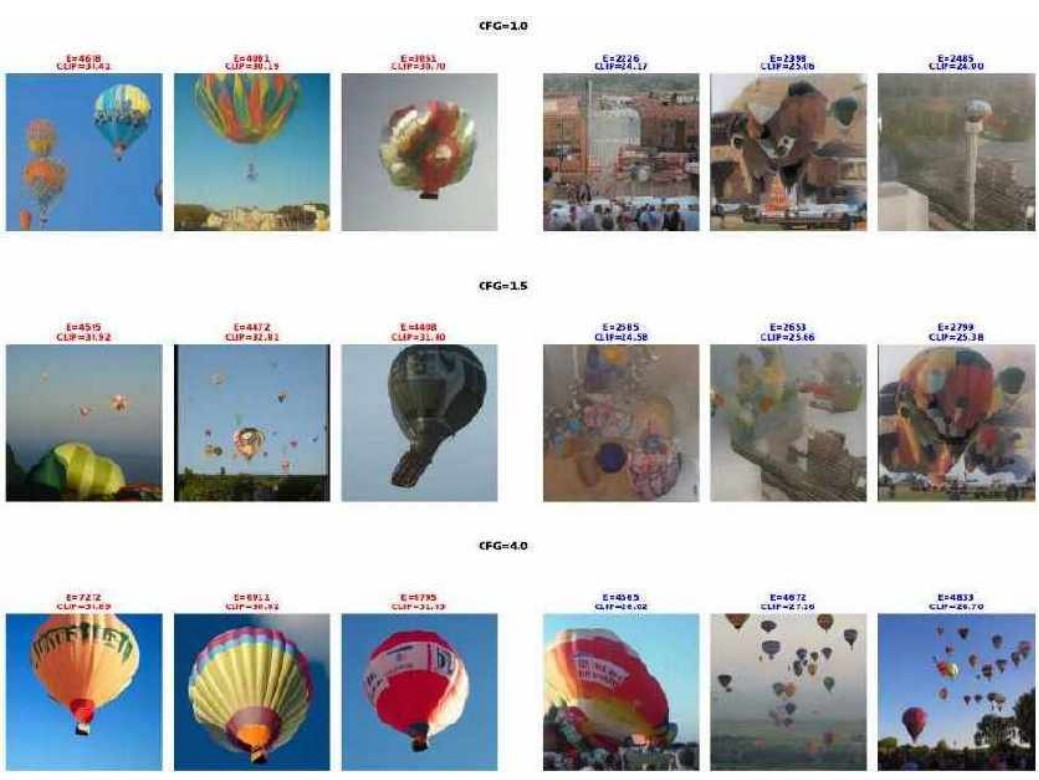

*Figure 13.* **Hot Air Balloon** (ImageNet-256): High-KPE (left) vs. low-KPE (right) across CFG scales 1.0, 1.5, 4.0. Higher KPE shows clearer structures and better color saturation.

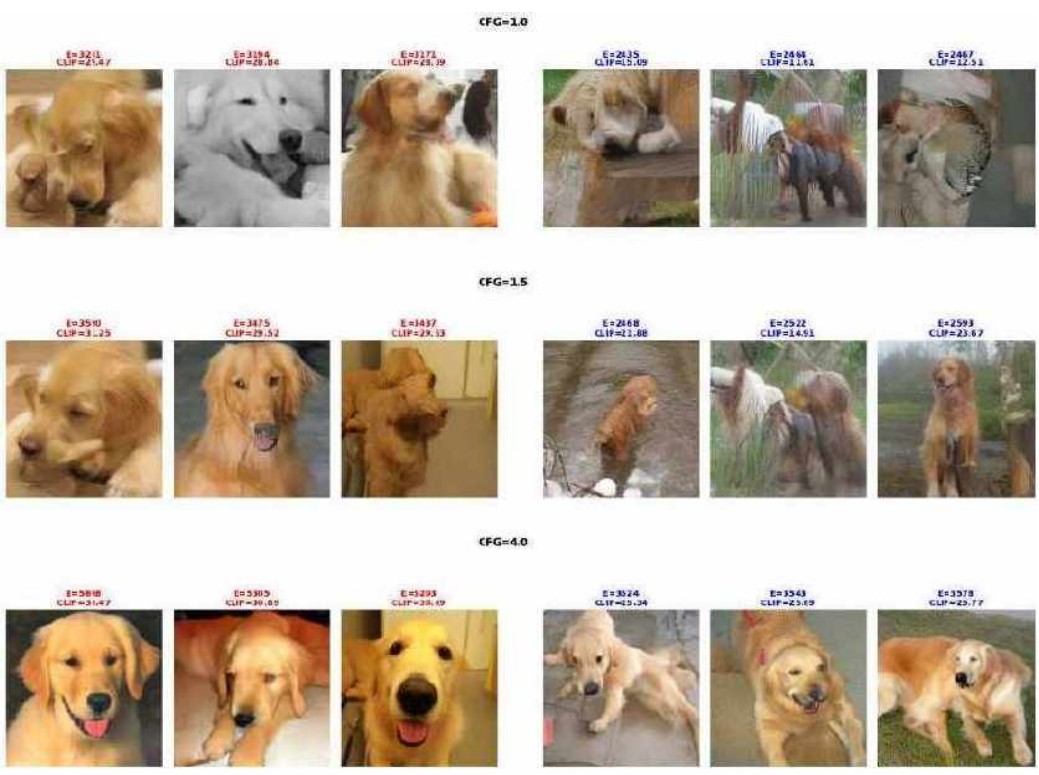

*Figure 14.* **Golden Retriever** (ImageNet-256): High-KPE (left) vs. low-KPE (right) across CFG scales 1.0, 1.5, 4.0. Higher KPE produces finer textures and clearer facial features.

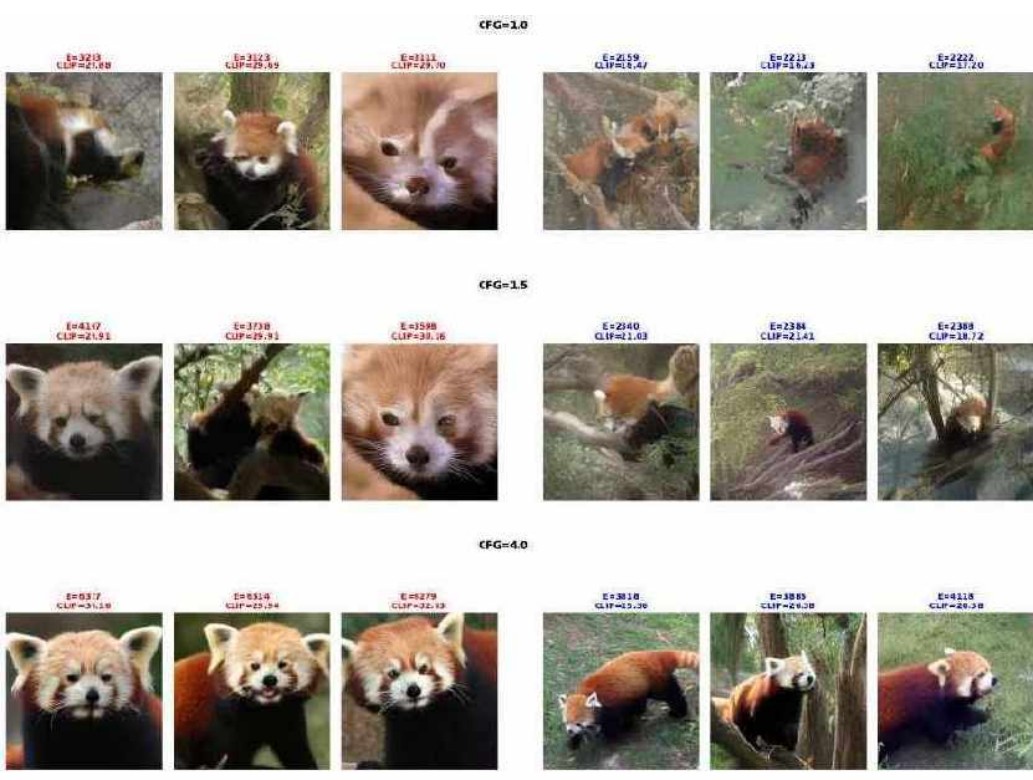

*Figure 15.* **African Elephant** (ImageNet-256): High-KPE (left) vs. low-KPE (right) across CFG scales 1.0, 1.5, 4.0. Higher KPE shows more defined features and better skin texture.

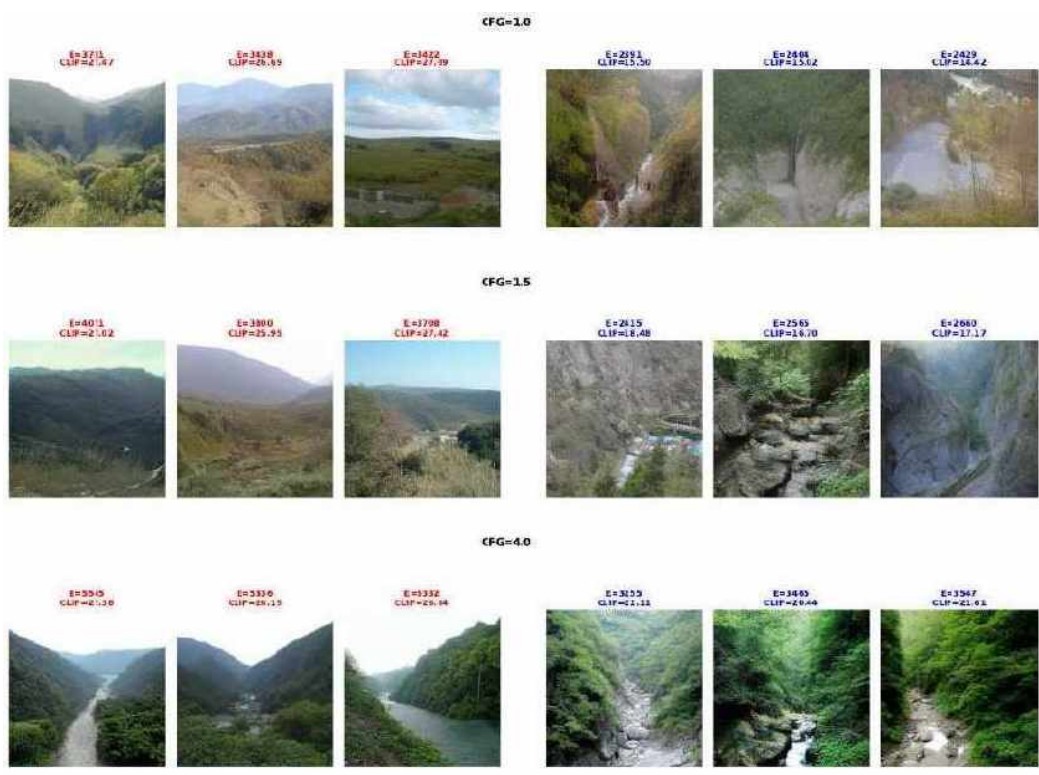

*Figure 16.* **Valley** (ImageNet-256): High-KPE (left) vs. low-KPE (right) across CFG scales 1.0, 1.5, 4.0. Higher KPE generates more detailed terrain and better depth perception.

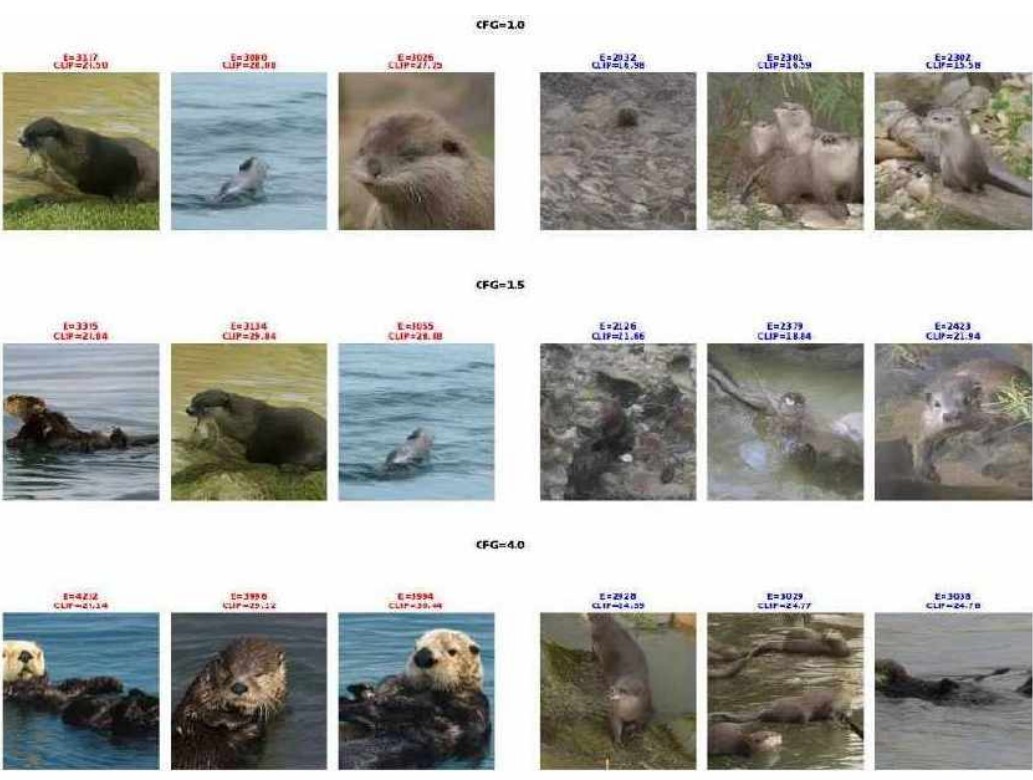

*Figure 17.* **Otter** (ImageNet-256): High-KPE (left) vs. low-KPE (right) across CFG scales 1.0, 1.5, 4.0. Higher KPE shows sharper outlines and more realistic details.

