# OpenReview forum: "A Kinetic Energy Perspective of Flow Matching"
_ICML.cc/2026/Conference — ICML 2026 spotlight_

### Official Review · Reviewer_57oT · 2026-03-09

**Soundness:** 2
**Presentation:** 4
**Significance:** 3
**Originality:** 4
**Overall Recommendation:** 5
**Confidence:** 3

**Summary:**

This article proposes to study the ‘kinetic" energy along sampling trajectories in flow matching algorithms. The authors derive the KPE (Kinetic path energy) metric to investigate samples. Empirically, higher KPE is associated with semantically stronger images and with endpoints lying in low-density regions of the data manifold. However, the authors show that too high KPE lead to memorized sample. Using these observations, the author design a novel sampling schedule with high kinetic energy at the beginning of the sampling trajectory, and fewer later. The resulting sampler decrease the number of memorized sampled, while keeping a good FID.

**Compliance With Llm Reviewing Policy:**

Affirmed.

**Final Justification:**

The rebuttal addressed my concern, I have update my score accordingly

**Key Questions For Authors:**

Questions:

1. How stable is KPE as a per-sample ranking? If you recompute KPE for the same generated sample using different NFE (e.g., 50/100/200) or a higher-order solver, do the relative KPE rankings (low/mid/high bins) stay consistent?
2. Why the particular KTS functional form? The linear launch + exponential damping is plausible, but it would help to justify why those forms (vs. e.g. smooth polynomial splines, cosine schedules, or a constrained optimization over \eta(t) with a fixed energy budget).
3. Memorization measurement details: What feature space is used for k-NN neighbor search and the “gap” (pixel space? embeddings?), and how sensitive are results to that choice? Please provide a fully specified recipe.

**Limitations:**

For me the main limitation is that KPE depends on the numerical ODE solver (NFE/step size/order), so trends could partly reflect discretization. This point should more thoroughly discussed.

**Strengths And Weaknesses:**

Strength :
1. Clear conceptual contribution: KPE is an interpretable scalar diagnostic tied to the trajectory rather than just endpoint metrics; this is a useful perspective for debugging FM failure modes.
2. Presentation: the article is clearly written, with plenty of meaningful experiments, and it reads well. In addition the empirical story is coherent (high energy helps until it hurt), and properly backed by experiments.
3. KTS is simple but effective : Just a simple scheduling of the scheduling seems to mitigate memorization problems while keeping high quality samples.
4. Originality: I have enjoyed the physical interpretation of the flow mathcing process, this is not so common in the field.

Weaknesses:
1. Major : Impact of discretization/solver choices: As presented by the authors, the KPE is computed along a discretized ODE solve (Euler/midpoint, fixed NFE in several experiments). Since energy is an integral over \|v(t)\|^2, it can be sensitive to timestep, solver type. However, there is no experiments showing the impact of both (time-steps, solver type), to demonstrate that the shown results still hold in various setting.
2. Major: KTS is not benchmarked against other known technics mitigating memorization risks (i.e. noise injection, adaptative solvers,  stochastic sampler…).
3. About the density estimation : The paper acknowledges that density estimates are computed in a handcrafted descriptor/PCA space and are not calibrated pixel-manifold densities. While the trends are interesting, the conclusions should be phrased more cautiously (e.g., “low support in descriptor space” rather than “low density regions of the data manifold”), and sensitivity to the descriptor choice would help. Also, there are plenty of deep learning algorithms that seems a better choice for density estimation on image dataset (e.g. energy-based model could give a more accurate proxy of unormalized density).
4. Minor (but important for clarity) : the memorization metric is not well enough explained: there is not formula/algorithms in the main text for the gap ratio and the exact decision rule; this makes it hard to assess robustness or reproduce without chasing references. At minimum, provide the explicit formula, the feature space used for neighbors, and how ties / preprocessing are handled.

---

> ### Author Rebuttal · Authors · 2026-03-30
>
> ## W1 + Q1: Impact of discretization/solver choices on KPE
> **(a) KPE rankings are near-invariant.** We compute per-sample KPE under 6 configs: {Euler, Heun} × NFE $\in$ {50, 100, 200}, $n=2000$.
>
> **Table 9: Pairwise Spearman $\rho$ of per-sample KPE rankings.**
>
> |                   | Euler 50 | Euler 100 | Euler 200 | Heun 50 | Heun 100 | Heun 200 |
> |-------------------|:--------:|:---------:|:---------:|:-------:|:--------:|:--------:|
> | **Euler 50**  | 1.000 | 0.998 | 0.995 | 0.993 | 0.992 | 0.992 |
> | **Euler 100** | 0.998 | 1.000 | 0.999 | 0.998 | 0.998 | 0.997 |
> | **Euler 200** | 0.995 | 0.999 | 1.000 | 0.999 | 0.999 | 0.999 |
> | **Heun 50**   | 0.993 | 0.998 | 0.999 | 1.000 | 1.000 | 1.000 |
> | **Heun 100**  | 0.992 | 0.998 | 0.999 | 1.000 | 1.000 | 1.000 |
> | **Heun 200**  | 0.992 | 0.997 | 0.999 | 1.000 | 1.000 | 1.000 |
>
> **Conclusion:** All pairwise $\rho \geq 0.992$, confirming KPE rankings are solver-invariant.
>
> **(b) KTS improvements hold across solvers, NFE, and schedules.**
>
> **Table 10:** CelebA, $\alpha_0 = \beta_0 = 0.01$, $\tau_{\text{split}}=0.6$.
>
> | Solver | NFE | Schedule | FID (FM) | FID (KTS) | $\Delta$FID | $F_{\text{mem}}$ (FM) | $F_{\text{mem}}$ (KTS) | $\Delta F_{\text{mem}}$ |
> |:-------|:---:|:--------:|:---:|:---:|:---:|:---:|:---:|:---:|
> | Euler    | 100 | uniform | 16.68 | 14.35 | −2.33 | 37.34 | 31.22 | −6.12 |
> | Euler    | 250 | uniform | 13.11 | 12.87 | −0.24 | 37.83 | 30.23 | −7.60 |
> | Midpoint | 100 | uniform | 19.20 | 18.88 | −0.32 | 37.86 | 30.19 | −7.67 |
> | Midpoint | 250 | uniform | 15.80 | 15.00 | −0.80 | 37.68 | 29.72 | −7.96 |
> | Euler    | 100 | cosine  | 15.21 | 15.20 | −0.01 | 38.19 | 28.54 | −9.65 |
> | Midpoint | 100 | cosine  | 15.42 | 15.36 | −0.06 | 37.80 | 30.84 | −6.96 |
>
> **Conclusion:** KTS reduces $F_{\text{mem}}$ by 6–10 pp across all configurations.
>
> ---
> ## W2: KTS not benchmarked against competing memorization-mitigation methods
> The primary purpose of KTS is to validate our analytical findings, rather than to serve as a standalone method. Our analysis suggests that early kinetic energy supports semantic refinement, while terminal energy spikes drive memorization.  We compare against noise injection (CelebA, $N=100$, 10k samples):
>
> **Table 11:** KTS vs. competing methods.
>
> | Method | FID@10k | $F_{\text{mem}}$ (%) |
> |:-------|:---:|:---:|
> | FM Baseline (Euler ODE) | 16.68 | 37.34 |
> | + Noise injection ($\sigma=0.005$) | 17.48 | 35.31 |
> | + Noise injection ($\sigma=0.01$) | 37.33 | 33.25 |
> | **KTS ($\alpha_0{=}\beta_0{=}0.01$)** | **14.35** | **31.22** |
>
> **Conclusion:** KTS achieves lower FID and $F_{\text{mem}}$, outperforming noise injection.
>
> ---
> ## W3: Density estimation concerns
>
> **1. Phrasing.** Agreed, we will revise to "local sparsity in representation space" in revision.
>
> **2. Descriptor sensitivity.**
>
> **Table 12:** ImageNet-256, $NFE=10$, CFG=1.5, $n=4{,}000$, k-NN ($k=50$).
>
> | Feature Space | Dim | Pearson $r$ | Spearman $\rho$ |
> |:---|:---:|:---:|:---:|
> | Descriptors+PCA | 22→2 | −0.34 | −0.38 |
> | Descriptors | 22 | −0.66 | −0.67 |
> | VAE + PCA | 4096→10 | -0.69 | -0.72|
> | VAE + PCA | 4096→22 | −0.72 | -0.74|
>
> The negative correlation between local sparsity and KPE holds across all feature spaces.
>
> **Table 13:** k-NN k-neighbor sensitivity (Descriptors, CIFAR-10).
>
> | $k$ | $\rho$ (NFE=150) | $\delta$ (NFE=150) | $\rho$ (NFE=10) | $\delta$ (NFE=10) |
> |:---:|:---:|:---:|:---:|:---:|
> | 5 | −0.57 | −0.87 | −0.44 | −0.70 |
> | 10 | −0.62 | −0.91 | −0.59 | −0.87 |
> | 20 | −0.63 | −0.92 | −0.60 | −0.86 |
> | 50 | −0.65 | −0.93 | −0.54 | −0.83 |
> | 100 | −0.63 | −0.91 | −0.56 | −0.83 |
>
> **Conclusion (Tables 12-13):** Negative correlation robust across feature spaces and stable across $k \in [5, 100]$.
>
> **3. EBMs for density.** Our claim is a *relative ranking*, not calibrated density. k-NN/KDE is standard for this (Kynkäänniemi et al., 2019); EBMs are noted as future work.
>
> ---
> ## W4 + Q3: On memorization metric
> A sample $\mathbf{x}$ is memorized if:
>
> $$r_{\text{gap}} = || \mathbf{x} - \mathbf{a}^{\mu_1} ||_2 / || \mathbf{x} - \mathbf{a}^{\mu_2} ||_2 < 1/3,$$
>
> where $\mathbf{a}^{\mu_1}, \mathbf{a}^{\mu_2}$ are nearest and second-nearest training neighbors in $L_2$ pixel space (Yoon et al., 2023; Gu et al., 2025; Bonnaire et al., 2025). The memorization fraction is $F_{\text{mem}} = \frac{1}{n} \sum_{i=1}^{n} \mathbf{1}[r_{\text{gap}}^{(i)} < 1/3].$
> Will add these to main text.
>
> ---
> ## Q2: Justification of the KTS functional form
>
> **Table 14:** $\tau_{\text{split}}=0.6$, $\alpha_0=\beta_0=0.01$, Euler, $N=100$.
>
> | Early $\alpha(t)$ | Late $\beta(t)$ | FID@10k | $F_{\text{mem}}$ (%) |
> |:---:|:---:|:---:|:---:|
> | Baseline (no KTS) | — | 16.68 | 37.34 |
> | linear (paper) | exponential (paper) | 14.35 | **31.22** |
> | linear | constant |12.47 | 35.50 |
> | linear | linear |11.91 |36.73 |
> |constant | linear | **11.72** | 37.15 |
> |exponential | linear | 11.93 | 36.76 |
>
> **Conclusion:** All two-phase designs improve over baseline, confirming the gain is structural.

---

> > ### Author Rebuttal · Reviewer_57oT · 2026-04-01
> >
> > The concern concerning discretization and the type of solver are fully resolved. I will increase my rate accordingly.

---

> > > ### Author Response · Authors · 2026-04-06
> > >
> > > We appreciate your careful review and are glad that we addressed your concerns. Thank you for your time and consideration.

---

### Official Review · Reviewer_JwFo · 2026-03-11

**Soundness:** 3
**Presentation:** 4
**Significance:** 3
**Originality:** 3
**Overall Recommendation:** 5
**Confidence:** 4

**Summary:**

This paper analyzes Flow Matching from a physics perspective, introducing Kinetic Path Energy as a per-sample diagnostic metric to evaluate generation trajectories. The study reveals that higher KPE improves semantic fidelity, and trajectories reaching low-density regions consume more energy. However, due to a velocity singularity in empirical flow matching as $t \to 1$, extremely high energy paradoxically leads to memorization rather than better quality. Based on this Goldilocks Principle, the authors propose Kinetic Trajectory Shaping, a training-free inference strategy. By using an early Launch phase to accelerate exploration and a late ``Soft-Landing'' phase to suppress the singularity, KTS effectively improves generation quality and significantly reduces memorization across multiple datasets.

**Compliance With Llm Reviewing Policy:**

Affirmed.

**Final Justification:**

This manuscript clearly meets the acceptance criteria. It introduces novel findings regarding flow matching, provides a reasonable degree of theoretical justification, and proposes an effective solution to the identified issues. I especially appreciate the convincing way these findings are presented; the manuscript is very well-written.

**Key Questions For Authors:**

1. Generality of $\tau_{split}$: KTS heavily relies on the hyperparameter $\tau_{split} = 0.6$. Given that different datasets and network architectures may exhibit entirely different velocity field dynamics, how was this specific value determined? For practical applications, is there a viable adaptive method to dynamically trigger the ``Soft-Landing'' phase (e.g., by monitoring the first derivative of the instantaneous power $||v(t)||^2$ in real-time)?

2. Distributional shift from KTS: By scaling the velocity field with $\eta(t)$, the generated distribution mathematically loses strict equivalence to the target data distribution. Can the authors provide mathematical intuition or a formal argument explaining why this non-canonical velocity scaling avoids severely shifting the final generated distribution? Alternatively, what specific constraints must be imposed on the parameters within $\eta(t)$ to prevent such deviations?

**Limitations:**

Yes. The authors have adequately discussed the limitations and potential negative societal impact of their work.

**Strengths And Weaknesses:**

Strengths:

1. Excellent structure and narrative flow: The paper logically transitions from intuitive physical metaphors (kinetic energy) to empirical observations, mathematical proofs, and a practical engineering solution, making it a highly engaging read.

2. Strong and clear empirical validation: The experimental results are comprehensive and compelling, supported by well-designed and easy-to-interpret figures.

Weaknesses:

1. Lack of ablation on $\tau_{split}$: The paper does not provide experimental support (e.g., evaluating FID and $F_{mem}$ across varying values) to justify the optimal selection of the phase split hyperparameter $\tau_{split}$, nor does it assess the model's sensitivity to it.

2. A core claim of this paper is the existence of a non-monotonic relationship between kinetic energy and generation quality (i.e., too low leads to underfitting, moderate is optimal, and too high induces memorization). However, the current theoretical analyses (Theorem 4.2 and Proposition 5.2) only independently establish the 'negative correlation between energy and data density' and the 'terminal energy blow-up of EFM.' They fail to mathematically bridge these two findings to formally prove this 'non-monotonic, inverted U-shaped relationship.' Could the authors provide a more rigorous mathematical argument demonstrating that this non-linear/non-monotonic relationship is an inherent property of the continuous time evolution, rather than merely a byproduct of empirical observations?

---

> ### Author Rebuttal · Authors · 2026-03-30
>
> ## W1: Lack of ablation on $\tau$ +  Q1: Generality of $\tau$ and adaptive methods
>
> We appreciate this concern. We have conducted a comprehensive ablation on $\tau_{\text{split}}$ and schedule functional form.
>
> ### Table 6: $\tau_{\text{split}}$ sensitivity ###
> (Fixed: $\alpha_0=0.01, \beta_0=0.01$, Euler, $N=100$, uniform schedule.)
>
> | $\tau_{\text{split}}$ | FID@10k | $F_{\text{mem}}$ (%) |
> |:-----:|:-------:|:---:|
> | 0.2 | 60.31 | 23.66 |
> | 0.4 | 48.58 | 27.15 |
> | **0.6** | **14.35** | **31.22** |
> | 0.8 | 21.07 | 34.30 |
>
> **Conclusion:** $\tau=0.6$ is optimal. As shown in Figure 6, terminal spikes begin at $t \in [0.50, 0.70]$; $\tau=0.6$ is a conservative midpoint of this onset range.
>
> Additionally, we provide an ablation on the schedule functional form:
> ### Table 7: Schedule functional form ###
> (Fixed: $\tau_{\text{split}}=0.6$, $\alpha_0=0.01, \beta_0=0.01$, Euler, $N=100$.)
>
> | Early $\alpha(t)$ | Late $\beta(t)$ | FID@10k | $F_{\text{mem}}$ (%) |
> |:---:|:---:|:---:|:---:|
> | Baseline (no KTS) | — | 16.68 | 37.34 |
> | linear (paper) | exponential (paper) | 14.35 | **31.22** |
> | linear | constant |12.47 | 35.50 |
> | linear | linear |11.91 |36.73 |
> |constant | linear | **11.72** | 37.15 |
> |exponential | linear | 11.93 | 36.76 |
>
> **Conclusion:** All two-phase designs improve over baseline, confirming the gain is structural. We will include both tables in the revision.
>
> ---
> ## W2: Missing formal proof of the non-monotonic relationship
>
> Thank you for this question. We agree that a formal proof of the inverted-U relationship would require a rigorous definition of "quality" tied to continuous-time dynamics, which is beyond the current scope and an interesting future direction.
>
> What we do provide is a rigorous characterization of two complementary regimes: (i) higher kinetic energy → traversal through lower-density regions (under posterior dominance), and (ii) sufficiently large terminal energy → memorization via the $1/(1-t)$ singularity. Together, these establish a non-monotonic mechanism: increasing energy is beneficial up to a point, beyond which it leads to degeneration.
>
> ---
> ## Q2: Distributional shift from KTS
>
> KTS modifies the velocity as $\tilde{v}(x_t, t) = \eta(t)\, v_\theta(x_t, t)$ with $\eta(t) > 0$, so it preserves the direction and only rescales the magnitude. We argue the shift is bounded via three observations:
>
> **Small perturbation regime.** In our balanced setting $(\alpha_0, \beta_0) = (0.01, 0.01)$, we have:
>   - Early phase: $\eta(t) = 1 + \alpha(t)$ with $\max_t \alpha(t) = 0.01$, so the velocity is modified by at most one percent, and the boost decays to zero at $t = \tau$.
>   - Late phase: $\eta(t) = 1 - \beta(t)$, with $\beta(t) = \beta_0 \big(e^{k(t-\tau)} - 1\big)$, $\beta_0 = 0.01$, and $k=3$, giving $\beta(1) = 0.01\big(e^{3(1-\tau)} - 1\big) \approx 0.023$. This damping targets the regime where the velocity is already pathological: the $1/(1-t)$ singularity produces excessively large velocities near $t = 1$. Suppressing this overshoot reduces terminal instability.
>
> **Stability bound (via a Grönwall-type argument).** Let $x(t)$ and $\tilde{x}(t)$ solve
> $\dot{x}(t) = v_\theta(x(t), t)$ and
> $\dot{\tilde{x}}(t) = \eta(t),  v_\theta(\tilde{x}(t), t)$ respectively, with $x(0)=\tilde{x}(0)$. Define $\Delta(t) = \tilde{x}(t) - x(t)$.
> Assume that $v_\theta(\cdot,t)$ is uniformly Lipschitz in $x$, i.e. $$||v_\theta(x,t) - v_\theta(y,t)|| \le L ||x - y||
> \quad \text{for all } x,y,t. $$
>
> Then, for almost every $t$,
> $$\frac{d}{dt}||\Delta(t)|| \le L |\eta(t)|  ||\Delta(t)|| + |\eta(t)-1|  ||v_\theta(x(t),t)||.$$
>
> Applying Grönwall’s inequality yields
> $$||\tilde{x}(1) - x(1)|| \le \exp\left(L \int_0^1 |\eta(s)| \ ds\right) \int_0^1 |\eta(t)-1|  ||v_\theta(x(t),t)|| \ dt.$$
> The terminal shift is controlled (up to amplification governed by the Lipschitz constant) by the weighted deviation $\int_0^1 |\eta(t)-1|  ||v_\theta(x(t),t)|| \ dt$, which could be controlled by design (choosing $\alpha_0, \beta_0 \ll 1$).
>
> **Empirical confirmation.**
>
> **Table 8:** Precision/Recall on ImageNet.
>
> | Setting | Precision | Recall |
> |:---|:---:|:---:|
> | FM Baseline | 0.728 | 0.655 |
> | Balanced ($\alpha_0{=}\beta_0{=}0.01$) | 0.729 | 0.653 |
>
> **Conclusion:** Near-identical values confirm KTS does not meaningfully distort the generated distribution.

---

> > ### Author Rebuttal · Reviewer_JwFo · 2026-04-01
> >
> > I appreciate the authors’ analysis of distributional shift and the additional numerical results. I especially appreciate the paper’s new findings on flow matching and the convincing way these findings are presented. I will maintain my evaluation of the paper and recommend acceptance.

---

> > > ### Author Response · Authors · 2026-04-06
> > >
> > > Thank you for your careful review and for acknowledging that your concerns have been satisfactorily addressed. We greatly appreciate your time and consideration.

---

### Official Review · Reviewer_ttmp · 2026-03-13

**Soundness:** 3
**Presentation:** 3
**Significance:** 3
**Originality:** 3
**Overall Recommendation:** 5
**Confidence:** 4

**Summary:**

The paper analyzes the sampling process of flow matching generative models from the perspective of trajectory-level behavior, rather than focusing only on the terminal distribution. The authors interpret ODE sampling trajectories as the motion of particles in a velocity field and, inspired by the kinetic energy / action viewpoint in classical mechanics, propose a new per-sample diagnostic quantity, KPE, to measure the accumulated kinetic cost along an individual generation trajectory. Building on this quantity, the paper mainly investigates the relationship between KPE and the properties of generated samples, and presents three main findings.

First, the authors show empirically that higher KPE is typically associated with stronger semantic fidelity and better class distinguishability: samples with higher KPE achieve better performance in terms of CLIP score and CLIP margin. Second, the paper explores the relationship between KPE and data density from both empirical and theoretical perspectives, showing that high-KPE trajectories tend to terminate in low-density regions of the data manifold, and further provides an analysis under EFM framework linking squared velocity to negative log-density. Third, the authors find that this relationship between higher KPE and better generation is not monotonic. In extremely high-energy cases, especially when energy spikes appear near the terminal stage, the generation process degenerates into a memorization behavior that is close to reproducing training samples. Based on this observation, the paper proposes a training-free inference-time method, KTS, which modulates the velocity in two stages—enhancing it in the early phase and suppressing it in the late phase—to improve generation quality while reducing memorization.

Overall, the paper aims to establish a complete chain from trajectory diagnosis to sampling control, offering a new perspective for understanding sample quality, rarity, and memorization phenomena in flow matching.

**Compliance With Llm Reviewing Policy:**

Affirmed.

**Final Justification:**

Thank you for the authors’response. The rebuttal provides additional experimental analysis and clarifications. Based on these additions, I believe that my main concerns have been addressed. I will accordingly raise my score to 5 and recommend acceptance.

**Key Questions For Authors:**

1. Regarding the conclusion about low-density regions, I would like the authors to clarify this point more carefully.The theory in the paper discusses probability density in a relatively standard sense, but in the real-image experiments, the quantity actually used to estimate “density” is computed by first extracting features and reducing dimensionality, and then applying k-NN/KDE in that representation space. This choice is understandable, but the two are ultimately not exactly the same thing. Could the authors clarify more explicitly to what extent this “density proxy” used in the experiments can represent the notion of density discussed in the theory? In addition, if a different feature representation were used, would this negative correlation still remain stable?

2. The theoretical analysis of terminal energy spikes and memorization is mainly carried out under the closed-form solution of empirical flow matching, and I think this part is valuable. However, the actual experiments are conducted with neural-network-parameterized flow matching models, and there is still a gap between the two. Could the authors further explain why the mechanism identified in EFM can also account for the phenomenon observed in neural FM? Is there more direct evidence showing that the memorization observed in neural FM is indeed mainly associated with similar terminal high-energy / high-velocity behavior?

3. Regarding KTS, I would like to see a more complete robustness analysis. The overall intuition behind KTS is easy to understand: push more in the early stage, and pull back somewhat in the later stage. However, its current specific form—for example, how the split point is chosen, why linear enhancement is used in the first stage, and why exponential damping is used in the second stage—still appears fairly empirical. Could the authors provide a more systematic sensitivity analysis, for example, by showing whether the results remain stable as parameters vary?

4. KTS changes the velocity distribution during sampling, so its effect may not be limited to quality and memorization; it may also affect the coverage of the generated samples. Could the authors further analyze whether early-stage enhancement and late-stage damping might, in some cases, make the samples collapse more easily into a narrower set of modes?

**Limitations:**

Yes.

**Strengths And Weaknesses:**

Strengths：

1.	The paper addresses a worthwhile problem. It analyzes sample-wise sampling trajectories in flow matching. This perspective is valuable for understanding differences in sample quality, behavior in low-density regions, and memorization phenomena.

2.	The proposed KPE is simple in form. It is built directly on ODE sampling trajectories and the velocity field, incurs very low computational cost, and remains fairly interpretable. Although the formula itself is not complicated, it provides a useful trajectory-level analysis tool for flow matching.

3.	The empirical results support two interesting findings. On the one hand, samples with higher KPE tend to exhibit stronger semantic alignment and better class separability. On the other hand, higher KPE shows a stable negative correlation with low-density regions. Both observations are quite insightful.

4.	The non-monotonic view that higher energy is not always better is novel. The paper does not stop at the simple empirical conclusion that higher KPE corresponds to higher quality, but further points out that when the energy becomes excessively high, the behavior shifts from stronger semantics toward memorization.

5.	The KTS method is lightweight and shows some practical potential. KTS does not require retraining the model and only modulates the velocity during inference, so its engineering cost is low. The experiments also suggest that it provides a tunable trade-off between quality and memorization.


Weaknesses：

1.	Regarding the relationship between KPE and low-density regions, I think the authors could be a bit more cautious in the way this is phrased for the real-image experiments. The theory discusses density in a more standard sense, whereas the “density” used in the experiments is actually a proxy obtained from descriptor features, followed by PCA dimensionality reduction, and then estimated with k-NN/KDE. This choice itself is acceptable. My concern is not that the authors use a proxy, but that the distinction should be made more explicit in the main text. Based on the current evidence, a more precise statement may be that KPE is correlated with local sparsity in a particular representation space, rather than already demonstrating a direct correspondence to data-manifold density in the stronger sense.

2.	Although the current experimental results suggest it is promising, the paper still looks more like a well-motivated inference-time heuristic. For example, the choice of the split point, the use of linear amplification in the early stage, and exponential damping in the later stage all seem to contain a fairly strong empirical component. The current experiments can support that this design works under the authors’ present setting, but they are not yet sufficient to fully convince me that the same stable gains would hold across a wider parameter range or under other settings.

---

> ### Author Rebuttal · Authors · 2026-03-30
>
> ## W1: Phrasing issue: density proxy vs. data-manifold density
> Agreed. We will revise to distinguish theoretical density from the experimental proxy, adopting: "KPE correlates with local sparsity in representation space."
>
> ---
> ## W2 + Q3: More complete KTS robustness analysis
> We have conducted comprehensive ablation studies:
> - **$\tau_{\text{split}}$ sensitivity:**
>
> **Table 5a:** $\tau_{\text{split}}$ sensitivity ($\alpha_0=\beta_0=0.01$, Euler, $NFE=100$, uniform).
>
> | $\tau_{\text{split}}$ | FID@10k | $F_{\text{mem}}$ (%) |
> |:-----:|:-------:|:---:|
> | 0.2 | 60.31 | 23.66 |
> | 0.4 | 48.58 | 27.15 |
> | **0.6** | **14.35** | **31.22** |
> | 0.8 | 21.07 | 34.30 |
>
> **Conclusion:** $\tau=0.6$ is optimal; earlier splits hurt FID, later splits miss the spike. As shown in Figure 6, the instantaneous power $||v(t)||^2$ curves exhibit terminal spikes beginning at $t \in [0.50, 0.70]$. We chose $\tau=0.6$ as a conservative midpoint.
>
> - **Schedule functional form:**
>
> **Table 5b:** $\tau_{\text{split}}=0.6$, $\alpha_0=\beta_0=0.01$, Euler, $NFE=100$.
>
> | Early $\alpha(t)$ | Late $\beta(t)$ | FID@10k | $F_{\text{mem}}$ (%) |
> |:---:|:---:|:---:|:---:|
> | Baseline (no KTS) | — | 16.68 | 37.34 |
> | linear (paper) | exponential (paper) | 14.35 | **31.22** |
> | linear | constant |12.47 | 35.50 |
> | linear | linear |11.91 |36.73 |
> |constant | linear | **11.72** | 37.15 |
> |exponential | linear | 11.93 | 36.76 |
>
> **Conclusion:** All two-phase designs improve over baseline, confirming the gain is structural.
>
> - **Solver/NFE robustness:**
>
> **Table 5c:** CelebA, $\alpha_0 = \beta_0 = 0.01$, $\tau_{\text{split}}=0.6$.
>
> | Solver | NFE | Schedule | FID (FM) | FID (KTS) | $\Delta$FID | $F_{\text{mem}}$ (FM) | $F_{\text{mem}}$ (KTS) | $\Delta F_{\text{mem}}$ |
> |:-------|:---:|:--------:|:---:|:---:|:---:|:---:|:---:|:---:|
> | Euler    | 100 | uniform | 16.68 | 14.35 | −2.33 | 37.34 | 31.22 | −6.12 |
> | Euler    | 250 | uniform | 13.11 | 12.87 | −0.24 | 37.83 | 30.23 | −7.60 |
> | Midpoint | 100 | uniform | 19.20 | 18.88 | −0.32 | 37.86 | 30.19 | −7.67 |
> | Midpoint | 250 | uniform | 15.80 | 15.00 | −0.80 | 37.68 | 29.72 | −7.96 |
> | Euler    | 100 | cosine  | 15.21 | 15.20 | −0.01 | 38.19 | 28.54 | −9.65 |
> | Midpoint | 100 | cosine  | 15.42 | 15.36 | −0.66 | 37.80 | 30.84 | −6.96 |
>
> **Conclusion:** KTS consistently reduces $F_{\text{mem}}$ by 6–10 pp across all configurations.
>
> ---
> ## Q1: Does the negative correlation remain stable across feature representations?
> Yes. We validate across 4 feature spaces (Reply to Reviewer 1, Table 2): the negative KPE–density correlation holds robustly ($\rho$: −0.38 to −0.74). Additionally, k-NN $k$ sensitivity analysis (Reply to Reviewer 1, Table 3) shows stability across $k \in [5, 100]$.
>
> ---
> ## Q2: EFM theory → neural FM memorization gap
>
> **Role of EFM:** EFM is the closed-form optimum that neural FM is trained to approximate. We use it to make the energy–memorization link *provable* (Proposition 1), not to claim neural FM behaves identically.
>
> **Direct evidence on neural FM:** The CelebA training dynamics (Fig. 7a) show this on a neural U-Net: as training progresses, KPE rises ($\sim$15 → $\sim$540) and $F_{\text{mem}}$ rises ($\sim$0%→98%) in tandem, the same terminal high-velocity mechanism identified in EFM. For a comprehensive summary of all neural FM validations, see Reply to Reviewer 1, Table 1.
>
> ---
> ## Q4: Could KTS cause mode collapse?
> Our ImageNet experiments report Precision and Recall, which directly measure this:
>
> **Table 5d:** Precision/Recall under different KTS settings (ImageNet).
>
> | Setting | $\alpha_0$ | $\beta_0$ | Precision | Recall |
> |:---|:---:|:---:|:---:|:---:|
> | FM Baseline | 0 | 0 | 0.728 | 0.655 |
> | Quality-focused | 0.05 | 0 | 0.731 (+0.003) | 0.630 (−0.025) |
> | Coverage-focused | 0 | 0.05 | 0.721 (−0.007) | **0.657 (+0.002)** |
> | Balanced | 0.01 | 0.01 | 0.729 (+0.001) | 0.653 (−0.002) |
>
> **Conclusion:** Early-stage boost ($\alpha_0$) mildly narrows coverage (lower recall). However, late-stage damping ($\beta_0$) counteracts this by preventing trajectories from collapsing onto memorized points. The balanced setting achieves FID and CLIP improvements with essentially no change in precision-recall balance.

---

### Official Review · Reviewer_BuHf · 2026-03-13

**Soundness:** 2
**Presentation:** 2
**Significance:** 2
**Originality:** 2
**Overall Recommendation:** 4
**Confidence:** 3

**Summary:**

This paper introduces kinetic path energy, defined as the time integral of squared speed along ODE sampling trajectories in flow matching, and uses it as a per-sample trajectory metric to analyze how sampling dynamics relate to semantic quality and rarity. It provides an analyzable empirical flow matching theory linking kinetic energy to negative log density, characterizes a terminal energy spike mechanism associated with memorization.

**Compliance With Llm Reviewing Policy:**

Affirmed.

**Final Justification:**

The paper has a clear and interesting trajectory-level perspective, and the rebuttal substantially strengthened the empirical case through added robustness and sensitivity analyses on solver choice and discretization.

**Key Questions For Authors:**

- The energy density theory relies on a local dominance condition. Can you quantify how often this condition holds on your ImageNet model across time?

- Rarity and memorization results use feature-space nearest neighbor and density proxies. How sensitive are the reported correlations and memorization fractions to the choice of representation and distance?

- Under inference-only changes (ODE solver choice, step size schedule, or a simple time reparameterization) with the model fixed, does KTS still produce comparable improvements?

**Limitations:**

Yes.

**Strengths And Weaknesses:**

Strengths:

- The paper proposes a simple trajectory level energy metric for flow matching sampling and shows it consistently stratifies semantic alignment in the reported ImageNet experiments.

- The analysis offers a concrete mechanism for terminal time energy spikes and introduces a lightweight inference time rescaling to reduce these spikes.

Weaknesses:

- The theory is developed in an analyzable empirical setting where intermediate distributions are modeled as mixtures over training points. It is unclear how directly the conclusions transfer to typical neural flow matching models trained on large-scale data.

- The rarity and memorization evidence uses proxy measurements in a chosen feature space, which makes the robustness and attribution of gains remain uncertain.

---

> ### Author Rebuttal · Authors · 2026-03-30
>
> ## W1: Unclear transfer of EFM conclusions to neural FM
>
> All core findings are validated on neural FM. EFM provides the *provable explanation*, not the only evidence.
>
> **Table 1: Neural FM validation across all findings.** Every finding is confirmed on real datasets.
>
> | Finding | Model | Dataset | Key result |
> |---------|-------|---------|------------|
> | KPE ↔ semantics | SiT-XL/2 | ImageNet-256 | +2.75 CLIP score (low→high KPE), Cohen's d ∈ [0.45, 0.65], all p < 0.008 |
> | KPE ↔ density | OT-CFM | CIFAR-10 | Spearman ρ = −0.65 |
> | KPE ↔ density | SiT-XL/2 | ImageNet-256 | Spearman ρ ≈ −0.31 to −0.42 |
> | Energy paradox | U-Net FM | CelebA | KPE: 15→540, F_mem: ~0%→98% as training progresses (Fig. 7a) |
> | KTS method | U-Net FM | CelebA | FID: 16.68→14.35, F_mem: 37.3%→31.2% |
> | KTS method | SiT-XL/2 | ImageNet-256 | FID: 11.70→11.63, CLIP: improved |
>
>
> ---
> ## W2 + Q2: Rarity and memorization evidence use proxy measurements
>
> **Rarity (density) — proxy, but well-validated.** (1) k-NN and KDE give nearly identical correlations (ρ = −0.65 vs. −0.64, Table 2); (2) on three synthetic datasets with *exact* ground-truth density, the same inverse relationship holds (Figure 3, p < 10⁻³); (3) we validate across diverse feature spaces:
>
> **Table 2: Feature space sensitivity.** ImageNet-256, $N=10$, CFG=1.5, $n=4{,}000$, k-NN ($k=50$).
>
> | Feature Space | Dim | Pearson $r$ | Spearman $\rho$ |
> |:---|:---:|:---:|:---:|
> | Descriptors+PCA | 22→2 | −0.34 | −0.38 |
> | Descriptors | 22 | −0.66 | −0.67 |
> | VAE + PCA | 4096→10 | -0.69 | -0.72|
> | VAE + PCA | 4096→22 | −0.72 | -0.74|
>
>
> **Table 3: k-NN k sensitivity** (Descriptors, CIFAR-10).
>
> | $k$ | $\rho$ (NFE=150) | $\delta$ (NFE=150) | $\rho$ (NFE=10) | $\delta$ (NFE=10) |
> |:---:|:---:|:---:|:---:|:---:|
> | 5 | −0.57 | −0.87 | −0.44 | −0.70 |
> | 10 | −0.62 | −0.91 | −0.59 | −0.87 |
> | 20 | −0.63 | −0.92 | −0.60 | −0.86 |
> | 50 | −0.65 | −0.93 | −0.54 | −0.83 |
> | 100 | −0.63 | −0.91 | −0.56 | −0.83 |
>
> **Conclusion:** Tables 2–3 show the negative KPE–density correlation is robust across feature spaces ($\rho$: −0.38 to −0.74) and stable across $k \in [5, 100]$.
>
> **Memorization (F_mem) — not a proxy.** F_mem is computed directly in **pixel space** with no feature extraction: a sample $\mathbf{x}$ is memorized if $||\mathbf{x} - \mathbf{a}^{\mu_1}||_2 / ||\mathbf{x} - \mathbf{a}^{\mu_2}||_2 < 1/3$, where $\mathbf{a}^{\mu_1}, \mathbf{a}^{\mu_2}$ are its nearest and second-nearest training neighbors in L₂. This follows the standard protocol of [1,2,3]. Figure 7(b) further confirms this visually: late-checkpoint samples are near-identical copies of training images.
>
> **Attribution of KTS gains is independent of the density proxy.** KTS is evaluated via FID and $F_{\text{mem}}$ (pixel-space), neither depends on density estimation. The reported gains (FID: 16.68→14.35, $F_{\text{mem}}$: 37.3%→31.2%) stand regardless of density proxy precision.
>
> - [1] Diffusion probabilistic models generalize when they fail to memorize
>
> - [2] On memorization in diffusion models
>
> - [3] Why Diffusion Models Don't Memorize
>
> ---
> ## Q1: How often does the posterior dominance condition hold on ImageNet?
> The posterior dominance assumption does not hold at early times when the intermediate distribution is diffuse and many training samples overlap. However, as $t \to 1$ the posterior weights collapse onto the nearest training example, making dominance increasingly accurate. Since the key phenomena we analyze, terminal energy blow-up and memorization are driven by precisely this late-time regime. Thus, while the assumption is not globally valid, it is well-justified in the regime that governs the behavior of interest.
>
> ---
> ## Q3: Does KTS still work under different inference settings?
>
> We ablate across solvers (Euler, midpoint), NFE (100, 250), and time schedules (uniform, cosine):
>
> **Table 4: KTS robustness across inference settings.**
>
> | Solver | NFE | Schedule | FID (FM) | FID (KTS) | $\Delta$FID | $F_{\text{mem}}$ (FM) | $F_{\text{mem}}$ (KTS) | $\Delta F_{\text{mem}}$ |
> |:-------|:---:|:-------------:|:---:|:---:|:---:|:---:|:---:|:---:|
> | Euler    | 100 | uniform | 16.68 | 14.35 | −2.33 | 37.34 | 31.22 | −6.12 |
> | Euler    | 250 | uniform | 13.11 | 12.87 | −0.24 | 37.83 | 30.23 | −7.60 |
> | Midpoint | 100 | uniform | 19.20 | 18.88 | −0.32 | 37.86 | 30.19 | −7.67 |
> | Midpoint | 250 | uniform | 15.80 | 15.00 | −0.80 | 37.68 | 29.72 | −7.96 |
> | Euler    | 100 | cosine  | 15.21 | 15.20 | −0.01 | 38.19 | 28.54 | −9.65 |
> | Midpoint | 100 | cosine  | 15.42 | 15.36 | −0.06 | 37.80 | 30.84 | −6.96 |
>
> **Conclusion:** KTS consistently reduces memorization ($\Delta F_{\text{mem}}$: −6.12 to −9.65) across all six configurations, with FID comparable or improved.

---

> > ### Author Rebuttal · Reviewer_BuHf · 2026-04-04
> >
> > The rebuttal strengthens the paper and addresses several of my main empirical concerns. The added analyses make the empirical claims more convincing, and I will raise my score accordingly.

---

> > > ### Author Response · Authors · 2026-04-06
> > >
> > > Thank you for your thoughtful evaluation and for acknowledging our rebuttal. We sincerely appreciate your time and constructive feedback.

---

### Decision · Program_Chairs · 2026-04-30

**Decision:**

Accept (spotlight)

**Comment:**

This paper introduces Kinetic Path Energy (KPE) to analyze flow matching sampling trajectories, revealing a novel non-monotonic relationship where moderate energy enhances semantic fidelity while extreme energy induces memorization. Reviewers unanimously praised the clear physical perspective, strong empirical validation, and the practical utility of the proposed Kinetic Trajectory Shaping (KTS) method, noting that the authors successfully addressed all initial concerns regarding solver sensitivity and density proxies during the rebuttal phase.